# ON THE PROVABLE ADVANTAGE OF UNSUPERVISED PRETRAINING

**Jiawei Ge**[*†]     **Shange Tang** [*†]     **Jianqing Fan**[†]     **Chi Jin**[‡]

## ABSTRACT

Unsupervised pretraining, which learns a useful representation using a large amount of unlabeled data to facilitate the learning of downstream tasks, is a critical component of modern large-scale machine learning systems. Despite its tremendous empirical success, the rigorous theoretical understanding of why unsupervised pretraining generally helps remains rather limited—most existing results are restricted to particular methods or approaches for unsupervised pretraining with specialized structural assumptions. This paper studies a generic framework, where the unsupervised representation learning task is specified by an abstract class of latent variable models $\Phi$ and the downstream task is specified by a class of prediction functions $\Psi$. We consider a natural approach of using Maximum Likelihood Estimation (MLE) for unsupervised pretraining and Empirical Risk Minimization (ERM) for learning downstream tasks. We prove that, under a mild "informative" condition, our algorithm achieves an excess risk of $\tilde{\mathcal{O}}(\sqrt{\mathcal{C}_\Phi/m} + \sqrt{\mathcal{C}_\Psi/n})$ for downstream tasks, where $\mathcal{C}_\Phi, \mathcal{C}_\Psi$ are complexity measures of function classes $\Phi, \Psi$, and $m, n$ are the number of unlabeled and labeled data respectively. Comparing to the baseline of $\tilde{\mathcal{O}}(\sqrt{\mathcal{C}_{\Phi \circ \Psi}/n})$ achieved by performing supervised learning using only the labeled data, our result rigorously shows the benefit of unsupervised pretraining when $m \gg n$ and $\mathcal{C}_{\Phi \circ \Psi} > \mathcal{C}_\Psi$. This paper further shows that our generic framework covers a wide range of approaches for unsupervised pretraining, including factor models, Gaussian mixture models, and contrastive learning.

## 1 INTRODUCTION

Unsupervised pretraining aims to efficiently use a large amount of unlabeled data to learn a useful representation that facilitates the learning of downstream tasks. This technique has been widely used in modern machine learning systems including computer vision (Caron et al., 2019; Dai et al., 2021), natural language processing (Radford et al., 2018; Devlin et al., 2018; Song et al., 2019) and speech processing (Schneider et al., 2019; Baevski et al., 2020). Despite its tremendous empirical success, it remains elusive why pretrained representations, which are learned without the information of downstream tasks, often help to learn the downstream tasks.

There have been several recent efforts trying to understand various approaches of unsupervised pretraining from theoretical perspectives, including language models Saunshi et al. (2020); Wei et al. (2021), contrastive learning Arora et al. (2019); Tosh et al. (2021b;a); HaoChen et al. (2021); Saunshi et al. (2022), and reconstruction-based self-supervised learning Lee et al. (2021). While this line of works justifies the use of unsupervised pretraining in the corresponding regimes, many of them do not prove the advantage of unsupervised learning, in terms of sample complexity, even when compared to the naive baseline of performing supervised learning purely using the labeled data. Furthermore, these results only apply to particular approaches of unsupervised pretraining considered in their papers, and crucially rely on the specialized structural assumptions, which do not generalize beyond the settings they studied. Thus, we raise the following question: **Can we develop**

---

[*]equal contribution

[†]Department of Operations Research and Financial Engineering, Princeton University; {jg5300,shangetang,jqfan}@princeton.edu

[‡]Department of Electrical and Computer Engineering, Princeton University; chij@princeton.edu

**a generic framework which provably explains the advantage of unsupervised pretraining?**
This paper answers this highlighted question positively.

We consider the generic setup where the data $x$ and its label $y$ are connected by an unobserved representation $z$. Concretely, we assume $(x, z)$ is sampled from a latent variable model $\phi^*$ in an abstract class $\Phi$, and the distribution of label $y$ conditioned on representation $z$ is drawn from distributions $\psi^*$ in class $\Psi$. We consider a natural approach of using Maximum Likelihood Estimation (MLE) for unsupervised pretraining, which approximately learns the latent variable model $\phi^*$ using $m$ unlabeled data. We then use the results of representation learning and Empirical Risk Minimization (ERM) to learn the downstream predictor $\psi^*$ using $n$ labeled data. We remark that MLE is one of the most important underlying principle for designing unsupervised learning algorithms—a large number of modern unsupervised pretraining algorithms compute MLE or its proxies (such as optimizing the variational lower bound) due to computational constraints. These examples include contrastive learning (see Section 6), Variational AutoEncoder (VAE) (Kingma & Welling, 2013) and diffusion model (Sohl-Dickstein et al., 2015). Investigating this generic setup allows us to bypass the limitation of prior works that are restricted to the specific approaches for unsupervised pretraining.

We prove that, under a mild "informative" condition (Assumption 3.2), our algorithm achieves a excess risk of $\tilde{\mathcal{O}}(\sqrt{\mathcal{C}_\Phi/m} + \sqrt{\mathcal{C}_\Psi/n})$ for downstream tasks, where $\mathcal{C}_\Phi, \mathcal{C}_\Psi$ are complexity measures of function classes $\Phi, \Psi$, and $m, n$ are the number of unlabeled and labeled data respectively. Comparing to the baseline of $\tilde{\mathcal{O}}(\sqrt{\mathcal{C}_{\Phi \circ \Psi}/n})$ achieved by performing supervised learning using only the labeled data, our result rigorously shows the benefit of unsupervised pretraining when we have abundant unlabeled data $m \gg n$ and when the complexity of composite class $\mathcal{C}_{\Phi \circ \Psi}$ is much greater than the complexity of downstream task alone $\mathcal{C}_\Psi$.

Our generic framework enables a simple and standardlized approach to understand and analyze a wide range of unsupervised pretraining models. Consider the scenario where a new model of unsupervised pretraining is proposed, and we would like to evaluate the effectiveness of this pretraining method. We can directly apply our framework to compute the "informative" condition presented in this paper, providing a concrete starting point for analysis. If the "informative" condition is satisfied, our main results are directly applicable.

Finally, we highlight that our generic framework (including the "informative" condition) captures a wide range of setups for unsupervised pretraining. We underscore this applicability with three concrete examples, including (1) factor models with linear regression as downstream tasks; (2) Gaussian mixture models with classification as downstream tasks; and (3) Contrastive learning with linear regression as downstream tasks.

## 1.1 RELATED WORK

**Applications and methods for unsupervised pretraining.** Unsupervised pretraining has achieved tremendous success in image recognition (Caron et al., 2019), objective detection (Dai et al., 2021), natural language processing (Devlin et al., 2018; Radford et al., 2018; Song et al., 2019) and speech recognition (Schneider et al., 2019; Baevski et al., 2020). Two most widely-used pretraining approaches are (1) feature-based approaches (Brown et al., 1992; Mikolov et al., 2013; Melamud et al., 2016; Peter et al., 2018), which pretrains a model to extract representations and directly uses the pretrained representations as inputs for the downstream tasks; (2) fine-tuning based approaches, (see, e.g., Devlin et al., 2018), which fine-tunes all the model parameters in the neighborhood of pretrained representations based on downstream tasks. Erhan et al. (2010) provides the first empirical understanding on the role of pretraining. They argue that pretraining serves as a form of regularization that effectively guides the learning of downstream tasks.

A majority of settings where pretraining is used fall into the category of semi-supervised learning (see, e.g., Zhu, 2005), where a large amount of unlabeled data and a small amount of labeled data are observed during the training process. Semi-supervised learning methods aim to build a better predictor by efficiently utilizing the unlabeled data. Some traditional methods include: generative models (e.g. Ratsaby & Venkatesh, 1995), low-density separation (Joachims et al., 1999; Lawrence & Jordan, 2004; Szummer & Jaakkola, 2002), and graph-based methods (Belkin et al., 2006). While most works in this line propose new methods and show favorable empirical performance, they do not provide rigorous theoretical understanding on the benefit of unsupervised pretraining.

**Theoretical understanding of unsupervised pretraining.** Recent years witness a surge of theoretical results that provide explanations for various unsupervised pretraining methods that extract representations from unlabeled data. For example, (Saunshi et al., 2020; Wei et al., 2021) considers pretraining vector embeddings in the language models, while (Arora et al., 2019; Tosh et al., 2021b;a; HaoChen et al., 2021; Saunshi et al., 2022; Lee et al., 2021) consider several Self-Supervised Learning (SSL) approaches for pretraining. In terms of results, Wei et al. (2021) shows that linear predictor on the top of pretrained languange model can recover their ground truth model; Arora et al. (2019); Saunshi et al. (2020); Tosh et al. (2021b;a); Saunshi et al. (2022) show that the prediction loss of downstream task can be bounded by the loss of unsupervised pretraining tasks. These two lines of results do not prove the sample complexity advantage of unsupervised learning when compared to the baseline of performing supervised learning purely using the labeled data.

The most related results are Lee et al. (2021); HaoChen et al. (2021), which explicitly show the sample complexity advantage of certain unsupervised pretraining methods. However, Lee et al. (2021) focuses on reconstruction-based SSL, and critically relies on a conditional independency assumption on the feature and its reconstruction conditioned on the label; HaoChen et al. (2021) considers contrastive learning, and their results relies on deterministic feature map and the spectral conditions of the normalized adjacency matrix. Both results only apply to the specific setups and approaches of unsupervised pretraining in their papers, which do not apply to other setups in general (for instance, the three examples in Section 4, 5, 6). On the contrary, this paper develops a generic framework for unsupervised pretraining using only abstract function classes, which applies to a wide range of setups.

**Other approaches for representation learning.** There is another line of recent theoretical works that learn representation via multitask learning. Baxter (2000) provides generalization bounds for multitask transfer learning assuming a generative model and a shared representation among tasks. Maurer et al. (2016) theoretically analyses a general method for learning representations from multitasks and illustrates their method in a linear feature setting. Tripuraneni et al. (2021); Du et al. (2020) provide sample efficient algorithms that solve the problem of multitask linear regression. Tripuraneni et al. (2020) further considers generic nonlinear feature representations and shows sample complexity guarantees for diverse training tasks. Their results differ from our work because they learn representations by supervised learning using labeled data of other tasks, while our work learns representations by unsupervised learning using unlabeled data.

## 2 PROBLEM SETUP

**Notation.** We denote by $\mathbb{P}(x)$ and $p(x)$ the cumulative distribution function and the probability density function defined on $x \in \mathcal{X}$, respectively. We define $[n] = \{1, 2, \ldots, n\}$. The cardinality of set $\mathcal{A}$ is denoted by $|\mathcal{A}|$. Let $\| \cdot \|_2$ be the $\ell_2$ norm of a vector or the spectral norm of a matrix. We denote by $\| \cdot \|_F$ the Frobenius norm of a matrix. For a matrix $M \in \mathbb{R}^{m \times n}$, we denote by $\sigma_{\min}(M)$ and $\sigma_{\max}(M)$ the smallest singular value and the largest singular value of $M$, respectively. For two probability distributions $\mathbb{P}_1$ and $\mathbb{P}_2$, we denote the Total Variation (TV) distance and the Hellinger distance between these two distributions by $d_{\text{TV}}(\mathbb{P}_1, \mathbb{P}_2)$ and $H(\mathbb{P}_1, \mathbb{P}_2)$, respectively.

We denote by $x \in \mathcal{X}$ and $y \in \mathcal{Y}$ the input data and the objective of the downstream tasks, respectively. Our goal is to predict $y$ using $x$. We assume that $x$ is connected to $y$ through an unobserved latent variable $z \in \mathcal{Z}$ (which is also considered as a representation of $x$). Given the latent variable $z$, the data $x$ and the objective $y$ are independent of each other. Latent variable structure is general in statistics (for example, the hidden categories and low dimension factors) and applies to most unsupervised learning models (including contrastive learning, auto-encoder, etc). To incorporate a large class of real-world applications, such as contrastive learning, we consider the setup where learning can possibly have access to some side information $s \in \mathcal{S}$. We assume that $(x, s, z) \sim \mathbb{P}_{\phi^*}(x, s, z)$ and $y|z \sim \mathbb{P}_{\psi^*}(y|z)$, where $\mathbb{P}_{\phi^*}$ and $\mathbb{P}_{\psi^*}$ are distributions indexed by $\phi^* \in \Phi$ and $\psi^* \in \Psi$. It then holds that $\mathbb{P}_{\phi^*, \psi^*}(x, y) = \int \mathbb{P}_{\phi^*}(x, z) \mathbb{P}_{\psi^*}(y|z) \, dz$.

Let $\ell(\cdot, \cdot)$ be a loss function. For any pair $(\phi, \psi) \in \Phi \times \Psi$, the optimal predictor $g_{\phi, \psi}$ is defined as follows,

$$g_{\phi, \psi} \leftarrow \arg\min_g \mathbb{E}_{\mathbb{P}_{\phi, \psi}} \big[ \ell\big(g(x), y\big) \big], \tag{1}$$

where the minimum is taken on all the possible functions and $\mathbb{E}_{\mathbb{P}_{\phi, \psi}} := \mathbb{E}_{(x, y) \sim \mathbb{P}_{\phi, \psi}(x, y)}$. Our prediction function class is therefore given by $\mathcal{G}_{\Phi, \Psi} := \{g_{\phi, \psi} | \phi \in \Phi, \psi \in \Psi\}$.

---

**Algorithm 1** Two-Phase MLE+ERM

1: **Input:** $\{x_i, s_i\}_{i=1}^m, \{x_j, y_j\}_{j=1}^n$
2: Use unlabeled data and its corresponding side information $\{x_i, s_i\}_{i=1}^m$ to learn $\hat{\phi}$ via MLE:

$$\hat{\phi} \leftarrow \arg\max_{\phi \in \Phi} \sum_{i=1}^m \log p_\phi(x_i, s_i). \tag{3}$$

3: Fix $\hat{\phi}$ and use labeled data $\{x_j, y_j\}_{j=1}^n$ to learn $\hat{\psi}$ via ERM:

$$\hat{\psi} \leftarrow \arg\min_{\psi \in \Psi} \sum_{j=1}^n \ell\big(g_{\hat{\phi}, \psi}(x_j), y_j\big). \tag{4}$$

---

Our framework covers the standard setup (e.g., in large language models) which uses a large mount unlabeled data to pretrain a deep neural network as a representation, and then uses a small amount of labeled data to only fine-tunes the linear head for downstream tasks. Concretely, consider the setting where $\ell(\cdot, \cdot)$ is the squared loss and $y = \beta^{*T} z + \varepsilon$, where $\varepsilon \sim \mathcal{N}(0, \sigma^2)$ is a Gaussian noise independent of $z$. Then the optimal predictor $g_{\phi, \psi}(x) = \mathbb{E}_{\mathbb{P}_{\phi, \beta}}[y|x] = \beta^T \mathbb{E}_{\mathbb{P}_\phi}[z|x]$ and the prediction function class $\mathcal{G}_{\Phi, \Psi} = \{\beta^T \mathbb{E}_{\mathbb{P}_\phi}[z|x] \,|\, \phi \in \Phi, \beta \in \Psi\}$. Here, $\mathbb{E}_{\mathbb{P}_\phi}[z|x]$ corresponds to the representation learned by deep networks, and $\beta$ is the parameter of the linear head.

Given an estimator pair $(\hat{\phi}, \hat{\psi})$, we define the excess risk with respect to loss $\ell(\cdot, \cdot)$ as

$$\text{Error}_\ell(\hat{\phi}, \hat{\psi}) := \mathbb{E}_{\mathbb{P}_{\phi^*, \psi^*}}\big[\ell\big(g_{\hat{\phi}, \hat{\psi}}(x), y\big)\big] - \mathbb{E}_{\mathbb{P}_{\phi^*, \psi^*}}\big[\ell\big(g_{\phi^*, \psi^*}(x), y\big)\big], \tag{2}$$

where $\phi^*$ and $\psi^*$ are the ground truth parameters. By the definition of $g_{\phi^*, \psi^*}$, we have $\text{Error}(\hat{\phi}, \hat{\psi}) \geq 0$. We aim to learn an estimator pair $(\hat{\phi}, \hat{\psi})$ from data that achieves smallest order of the excess risk.

We consider the setting where the latent variable $z$ cannot be observed. Specifically, we are given many unlabeled data and its corresponding side information $\{x_i, s_i\}_{i=1}^m$ that are sampled i.i.d from an unknown distribution $\mathbb{P}_{\phi^*}(x, s)$ and only a few labeled data $\{x_j, y_j\}_{j=1}^n$ that are sampled i.i.d (also independent with the unlabeled data) from an unknown distribution $\mathbb{P}_{\phi^*, \psi^*}(x, y)$.

**Learning algorithm.** We consider a natural learning algorithm consisting of two phases (Algorithm 1). In the unsupervised pretraining phase, we use MLE to estimate $\phi^*$ based on the unlabeled data $\{x_i, s_i\}_{i=1}^m$. In the downstream tasks learning phase, we use ERM to estimate $\psi^*$ based on pretrained $\hat{\phi}$ and the labeled data $\{x_j, y_j\}_{j=1}^n$. See algorithm 1 for details.

We remark that another natural learning algorithm in our setting is to use a two-phase MLE. To be specific, in the unsupervised pretraining phase, we use MLE to estimate $\phi^*$ based on the unlabeled data $\{x_i, s_i\}_{i=1}^m$ as (3). In the downstream tasks learning phase, we again use MLE to estimate $\psi^*$ based on pretrained $\hat{\phi}$ and the labeled data $\{x_j, y_j\}_{j=1}^n$. However, we can show that this two-phase MLE scheme fails in the worst case. See Appendix E for the details.

**Complexity measures.** Sample complexity guarantee for Algorithm 1 will be phrased in terms of three complexity measurements, i.e., bracketing number, covering number and the Rademacher complexity, which are defined as follows. We denote by $\mathcal{P}_\mathcal{X}(\Phi) := \{p_\phi(x) \,|\, \phi \in \Phi\}$ a set of parameterized density functions $p_\phi(x)$ defined on $x \in \mathcal{X}$, where $\phi \in \Phi$ is the parameter.

**Definition 2.1** ($\epsilon$-Bracket and Bracketing Number). Let $\epsilon > 0$. Under $\|\cdot\|_1$ distance, a set of functions $\mathcal{N}_{[\,]}(\mathcal{P}_\mathcal{X}(\Phi), \epsilon)$ is an $\epsilon$-bracket of $\mathcal{P}_\mathcal{X}(\Phi)$ if for any $p_\phi(x) \in \mathcal{P}_\mathcal{X}(\Phi)$, there exists a function $\bar{p}_\phi(x) \in \mathcal{N}_{[\,]}(\mathcal{P}_\mathcal{X}(\Phi), \epsilon)$ such that: (1) $\bar{p}_\phi(x) \geq p_\phi(x), \forall x \in \mathcal{X}$; (2) $\|\bar{p}_\phi(x) - p_\phi(x)\|_1 = \int |\bar{p}_\phi(x) - p_\phi(x)| \, dx \leq \epsilon$. The bracketing number $N_{[\,]}(\mathcal{P}_\mathcal{X}(\Phi), \epsilon)$ is the cardinality of the smallest $\epsilon$-bracket needed to cover $\mathcal{P}_\mathcal{X}(\Phi)$. The entropy is defined as the logarithm of the bracketing number.

To measure the complexity of a function class, we consider the covering number and the Rademacher complexity defined as follows.

**Definition 2.2** ($\epsilon$-Cover and Covering Number). Let $\mathcal{F}$ be a function class and $(\mathcal{F}, \|\cdot\|)$ be a metric space. For each $\epsilon > 0$, a set of functions $\mathcal{N}(\mathcal{F}, \epsilon, \|\cdot\|)$ is called an $\epsilon$-cover of $\mathcal{F}$ if for any $f \in \mathcal{F}$,

there exists a function $g \in \mathcal{N}(\mathcal{F}, \epsilon, \|\cdot\|)$ such that $\|f - g\| \leq \epsilon$. The covering number $N(\mathcal{F}, \epsilon, \|\cdot\|)$ is defined as the cardinality of the smallest $\epsilon$-cover needed to cover $\mathcal{F}$.

**Definition 2.3** (Rademacher Complexity). Suppose that $x_1, \ldots, x_n$ are sampled i.i.d from a probability distribution $\mathcal{D}$ defined on a set $\mathcal{X}$. Let $\mathcal{G}$ be a class of functions mapping from $\mathcal{X}$ to $\mathbb{R}$. The empirical Rademacher complexity of $\mathcal{G}$ is defined as follows,

$$\hat{R}_n(\mathcal{G}) := \mathbb{E}_{\{\sigma_i\}_{i=1}^n \sim \text{Unif}\{\pm 1\}} \left[ \sup_{g \in \mathcal{G}} \frac{2}{n} \sum_{i=1}^n \sigma_i g(x_i) \right].$$

The Rademacher complexity of $\mathcal{G}$ is defined as $R_n(\mathcal{G}) := \mathbb{E}_{\{x_i\}_{i=1}^n \sim \mathcal{D}}[\hat{R}_n(\mathcal{G})]$.

## 3 MAIN RESULTS

In this section, we first introduce a mild "informative" condition for unsupervised pretraining. We show this "informative" condition is necessary for pretraining to benefit downstream tasks. We then provide our main results—statistical guarantees for unsupervised pretraining and downstream tasks for Algorithm 1. Finally, in Section 3.1, we generalize our results to a more technical but weaker version of the "informative" condition, which turns out to be useful in capturing our third example of contrastive learning (Section 6).

**Informative pretraining tasks.** We first note that under our generic setup, unsupervised pretraining may not benefit downstream tasks at all in the worst case if no further conditions are assumed.

**Proposition 3.1.** *There exist classes $(\Phi, \Psi)$ as in Section 2 such that, regardless of unsupervised pretraining algorithms used, pretraining using unlabeled data provides no additional information towards learning predictor $g_{\phi^*, \psi^*}$.*

Consider the latent variable model $z = Ax$, where $x \sim \mathcal{N}(0, I_d)$, $A \in \Phi$ is the parameter of the model. Then, no matter how many unlabeled $\{x_i\}$ we have, we can gain no information of $A$ from the data! In this case, unsupervised pretraining is not beneficial for any downstream task.

Therefore, it's crucial to give an assumption that guarantees our unsupervised pretraining is informative. As a thought experiment, suppose that in the pretraining step, we find an exact density estimator $\hat{\phi}$ for the marginal distribution of $x, s$, i.e., $p_{\hat{\phi}}(x, s) = p_{\phi^*}(x, s)$ holds for every $x, s$. We should expect that this estimator also fully reveals the relationship between $x$ and $z$, i.e., $p_{\hat{\phi}}(x, z) = p_{\phi^*}(x, z)$ holds for every $x, z$. Unfortunately, this condition does not hold in most practical setups and is often too strong. As an example, consider Gaussian mixture models, where $z \in [K]$ is the cluster that data point $x \in \mathbb{R}^d$ belongs to. Then in this case, it is impossible for us to ensure $p_{\hat{\phi}}(x, z) = p_{\phi^*}(x, z)$, since a permutation of $z$ makes no difference in the marginal distribution of $x$. However, notice that in many circumstances, a permutation of the class label will not affect the downstream task learning. In these cases, a permutation of the clusters is allowed. Motivated by this observation, we introduce the following informative assumption which allows certain "transformation" induced by the downstream task:

**Assumption 3.2** ($\kappa^{-1}$-informative condition). We assume that the model class $\Phi$ is $\kappa^{-1}$-informative with respect to a transformation group $\mathcal{T}_\Phi$. That is, for any $\phi \in \Phi$, there exists $T_1 \in \mathcal{T}_\Phi$ such that

$$d_{\text{TV}}\big(\mathbb{P}_{T_1 \circ \phi}(x, z), \mathbb{P}_{\phi^*}(x, z)\big) \leq \kappa \cdot d_{\text{TV}}\big(\mathbb{P}_\phi(x, s), \mathbb{P}_{\phi^*}(x, s)\big). \tag{5}$$

Here $\phi^*$ is the ground truth parameter. Furthermore, we assume that $\mathcal{T}_\Phi$ is induced by transformation group $\mathcal{T}_\Psi$ on $\Psi$, i.e., for any $T_1 \in \mathcal{T}_\Phi$, there exists $T_2 \in \mathcal{T}_\Psi$ such that for any $(\phi, \psi) \in \Phi \times \Psi$,

$$\mathbb{P}_{\phi, \psi}(x, y) = \mathbb{P}_{T_1 \circ \phi, T_2 \circ \psi}(x, y). \tag{6}$$

Under Assumption 3.2, if the pretrained $\hat{\phi}$ accurately estimates the marginal distribution of $x, s$ up to high accuracy, then it also reveals the correct relation between $x$ and representation $z$ up to some transformation $\mathcal{T}_\Phi$ which is allowed by the downstream task, which makes it possible to learn the downstream task using less labeled data.

Proposition 3.1 shows that the informative condition is necessary for pretraining to bring advantage since the counter example in the proposition is precisely 0-informative. We will also show this informative condition is rich enough to capture a wide range of unsupervised pretraining methods in Section 4, 5, 6, including factor models, Gaussian mixture models, and contrastive learning models.

**Guarantees for unsupervised pretraining.** Recall that $\mathcal{P}_{\mathcal{X} \times \mathcal{S}}(\Phi) := \{p_\phi(x, s) \mid \phi \in \Phi\}$. We have the following guarantee for the MLE step (line 2) of Algorithm 1.

**Theorem 3.3.** *Let $\hat{\phi}$ be the maximizer defined in* (3). *Then, with probability at least $1 - \delta$, we have*

$$d_{\mathrm{TV}}\big(\mathbb{P}_{\hat{\phi}}(x, s), \mathbb{P}_{\phi^*}(x, s)\big) \leq 3\sqrt{\frac{1}{m} \log \frac{N_{[\,]}(\mathcal{P}_{\mathcal{X} \times \mathcal{S}}(\Phi), \frac{1}{m})}{\delta}},$$

*where $N_{[\,]}$ is the bracketing number as in Definition 2.1.*

Theorem 3.3 claims that the TV error in estimating the joint distribution of $(x, s)$ decreases as $\mathcal{O}(\mathcal{C}_\Phi / m)$ where $m$ is the number of unlabeled data, and $\mathcal{C}_\Phi = \log N_{[\,]}(\mathcal{P}_{\mathcal{X} \times \mathcal{S}}(\Phi), 1/m)$ measures the complexity of learning the latent variable models $\Phi$. This result mostly follows from standard analysis of MLE (Van de Geer, 2000). We include the proof in Appendix A.1 for completeness.

**Guarantees for downstream task learning.** In practice, we can only learn an approximate downstream predictor using a small amount of labeled data. We upper bound the excess risk of Algorithm 1 as follows.

**Theorem 3.4.** *Let $\hat{\phi}$ and $\hat{\psi}$ be the outputs of Algorithm 1. Suppose that the loss function $\ell : \mathcal{Y} \times \mathcal{Y} \to \mathbb{R}$ is L-bounded and our model is $\kappa^{-1}$-informative. Then, with probability at least $1 - \delta$, the excess risk of Algorithm 1 is bounded as:*

$$\mathrm{Error}_\ell(\hat{\phi}, \hat{\psi}) \leq 2 \max_{\phi \in \Phi} R_n(\ell \circ \mathcal{G}_{\phi, \Psi}) + 12\kappa L \cdot \sqrt{\frac{1}{m} \log \frac{2N_{[\,]}(\mathcal{P}_{\mathcal{X} \times \mathcal{S}}(\Phi), 1/m)}{\delta}} + 2L \cdot \sqrt{\frac{2}{n} \log \frac{4}{\delta}}.$$

*Here $R_n(\cdot)$ denotes the Rademacher complexity, and $\ell \circ \mathcal{G}_{\phi, \Psi} := \big\{\ell\big(g_{\phi, \psi}(x), y\big) : \mathcal{X} \times \mathcal{Y} \to [-L, L] \mid \psi \in \Psi\big\}$.*

Note that the Rademacher complexity of a function class can be bounded by its metric entropy. We then have the following corollary.

**Corollary 3.5.** *Under the same preconditions as Theorem 3.4, we have:*

$$\mathrm{Error}_\ell(\hat{\phi}, \hat{\psi}) \leq \tilde{c} \max_{\phi \in \Phi} L \sqrt{\frac{\log N(\ell \circ \mathcal{G}_{\phi, \Psi}, L/\sqrt{n}, \|\cdot\|_\infty)}{n}} + 2L\sqrt{\frac{2}{n} \log \frac{4}{\delta}}$$

$$+ 12\kappa L \sqrt{\frac{1}{m} \log \frac{2N_{[\,]}(\mathcal{P}_{\mathcal{X} \times \mathcal{S}}(\Phi), 1/m)}{\delta}},$$

*where $\tilde{c}$ is an absolute constant, $N(\mathcal{F}, \delta, \|\cdot\|_\infty)$ is the $\delta-$covering number of function class $\mathcal{F}$ with respect to the metric $\|\cdot\|_\infty$.*

By Corollary 3.5, the excess risk of our Algorithm 1 is approximately $\tilde{\mathcal{O}}(\sqrt{\mathcal{C}_\Phi/m} + \sqrt{\mathcal{C}_\Psi/n})$, where $\mathcal{C}_\Phi$ and $\mathcal{C}_\Psi$ are roughly the log bracketing number of class $\Phi$ and the log covering number of $\Psi$. Note that excess risk for the baseline algorithm that learns downstream task using only labeled data is $\tilde{\mathcal{O}}(\sqrt{\mathcal{C}_{\Phi \circ \Psi}/n})$, where $\mathcal{C}_{\Phi \circ \Psi}$ is the log covering number of composite function class $\Phi \circ \Psi$. In many practical scenarios such as training a linear predictor on top of a pretrained deep neural networks, the complexity $\mathcal{C}_{\Phi \circ \Psi}$ is much larger than $\mathcal{C}_\Psi$. We also often have significantly more unlabeled data than labeled data ($m \gg n$). In these scenarios, our result rigorously shows the significant advantage of unsupervised pretraining compared to the baseline algorithm which directly performs supervised learning without using unlabeled data.

### 3.1 GUARANTEES FOR WEAKLY INFORMATIVE MODELS

We introduce a relaxed version of Assumption 3.2, which allows us to capture a richer class of examples.

**Assumption 3.6** ($\kappa^{-1}$-weakly-informative condition). We assume model $(\Phi, \Psi)$ is $\kappa^{-1}$-weakly-informative, that is, for any $\phi \in \Phi$, there exists $\psi \in \Psi$ such that

$$d_{\mathrm{TV}}\big(\mathbb{P}_{\phi, \psi}(x, y), \mathbb{P}_{\phi^*, \psi^*}(x, y)\big) \leq \kappa \cdot H\big(\mathbb{P}_\phi(x, s), \mathbb{P}_{\phi^*}(x, s)\big). \tag{7}$$

Here we denote by $\phi^*, \psi^*$ the ground truth parameters.

Assumption 3.6 relaxes Assumption 3.2 by making two modifications: (i) replace the LHS of (5) by the TV distance between the joint distribution of $(x, y)$; (ii) replace the TV distance on the RHS by the Hellinger distance. See more on the relation of two assumptions in Appendix A.4.1.

In fact, Assumption 3.6 is sufficient for us to achieve the same theoretical guarantee as that in Theorem 3.4.

**Theorem 3.7.** *Theorem 3.4 still holds under the $\kappa^{-1}$-weakly-informative assumptions.*

The proof of Theorem 3.7 requires a stronger version of MLE guarantee than Theorem 3.3, which guarantees the closeness in terms of Hellinger distance. We leave the details in Appendix A.4.

## 4 PRETRAINING VIA FACTOR MODELS

In this section, we instantiate our theoretical framework using the factor model with linear regression as a downstream task. Factor model (see, e.g., Lawley & Maxwell, 1971; Forni et al., 2005; Fan et al., 2021) is widely used in finance, computational biology, and sociology, where the high-dimensional measurements are strongly correlated. We rigorously show how unsupervised pretraining can help reduce sample complexity in this case.

**Model Setup.** For the latent variable model, we consider the factor model as follows.

**Definition 4.1** (Factor Model). Suppose that we have $d$-dimensional random vector $x$, whose dependence is driven by $r$ factors $z$ ($d \gg r$). The factor model assumes $x = B^* z + \mu$, where $B^*$ is a $d \times r$ factor loading matrix. Here $\mu \sim N(0, I_d)$ is the idiosyncratic component that is uncorrelated with the common factor $z \sim N(0, I_r)$. We assume that the ground truth parameters $B^* \in \mathcal{B}$, where $\mathcal{B} := \{B \in \mathbb{R}^{d \times r} \mid \|B\|_2 \le D\}$ for some $D > 0$.

For the downstream task, we consider the linear regression problem $y = \beta^{*T} z + \nu$, where $\nu \sim N(0, \varepsilon^2)$ is a Gaussian noise that is uncorrelated with the factor $z$ and the idiosyncratic component $\mu$. We assume that the ground truth parameters $\beta^* \in \mathcal{C}$, where $\mathcal{C} := \{\beta \in \mathbb{R}^r \mid \|\beta\|_2 \le D\}$ for some $D > 0$. The latent variable model (i.e., $\Phi$) and the the prediction class (i.e.,$\Psi$) are then represented by $\mathcal{B}$ and $\mathcal{C}$, respectively. In the sequel, we consider the case where no side information is available, i.e., we only have access to i.i.d unlabeled data $\{x_i\}_{i=1}^m$ and i.i.d labeled data $\{x_j, y_j\}_{j=1}^n$. For regression models, it is natural to consider the squared loss function $\ell(x, y) := (y - x)^2$.

**Informative condition.** We first show that Assumption 3.2 holds for the factor model with linear regression as downstream tasks. The idea of the factor model is to learn a low-dimensional representation $z$, where a rotation over $z$ is allowed since in the downstream task, we can also rotate $\beta$ to adapt to the rotated $z$.

**Lemma 4.2.** *Factor model with linear regression as downstream tasks is $\kappa^{-1}$-informative, where $\kappa = c_1(\sigma_{\max}^* + 1)^4 (\sigma_{\min}^*)^{-3}$. Here $c_1$ is some absolute constants, $\sigma_{\max}^*$ and $\sigma_{\min}^*$ are the largest and smallest singular value of $B^*$, respectively.*

**Theoretical results.** Recall that in Theorem 3.4, we assume a $L$-bounded loss function to guarantee the performance of Algorithm 1. Thus, instead of directly applying Algorithm 1 to the squared loss function, we consider Algorithm 1 with truncated squared loss, i.e.,

$$\tilde{\ell}(x, y) := (y - x)^2 \cdot \mathbb{1}_{\{(y-x)^2 \le L\}} + L \cdot \mathbb{1}_{\{(y-x)^2 > L\}}. \tag{8}$$

Here $L$ is a carefully chosen truncation level. To be more specific, in the first phase, we still use MLE to learn an estimator $\hat{B}$ as that in line 2 of Algorithm 1. In the second phase, we apply ERM to the truncated squared loss to learn an estimator $\hat{\beta}$. We then have the following theoretical guarantee.

**Theorem 4.3.** *We consider Algorithm 1 with truncated squared loss (8) with $L = (D^2 + 1)^3 \log n$. Let $\hat{B}, \hat{\beta}$ be the outputs of Algorithm 1. Then, for factor models with linear regression as downstream tasks, with probability at least $1 - \delta$, the excess risk can be bounded as follows,*

$$\mathrm{Error}_\ell(\hat{B}, \hat{\beta}) \le \tilde{\mathcal{O}} \left( \kappa L \sqrt{dr/m} + L \sqrt{r/n} \right),$$

*where $D$ is defined in the sets $\mathcal{B}$ and $\mathcal{C}$, and $\kappa$ is specified in Lemma 4.2. Here $\tilde{\mathcal{O}}(\cdot)$ omits absolute constants and the polylogarithmic factors in $m, d, r, D, 1/\delta$.*

Notice that the rate we obtain in Theorem 4.3 is not optimal for this specific task: by the nature of squared loss, if we consider a direct $d-$dimensional linear regression (from $x$ to $y$) with $n$ data, we can usually achieve the fast rate, where excess risk decreases as $\tilde{\mathcal{O}}(d/n)$. To fill this gap, we consider Algorithm 1 with $\Phi = \mathbb{R}^{d \times r}$ and $\Psi = \mathbb{R}^r$ and denote $D := \max\{\|B^*\|_2, \|\beta^*\|_2\}$. Following a more refined analysis, we could achieve a sharper risk rate that scales as $\tilde{\mathcal{O}}(d/m + r/n)$, which is much better than the usual linear regression when $m \gg n$. See Appendix B.5 for details.

## 5 PRETRAINING VIA GAUSSIAN MIXTURE MODELS

In this section, we show how pretraining using Gaussian Mixture Models (GMMs) can benefit the downstream classification tasks, under our theoretical framework.

**Model setup.** For the latent variable model, we consider a $d$-dimensional GMM with $K$ components and equal weights. To be specific, the latent variable $z$ that represents the cluster is sampled uniformly from $[K]$. In each cluster, the data is sampled from a standard Gaussian distribution, i.e., $x|z = i \sim \mathcal{N}(u_i^*, I_d)$ for any $i \in [K]$. It then holds that $x \sim \sum_{i=1}^{K} K^{-1} \mathcal{N}(u_i^*, I_d)$. We denote by $\mathcal{U}$ the parameter space with each element consisting of $K$ centers ($d$-dimensional vectors).

We assume that the set of parameters $\mathcal{U}$ satisfies the normalization condition—there exists $D > 0$ such that for any $\mathbf{u} = \{u_i\}_{i=1}^{K} \in \mathcal{U}$, we have $\|u_i\|_2 \leq D\sqrt{d \log K}$, $\forall i \in [K]$. We further assume the ground-truth centers $\{u_i^*\}_{i=1}^{K} \in \mathcal{U}$ satisfy the following separation condition.

**Assumption 5.1** (Separation condition). *The true parameters $\{u_i^*\}_{i=1}^{K} \in \mathcal{U}$ satisfies $\|u_i^* - u_j^*\|_2 \geq 100\sqrt{d \log K}$, $\forall i \neq j$.*

For the downstream task, we consider the binary classification problems with label $y \in \{0, 1\}$. We denote by $\Psi$ the set of $2^K$ classifiers such that for each $\psi \in \Psi$, and any $i \in [K]$, we have either $\mathbb{P}_\psi(y = 1|z = i) = 1 - \varepsilon$ or $\mathbb{P}_\psi(y = 0|z = i) = 1 - \varepsilon$, where $\varepsilon$ represents the noise. Then, the latent variable model and the prediction class are represented by $\mathcal{U}$ and $\Psi$, respectively. In the sequel, we consider the case where no side information is available, i.e., we only have access to i.i.d unlabeled data $\{x_i\}_{i=1}^{m}$ and i.i.d labeled data $\{x_j, y_j\}_{j=1}^{n}$. For classification problems, it is natural to consider the $0 - 1$ loss function $\ell(x, y) := \mathbb{1}_{\{x \neq y\}}$ which is bounded by 1.

**Informative condition.** We prove that Assumption 3.2 for the above model. We have the following guarantee.

**Lemma 5.2.** *Let $\tilde{\mathcal{U}} = \{\mathbf{u} \in \mathcal{U} \mid d_{\mathrm{TV}}(p_\mathbf{u}(x), p_{\mathbf{u}^*}(x)) \leq 1/(4K)\}$. Under Assumption 5.1, GMMs with parameters in $\tilde{\mathcal{U}}$ is $\mathcal{O}(1)$-informative with respect to the transformation group induced by downstream classification tasks.*

**Theoretical results** We have the following theoretical guarantee.

**Theorem 5.3.** *Let $\hat{\mathbf{u}}, \hat{\psi}$ be the outputs of Algorithm 1. Suppose that Assumption 5.1 holds and $m = \tilde{\Omega}(dK^3)$. Then, for the Gaussian mixture model with classification as downstream tasks, with probability at least $1 - \delta$, the excess risk can be bounded as follows,*

$$\mathrm{Error}_\ell(\hat{\mathbf{u}}, \hat{\psi}) \leq \tilde{\mathcal{O}}\left(\sqrt{dK/m} + \sqrt{K/n}\right),$$

*Here $\tilde{\mathcal{O}}(\cdot)$ omits some constants and the polylogarithmic factors in $m, d, K, D, 1/\delta$.*

Theorem 5.3 shows the power of unsupervised pretraining under this setting in the following sense: Note that the number of parameters of a GMM is $dK$, therefore if we directly do classification without unsupervised pretraining, the risk will scale as $\tilde{\mathcal{O}}(\sqrt{dK/n})$. When $d$ is large and $m \gg n$, we achieve a better risk bound than supervised learning that only uses the labeled data.

## 6 PRETRAINING VIA CONTRASTIVE LEARNING

In this section, we show how pretraining through contrastive learning (learning the embedding function) can benefit the downstream linear regression tasks under our theoretical framework.

**Model setup.** In the setting of contrastive learning, we assume that $x$ and $x'$ are sampled independently from the same distribution $\mathbb{P}(x)$. The similarity between $x$ and $x'$ is captured by a representation function $f_{\theta^*} : \mathcal{X} \to \mathbb{R}^r$ in the following sense,

$$\mathbb{P}(t = 1 \,|\, x, x') = (1 + e^{-f_{\theta^*}(x)^T f_{\theta^*}(x')})^{-1}, \quad \mathbb{P}(t = -1 \,|\, x, x') = (1 + e^{f_{\theta^*}(x)^T f_{\theta^*}(x')})^{-1}.$$

Here $t$ is a random variable that labels the similarity between $x$ and $x'$. If the data pair $(x, x')$ is similar, then $t$ tends to be 1. If the data pair $(x, x')$ is not similar (negative samples), then $t$ tends to be $-1$. We assume $(x, x', t) \sim \mathbb{P}_{f_{\theta^*}}(x, x', t)$. Here, $(x', t)$ can be viewed as side information. The latent variable $z$ is defined as $z := f_{\theta^*}(x) + \mu$, where $\mu \sim \mathcal{N}(0, I_r)$ is a Gaussian noise that is uncorrelated with $x$. We denote $(x, z) \sim \mathbb{P}_{f_{\theta^*}}(x, z)$.

For the downstream task, we consider the linear regression problem $y = \beta^{*T} z + \nu$, where $\nu \sim \mathcal{N}(0, 1)$ is a Gaussian noise. We assume that the true parameters $\theta^* \in \Theta$ and $\beta^* \in \mathcal{B}$, which satisfy a standard normalization assumption, i.e., $\|f_\theta(x)\|_2 \le 1$ for any $\theta \in \Theta$ and $x \in \mathcal{X}$ and $\|\beta\|_2 \le D$ for any $\beta \in \mathcal{B}$. We have access to i.i.d unlabeled data $\{x_i, x_i', t_i\}_{i=1}^m$ and i.i.d labeled data $\{x_j, y_j\}_{j=1}^n$. Here $(x_i', t_i)$ is the side information corresponding to $x_i$. In the sequel, we consider the same form of truncated squared loss as in (8).

**Weakly informative condition.** We first prove that the above model satisfies Assumption 3.6:

**Lemma 6.1.** *Contrastive learning with linear regression as downstream tasks is $\kappa^{-1}$-weakly-informative, where $\kappa = c_3 \cdot \sigma_{\min}^{-1/2}(\mathbb{E}[f_{\theta^*}(x) f_{\theta^*}(x)^T])$. Here $c_3$ is an absolute constant.*

**Theoretical results.** We define a set of density functions $\mathcal{P}_{\mathcal{X} \times \mathcal{S}}(\mathcal{F}_\theta) := \{p_{f_\theta}(x, x', t) \,|\, \theta \in \Theta\}$. We then have the following theoretical guarantee.

**Theorem 6.2.** *We consider Algorithm 1 with truncated squared loss (8) where $L = 36(D^2 + 1) \log n$. Let $\hat{\theta}, \hat{\beta}$ be the outputs of Algorithm 1. Then, for contrastive learning with linear regression as downstream tasks, with probability at least $1 - \delta$, the excess risk can be bounded as follows,*

$$\mathrm{Error}_\ell(\hat{\theta}, \hat{\beta}) \le \tilde{\mathcal{O}}\left(\kappa L \sqrt{\frac{\log N_{[\,]}(\mathcal{P}_{\mathcal{X} \times \mathcal{S}}(\mathcal{F}_\theta), 1/m^2)}{m}} + L\sqrt{\frac{1}{n}}\right),$$

*where $L = 36(D^2 + 1) \log n$ and $\kappa$ is specified in Lemma 6.1. Here $\tilde{\mathcal{O}}(\cdot)$ omits some constants and the polylogarithmic factors in $1/\delta$.*

Note that the excess risk of directly training with labeled data strongly depends on the complexity of the function class $\mathcal{F}_\theta$. In the case that $m \gg n$, the excess risk of Theorem 6.2 scales as $\tilde{O}(\sqrt{1/n})$, which beats the pure supervised learning if the complexity of $\mathcal{F}_\theta$ is quite large. Thus, the utility of unsupervised pretraining is revealed for contrastive learning.

When applying our generic framework to the specific context of contrastive learning, our result morally aligns with that in HaoChen et al. (2021), albeit with differing assumptions. In Theorem 4.3 of HaoChen et al. (2021), the risk is characterized by the eigenvalues of an adjacency matrix $\bar{A}$ whose elements measure the similarity between data pairs. As a counterpart, our excess risk incorporates the quantity $\kappa = \sigma_{\min}^{-1/2}(\mathbb{E}[f_{\theta^*}(x) f_{\theta^*}(x)^T])$, where $\mathbb{E}[f_{\theta^*}(x) f_{\theta^*}(x)^T]$ also plays the role of measuring the similarity. Notably, our generic framework covers a variety of approaches for unsupervised pretraining, extending beyond just contrastive learning.

# 7 CONCLUSIONS

This paper proposes a generic theoretic framework for explaining the statistical benefits of unsupervised pretraining. We study the natural scheme of using MLE for unsupervised pretraining and ERM for downstream task learning. We identify a natural "informative" condition, under which our algorithm achieves an excess risk bound that significantly improves over the baseline achieved by purely supervised learning in the typical practical regimes. We further instantiate our theoretical framework with three concrete approaches for unsupervised pretraining and provide corresponding guarantees.

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
