## A  PROOFS FOR SECTION 3

In Section A.1, we prove Theorem 3.3, which gives a TV distance guarantee for the MLE step in Algorithm 1. Our proof is inspired by Van de Geer (2000); Zhang (2006), and largely follows Agarwal et al. (2020); Liu et al. (2022). In Section A.2, we prove Theorem 3.4 that guarantees the performance of Algorithm 1 by upper bounding the excess risk. The proof relies on the fact that the labeled data $\{x_j, y_j\}_{j=1}^n$ are independent of the unlabeled data $\{x_i, s_i\}_{i=1}^m$. In Section A.3, we prove Corollary 3.5 based on the analysis of Gaussian complexity. In Section A.4, we prove Theorem 3.7 by first showing that the MLE step in Algorithm 1 actually guarantees an upper bound on the Hellinger distance, which is stronger than the TV distance guarantee mentioned in Theorem 3.3.

### A.1  PROOFS FOR THEOREM 3.3

In the sequel, we prove Theorem 3.3.

*Proof of Theorem 3.3.* For notation simplicity, we denote $\boldsymbol{x} := (x, s)$. Recall that we define $\mathcal{P}_{\mathcal{X} \times \mathcal{S}}(\Phi) := \{p_\phi(x, s) \mid \phi \in \Phi\}$. Let $\mathcal{N}_{[\,]}(\mathcal{P}_{\mathcal{X} \times \mathcal{S}}(\Phi), \epsilon)$ be the smallest $\epsilon$-bracket of $\mathcal{P}_{\mathcal{X} \times \mathcal{S}}(\Phi)$. We have $|\mathcal{N}_{[\,]}(\mathcal{P}_{\mathcal{X} \times \mathcal{S}}(\Phi), \epsilon)| = N_{[\,]}(\mathcal{P}_{\mathcal{X} \times \mathcal{S}}(\Phi), \epsilon)$, where $N_{[\,]}(\mathcal{P}_{\mathcal{X} \times \mathcal{S}}(\Phi), \epsilon)$ is the bracketing number of $\mathcal{P}_{\mathcal{X} \times \mathcal{S}}(\Phi)$. By Markov inequality and Boole's inequality, it holds with probability at least $1 - \delta$ that for all $\bar{p}_\phi(\boldsymbol{x}) \in \mathcal{N}_{[\,]}(\mathcal{P}_{\mathcal{X} \times \mathcal{S}}(\Phi), \epsilon)$

$$\frac{1}{2} \sum_{i=1}^m \log \frac{\bar{p}_\phi(\boldsymbol{x}_i)}{p_{\phi^*}(\boldsymbol{x}_i)} \le \log \mathbb{E}\left[e^{\frac{1}{2} \sum_{i=1}^m \log \frac{\bar{p}_\phi(\boldsymbol{x}_i)}{p_{\phi^*}(\boldsymbol{x}_i)}}\right] + \log \frac{N_{[\,]}(\mathcal{P}_{\mathcal{X} \times \mathcal{S}}(\Phi), \epsilon)}{\delta}. \tag{9}$$

Note that $\hat{\phi}$ is the maximizer of the likelihood function, i.e.

$$\hat{\phi} \leftarrow \arg\max_{\phi \in \Phi} \sum_{i=1}^m \log p_\phi(\boldsymbol{x}_i),$$

which implies

$$\frac{1}{2} \sum_{i=1}^m \log \frac{\bar{p}_{\hat{\phi}}(\boldsymbol{x}_i)}{p_{\phi^*}(\boldsymbol{x}_i)} \ge 0. \tag{10}$$

Then we have with probability at least $1 - \delta$ that

$$
\begin{aligned}
0 &\le \log \mathbb{E}\left[e^{\frac{1}{2} \sum_{i=1}^m \log \frac{\bar{p}_{\hat{\phi}}(\boldsymbol{x}_i)}{p_{\phi^*}(\boldsymbol{x}_i)}}\right] + \log \frac{N_{[\,]}(\mathcal{P}_{\mathcal{X} \times \mathcal{S}}(\Phi), \epsilon)}{\delta}, \\
&= m \log \mathbb{E}\left[\sqrt{\frac{\bar{p}_{\hat{\phi}}(\boldsymbol{x})}{p_{\phi^*}(\boldsymbol{x})}}\right] + \log \frac{N_{[\,]}(\mathcal{P}_{\mathcal{X} \times \mathcal{S}}(\Phi), \epsilon)}{\delta}, \\
&= m \log \int \sqrt{\bar{p}_{\hat{\phi}}(\boldsymbol{x}) p_{\phi^*}(\boldsymbol{x})}\, d\boldsymbol{x} + \log \frac{N_{[\,]}(\mathcal{P}_{\mathcal{X} \times \mathcal{S}}(\Phi), \epsilon)}{\delta}, \\
&\le m\left(\int \sqrt{\bar{p}_{\hat{\phi}}(\boldsymbol{x}) p_{\phi^*}(\boldsymbol{x})}\, d\boldsymbol{x} - 1\right) + \log \frac{N_{[\,]}(\mathcal{P}_{\mathcal{X} \times \mathcal{S}}(\Phi), \epsilon)}{\delta}, \tag{11}
\end{aligned}
$$

where the last inequality follows from the fact that $\log x \le x - 1$. By rearranging the terms, we have

$$1 - \int \sqrt{\bar{p}_{\hat{\phi}}(\boldsymbol{x}) p_{\phi^*}(\boldsymbol{x})}\, d\boldsymbol{x} \le \frac{1}{m} \log \frac{N_{[\,]}(\mathcal{P}_{\mathcal{X} \times \mathcal{S}}(\Phi), \epsilon)}{\delta}. \tag{12}$$

By the definition of bracket, we obtain

$$\int \bar{p}_{\hat{\phi}}(\boldsymbol{x}) d\boldsymbol{x} = \int (\bar{p}_{\hat{\phi}}(\boldsymbol{x}) - p_{\hat{\phi}}(\boldsymbol{x})) d\boldsymbol{x} + \int p_{\hat{\phi}}(\boldsymbol{x}) d\boldsymbol{x} \le \epsilon + 1,$$

which implies

$$\int \left( \sqrt{\bar{p}_{\hat{\phi}}(\boldsymbol{x})} - \sqrt{p_{\phi^*}(\boldsymbol{x})} \right)^2 d\boldsymbol{x} \leq 2\left( 1 - \int \sqrt{\bar{p}_{\hat{\phi}}(\boldsymbol{x})p_{\phi^*}(\boldsymbol{x})}d\boldsymbol{x} \right) + \epsilon \tag{13}$$

and

$$\int \left( \sqrt{\bar{p}_{\hat{\phi}}(\boldsymbol{x})} + \sqrt{p_{\phi^*}(\boldsymbol{x})} \right)^2 d\boldsymbol{x} \leq 2\int \bar{p}_{\hat{\phi}}(\boldsymbol{x}) + p_{\phi^*}(\boldsymbol{x})\, d\boldsymbol{x} \leq 2\epsilon + 4. \tag{14}$$

Combining (12) and (13), we show that

$$\int \left( \sqrt{\bar{p}_{\hat{\phi}}(\boldsymbol{x})} - \sqrt{p_{\phi^*}(\boldsymbol{x})} \right)^2 d\boldsymbol{x} \leq \frac{2}{m} \log \frac{N_{[\,]}(\mathcal{P}_{\mathcal{X}\times\mathcal{S}}(\Phi), \epsilon)}{\delta} + \epsilon. \tag{15}$$

By Cauchy-Schwarz inequality, it then holds that

$$\left( \int |\bar{p}_{\hat{\phi}}(\boldsymbol{x}) - p_{\phi^*}(\boldsymbol{x})|\, d\boldsymbol{x} \right)^2 \leq \int \left( \sqrt{\bar{p}_{\hat{\phi}}(\boldsymbol{x})} + \sqrt{p_{\phi^*}(\boldsymbol{x})} \right)^2 d\boldsymbol{x} \cdot \int \left( \sqrt{\bar{p}_{\hat{\phi}}(\boldsymbol{x})} - \sqrt{p_{\phi^*}(\boldsymbol{x})} \right)^2 d\boldsymbol{x},$$

$$\leq (2\epsilon + 4) \cdot \left( \frac{2}{m} \log \frac{N_{[\,]}(\mathcal{P}_{\mathcal{X}\times\mathcal{S}}(\Phi), \epsilon)}{\delta} + \epsilon \right), \tag{16}$$

where the last inequality follows from (14) and (15). Note that

$$\left( \int |p_{\hat{\phi}}(\boldsymbol{x}) - p_{\phi^*}(\boldsymbol{x})|\, d\boldsymbol{x} \right)^2 - \left( \int |\bar{p}_{\hat{\phi}}(\boldsymbol{x}) - p_{\phi^*}(\boldsymbol{x})|\, d\boldsymbol{x} \right)^2$$

$$= \left( \int |p_{\hat{\phi}}(\boldsymbol{x}) - p_{\phi^*}(\boldsymbol{x})| + |\bar{p}_{\hat{\phi}}(\boldsymbol{x}) - p_{\phi^*}(\boldsymbol{x})|\, d\boldsymbol{x} \right) \cdot \left( \int |p_{\hat{\phi}}(\boldsymbol{x}) - p_{\phi^*}(\boldsymbol{x})| - |\bar{p}_{\hat{\phi}}(\boldsymbol{x}) - p_{\phi^*}(\boldsymbol{x})|\, d\boldsymbol{x} \right)$$

$$\leq \left( \int |p_{\hat{\phi}}(\boldsymbol{x}) - p_{\phi^*}(\boldsymbol{x})| + |\bar{p}_{\hat{\phi}}(\boldsymbol{x}) - p_{\phi^*}(\boldsymbol{x})|\, d\boldsymbol{x} \right) \cdot \int |p_{\hat{\phi}}(\boldsymbol{x}) - \bar{p}_{\hat{\phi}}(\boldsymbol{x})|\, d\boldsymbol{x}$$

$$\leq (\epsilon + 4) \cdot \epsilon. \tag{17}$$

Adding (16) and (17) together, we have

$$\left( \int |p_{\hat{\phi}}(\boldsymbol{x}) - p_{\phi^*}(\boldsymbol{x})|\, d\boldsymbol{x} \right)^2 \leq (2\epsilon + 4) \cdot \left( \frac{2}{m} \log \frac{N_{[\,]}(\mathcal{P}_{\mathcal{X}\times\mathcal{S}}(\Phi), \epsilon)}{\delta} + \epsilon \right) + (\epsilon + 4) \cdot \epsilon, \tag{18}$$

which implies

$$d_{\mathrm{TV}}\big(\mathbb{P}_{\hat{\phi}}(\boldsymbol{x}), \mathbb{P}_{\phi^*}(\boldsymbol{x})\big) = \frac{1}{2} \int |p_{\hat{\phi}}(\boldsymbol{x}) - p_{\phi^*}(\boldsymbol{x})|\, d\boldsymbol{x}$$

$$\leq \frac{1}{2}\sqrt{(2\epsilon + 4) \cdot \left( \frac{2}{m} \log \frac{N_{[\,]}(\mathcal{P}_{\mathcal{X}\times\mathcal{S}}(\Phi), \epsilon)}{\delta} + \epsilon \right) + (\epsilon + 4) \cdot \epsilon}. \tag{19}$$

Setting $\epsilon = 1/m$, we have with probability at least $1 - \delta$ that

$$d_{\mathrm{TV}}\big(\mathbb{P}_{\hat{\phi}}(\boldsymbol{x}), \mathbb{P}_{\phi^*}(\boldsymbol{x})\big) \leq \frac{1}{2}\sqrt{\left( \frac{2}{m} + 4 \right) \cdot \left( \frac{2}{m} \log \frac{N_{[\,]}(\mathcal{P}_{\mathcal{X}\times\mathcal{S}}(\Phi), 1/m)}{\delta} + \frac{1}{m} \right) + \left( \frac{1}{m} + 4 \right) \cdot \frac{1}{m}}$$

$$\leq 3 \cdot \sqrt{\frac{1}{m} \log \frac{N_{[\,]}(\mathcal{P}_{\mathcal{X}\times\mathcal{S}}(\Phi), 1/m)}{\delta}}. \tag{20}$$

Thus, we prove Theorem 3.3. $\qquad\square$

## A.2 PROOFS FOR THEOREM 3.4

Before proving the theorem, we first present some useful results that will be used in the proof of Theorem 3.4. Lemma A.1 upper bounds the difference between empirical loss and population loss by an application of bounded difference inequality and a standard symmetrization argument. Lemma A.2 relates excess risks with the total variation distance between probability distributions. For notation simplicity, we denote $\mathbb{E}_{(x,y)\sim\mathbb{P}_{\phi,\psi}(x,y)}$ by $\mathbb{E}_{\phi,\psi}$ in the following. We further denote by $\mathbb{E}$ the expectation taken over the ground truth parameter, i.e., $\mathbb{E} := \mathbb{E}_{(x,y)\sim\mathbb{P}_{\phi^*,\psi^*}(x,y)}$.

**Lemma A.1.** *Suppose that $\ell(\cdot, \cdot)$ is a $L$-bounded loss function. For any given $\phi \in \Phi$, with probability at least $1 - \delta$,*

$$\sup_{\psi \in \Psi} \left| \mathbb{E}[\ell(g_{\phi,\psi}(x), y)] - \frac{1}{n} \sum_{j=1}^{n} \ell(g_{\phi,\psi}(x_j), y_j) \right| \leq R_n(\ell \circ \mathcal{G}_{\phi,\Psi}) + L\sqrt{\frac{2\log(2/\delta)}{n}}, \quad (21)$$

*where $R_n(\ell \circ \mathcal{G}_{\phi,\Psi})$ is the Rademacher complexity of the function class $\ell \circ \mathcal{G}_{\phi,\Psi}$ defined in Theorem 3.4.*

*Proof of Lemma A.1.* First notice that, when a pair $(x_j, y_j)$ changes, since $\ell$ is $L$-bounded, the random variable

$$\sup_{\psi \in \Psi} \left( \mathbb{E}[\ell(g_{\phi,\psi}(x), y)] - \frac{1}{n} \sum_{j=1}^{n} \ell(g_{\phi,\psi}(x_j), y_j) \right) \quad (22)$$

can change by no more than $2L/n$. McDiarmid's inequality implies that with probability at least $1 - \delta/2$,

$$\sup_{\psi \in \Psi} \left( \mathbb{E}[\ell(g_{\phi,\psi}(x), y)] - \frac{1}{n} \sum_{j=1}^{n} \ell(g_{\phi,\psi}(x_j), y_j) \right)$$

$$\leq \mathbb{E}\left[ \sup_{\psi \in \Psi} \left( \mathbb{E}[\ell(g_{\phi,\psi}(x), y)] - \frac{1}{n} \sum_{j=1}^{n} \ell(g_{\phi,\psi}(x_j), y_j) \right) \right] + L\sqrt{\frac{2\log(2/\delta)}{n}}. \quad (23)$$

Let $\{x_j', y_j'\}_{j=1}^{n}$ be independent copies of $\{x_j, y_j\}_{j=1}^{n}$ and $\{\sigma_j\}_{j=1}^{n}$ be i.i.d. Rademacher random variables. Using the standard symmetrization technique, we have

$$\mathbb{E}\left[ \sup_{\psi \in \Psi} \left( \mathbb{E}[\ell(g_{\phi,\psi}(x), y)] - \frac{1}{n} \sum_{j=1}^{n} \ell(g_{\phi,\psi}(x_j), y_j) \right) \right]$$

$$= \mathbb{E}\left[ \sup_{\psi \in \Psi} \mathbb{E}\left[ \frac{1}{n} \sum_{j=1}^{n} \ell(g_{\phi,\psi}(x_j'), y_j') - \frac{1}{n} \sum_{j=1}^{n} \ell(g_{\phi,\psi}(x_j), y_j) \Big| \{x_j, y_j\}_{j=1}^{n} \right] \right]$$

$$\leq \mathbb{E}\left[ \sup_{\psi \in \Psi} \left( \frac{1}{n} \sum_{j=1}^{n} \ell(g_{\phi,\psi}(x_j'), y_j') - \frac{1}{n} \sum_{j=1}^{n} \ell(g_{\phi,\psi}(x_j), y_j) \right) \right]$$

$$\leq \mathbb{E}\left[ \sup_{\psi \in \Psi} \frac{1}{n} \sum_{j=1}^{n} \sigma_j \left( \ell(g_{\phi,\psi}(x_j'), y_j') - \ell(g_{\phi,\psi}(x_j), y_j) \right) \right]$$

$$\leq 2\mathbb{E}\left[ \sup_{\psi \in \Psi} \frac{1}{n} \sum_{j=1}^{n} \sigma_j \ell(g_{\phi,\psi}(x_j), y_j) \right]$$

$$= R_n(\ell \circ \mathcal{G}_{\phi,\Psi}). \quad (24)$$

Therefore, with probability at least $1 - \delta/2$,

$$\sup_{\psi \in \Psi} \left( \mathbb{E}[\ell(g_{\phi,\psi}(x), y)] - \frac{1}{n} \sum_{j=1}^{n} \ell(g_{\phi,\psi}(x_j), y_j) \right) \leq R_n(\ell \circ \mathcal{G}_{\phi,\Psi}) + L\sqrt{\frac{2\log(2/\delta)}{n}} \quad (25)$$

Similarly, with probability at least $1 - \delta/2$,

$$\sup_{\psi \in \Psi} \left( \frac{1}{n} \sum_{j=1}^{n} \ell(g_{\phi,\psi}(x_j), y_j) - \mathbb{E}[\ell(g_{\phi,\psi}(x), y)] \right) \leq R_n(\ell \circ \mathcal{G}_{\phi,\Psi}) + L\sqrt{\frac{2\log(2/\delta)}{n}} \quad (26)$$

Combine these together, we prove Lemma A.1. $\qquad \square$

**Lemma A.2.** *Suppose that $\ell(\cdot, \cdot)$ is a $L$-bounded loss function. Then, it holds for any $\phi \in \Phi, \psi \in \Psi$ that*

$$\mathbb{E}[\ell(g_{\phi,\psi}(x), y)] - \mathbb{E}[\ell(g_{\phi^*,\psi^*}(x), y)] \leq 4L \cdot d_{\mathrm{TV}}(\mathbb{P}_{\phi,\psi}(x, y), \mathbb{P}_{\phi^*,\psi^*}(x, y)). \quad (27)$$

*Proof of Lemma A.2.*

$$\mathbb{E}_{\phi^*,\psi^*}[\ell(g_{\phi,\psi}(x),y)] - \mathbb{E}_{\phi^*,\psi^*}[\ell(g_{\phi^*,\psi^*}(x),y)]$$
$$= \mathbb{E}_{\phi^*,\psi^*}[\ell(g_{\phi,\psi}(x),y)] - \mathbb{E}_{\phi,\psi}[\ell(g_{\phi,\psi}(x),y)]$$
$$+ \mathbb{E}_{\phi,\psi}[\ell(g_{\phi,\psi}(x),y)] - \mathbb{E}_{\phi,\psi}[\ell(g_{\phi^*,\psi^*}(x),y)]$$
$$+ \mathbb{E}_{\phi,\psi}[\ell(g_{\phi^*,\psi^*}(x),y)] - \mathbb{E}_{\phi^*,\psi^*}[\ell(g_{\phi^*,\psi^*}(x),y)]. \tag{28}$$

First notice that, by definition of $g_{\phi,\psi}$,

$$\mathbb{E}_{\phi,\psi}[\ell(g_{\phi,\psi}(x),y)] - \mathbb{E}_{\phi,\psi}[\ell(g_{\phi^*,\psi^*}(x),y)] \le 0. \tag{29}$$

For the other two terms, based on the fact that $\ell$ is $L$-bounded, we have

$$|\mathbb{E}_{\phi^*,\psi^*}[\ell(g_{\phi,\psi}(x),y)] - \mathbb{E}_{\phi,\psi}[\ell(g_{\phi,\psi}(x),y)]|$$
$$= \left| \int \ell(g_{\phi,\psi}(x),y)p_{\phi_*,\psi_*}(x,y)\mathrm{d}x\mathrm{d}y - \int \ell(g_{\phi,\psi}(x),y)p_{\phi,\psi}(x,y)\mathrm{d}x\mathrm{d}y \right|$$
$$= \left| \int \ell(g_{\phi,\psi}(x),y)(p_{\phi_*,\psi_*}(x,y) - p_{\phi,\psi}(x,y))\mathrm{d}x\mathrm{d}y \right|$$
$$\le \int |\ell(g_{\phi,\psi}(x),y)||(p_{\phi_*,\psi_*}(x,y) - p_{\phi,\psi}(x,y))|\mathrm{d}x\mathrm{d}y$$
$$\le \int L|(p_{\phi_*,\psi_*}(x,y) - p_{\phi,\psi}(x,y))|\mathrm{d}x\mathrm{d}y$$
$$= 2L \cdot d_{\mathrm{TV}}(P_{\phi,\psi}(x,y), P_{\phi^*,\psi^*}(x,y)). \tag{30}$$

Similarly, it holds that

$$|\mathbb{E}_{\phi^*,\psi^*}[\ell(g_{\phi^*,\psi^*}(x),y)] - \mathbb{E}_{\phi,\psi}[\ell(g_{\phi^*,\psi^*}(x),y)]| \le 2L \cdot d_{\mathrm{TV}}(P_{\phi,\psi}(x,y), P_{\phi^*,\psi^*}(x,y)). \tag{31}$$

Combining (28), (29), (30) and (31), we obtain

$$\mathbb{E}_{\phi^*,\psi^*}[\ell(g_{\phi,\psi}(x),y)] - \mathbb{E}_{\phi^*,\psi^*}[\ell(g_{\phi^*,\psi^*}(x),y)] \le 4L \cdot d_{\mathrm{TV}}(P_{\phi,\psi}(x,y), P_{\phi^*,\psi^*}(x,y)). \tag{32}$$

$\square$

With Lemma A.1 and Lemma A.2, we are able to state our proofs for Theorem 3.4 in the following. The main idea of the proof is decomposing the risk. And a key observation is that the labeled data $\{x_j, y_j\}_{j=1}^n$ are independent of the pretrained $\hat{\phi}$, which is learned from the unlabeled data $\{x_i\}_{i=1}^m$.

*Proof of Theorem 3.4.* Let

$$\tilde{\psi} := \operatorname*{arg\,min}_{\psi \in \Psi} d_{\mathrm{TV}}(P_{\hat{\phi},\psi}(x,y), P_{\phi^*,\psi^*}(x,y)). \tag{33}$$

And for any $\phi \in \Phi, \psi \in \Psi$, we define

$$\Delta_{\phi,\psi} := \mathbb{E}_{\phi^*,\psi^*}[\ell(g_{\phi,\psi}(x),y)] - \frac{1}{n}\sum_{j=1}^n \ell(g_{\phi,\psi}(x_j),y_j). \tag{34}$$

Recall that the excess risk is defined in (2). It then holds that

$$\mathrm{Error}_\ell(\hat{\phi},\hat{\psi}) = \mathbb{E}_{\phi^*,\psi^*}[\ell(g_{\hat{\phi},\hat{\psi}}(x),y)] - \mathbb{E}_{\phi^*,\psi^*}[\ell(g_{\phi^*,\psi^*}(x),y)]$$

$$= \mathbb{E}_{\phi^*,\psi^*}[\ell(g_{\hat{\phi},\hat{\psi}}(x),y)] - \frac{1}{n}\sum_{j=1}^n \ell(g_{\hat{\phi},\hat{\psi}}(x_j),y_j)$$

$$+ \frac{1}{n}\sum_{j=1}^n \ell(g_{\hat{\phi},\hat{\psi}}(x_j),y_j) - \frac{1}{n}\sum_{j=1}^n \ell(g_{\hat{\phi},\tilde{\psi}}(x_j),y_j) \quad (\le 0, \text{ by ERM in Algorithm 1})$$

$$+ \frac{1}{n}\sum_{j=1}^n \ell(g_{\hat{\phi},\tilde{\psi}}(x_j),y_j) - \mathbb{E}_{\phi^*,\psi^*}[\ell(g_{\hat{\phi},\tilde{\psi}}(x),y)]$$

$$+ \mathbb{E}_{\phi^*,\psi^*}[\ell(g_{\hat{\phi},\tilde{\psi}}(x),y)] - \mathbb{E}_{\phi^*,\psi^*}[\ell(g_{\phi^*,\psi^*}(x),y)]$$

$$\le \Delta_{\hat{\phi},\hat{\psi}} - \Delta_{\hat{\phi},\tilde{\psi}} + \mathbb{E}_{\phi^*,\psi^*}[\ell(g_{\hat{\phi},\tilde{\psi}}(x),y)] - \mathbb{E}_{\phi^*,\psi^*}[\ell(g_{\phi^*,\psi^*}(x),y)]. \tag{35}$$

By lemma A.2, we have

$$\mathbb{E}_{\phi^*,\psi^*}[\ell(g_{\hat{\phi},\tilde{\psi}}(x),y)] - \mathbb{E}_{\phi^*,\psi^*}[\ell(g_{\phi^*,\psi^*}(x),y)]$$
$$\leq 4L \cdot d_{\mathrm{TV}}(P_{\hat{\phi},\tilde{\psi}}(x,y), P_{\phi^*,\psi^*}(x,y))$$
$$= 4L \cdot \min_{\psi \in \Psi} d_{\mathrm{TV}}(P_{\hat{\phi},\psi}(x,y), P_{\phi^*,\psi^*}(x,y)) \quad \text{(by definition of } \tilde{\psi})$$
$$\leq 4\kappa L \cdot d_{\mathrm{TV}}(P_{\hat{\phi}}(x,s), P_{\phi^*}(x,s)). \tag{36}$$

The last line holds, since by Assumption 3.2, for any $\hat{\phi} \in \Phi$, we choose $T_1$ that satisfies (5) and $T_2$ that satisfies (6). Let $\psi = T_2^{-1} \circ \psi^*$. It then holds that

$$\min_{\psi \in \Psi} d_{\mathrm{TV}}(P_{\hat{\phi},\psi}(x,y), P_{\phi^*,\psi^*}(x,y)) \leq d_{\mathrm{TV}}\left(\mathbb{P}_{\hat{\phi},\psi}(x,y), \mathbb{P}_{\phi^*,\psi^*}(x,y)\right)$$
$$= d_{\mathrm{TV}}\left(\mathbb{P}_{T_1 \circ \hat{\phi},\psi^*}(x,y), \mathbb{P}_{\phi^*,\psi^*}(x,y)\right)$$
$$\leq d_{\mathrm{TV}}\left(\mathbb{P}_{T_1 \circ \hat{\phi}}(x,z), \mathbb{P}_{\phi^*}(x,z)\right)$$
$$\leq \kappa \cdot d_{\mathrm{TV}}\left(\mathbb{P}_{\hat{\phi}}(x,s), \mathbb{P}_{\phi^*}(x,s)\right). \tag{37}$$

Combining (35) and (36), we have

$$\mathrm{Error}_\ell(\hat{\phi}, \hat{\psi}) \leq \Delta_{\hat{\phi},\hat{\psi}} - \Delta_{\hat{\phi},\tilde{\psi}} + 4\kappa L \cdot d_{\mathrm{TV}}(P_{\hat{\phi}}(x,s), P_{\phi^*}(x,s)). \tag{38}$$

We define the following events

$$D := \left\{ d_{\mathrm{TV}}(P_{\hat{\phi}}(x,s), P_{\phi^*}(x,s)) \leq 3\sqrt{\frac{1}{m} \log \frac{2N(\mathcal{P}_{\mathcal{X} \times \mathcal{S}}(\Phi), 1/m)}{\delta}} \right\} \tag{39}$$

and

$$R := \left\{ \sup_{\psi \in \Psi} |\Delta_{\hat{\phi},\psi}| \leq R_n(\ell \circ \mathcal{G}_{\hat{\phi},\Psi}) + L\sqrt{\frac{2\log(4/\delta)}{n}} \right\}. \tag{40}$$

It holds that

$$\mathbb{P}(D \cap R) = \mathbb{E}[\mathbb{1}_{D \cap R}] = \mathbb{E}[\mathbb{E}[\mathbb{1}_D \mathbb{1}_R | \hat{\phi}]] = \mathbb{E}[\mathbb{1}_D \mathbb{E}[\mathbb{1}_R | \hat{\phi}]] = \mathbb{E}[\mathbb{1}_D \mathbb{P}(R | \hat{\phi})], \tag{41}$$

where the third equation follows from the fact that $D$ is $\hat{\phi}$-measurable. Note that $\{x_j, y_j\}_{j=1}^n$ is independent of $\hat{\phi}$. By Lemma A.1, for any given $\hat{\phi}$, with probability at least $1 - \delta/2$,

$$\sup_{\psi \in \Psi} |\Delta_{\hat{\phi},\psi}| \leq R_n(\ell \circ \mathcal{G}_{\hat{\phi},\Psi}) + L\sqrt{\frac{2\log(4/\delta)}{n}}, \tag{42}$$

i.e.,

$$\mathbb{P}(R | \hat{\phi}) \geq 1 - \delta/2. \tag{43}$$

By Lemma 3.3, with probability at least $1 - \delta/2$, the output of the first step of our algorithm $\hat{\phi}$, satisfies

$$d_{\mathrm{TV}}(P_{\hat{\phi}}(x,s), P_{\phi^*}(x,s)) \leq 3\sqrt{\frac{1}{m} \log \frac{2N(\mathcal{P}_{\mathcal{X} \times \mathcal{S}}(\Phi), 1/m)}{\delta}} \tag{44}$$

i.e.,

$$\mathbb{P}(D) \geq 1 - \delta/2. \tag{45}$$

By (41), (43) and (45), we have

$$\mathbb{P}(D \cap R) \geq (1 - \delta/2)^2 \geq 1 - \delta. \tag{46}$$

Then, under event $D \cap R$, by our decomposition (38), we have

$$
\begin{aligned}
\text{Error}_\ell(\hat{\phi}, \hat{\psi}) &\leq \Delta_{\hat{\phi}, \hat{\psi}} - \Delta_{\hat{\phi}, \tilde{\psi}} + 4\kappa L \cdot d_{\text{TV}}(P_{\hat{\phi}}(x, s), P_{\phi^*}(x, s)) \\
&\leq 2 \sup_{\psi \in \Psi} |\Delta_{\hat{\phi}, \psi}| + 4\kappa L \cdot d_{\text{TV}}(P_{\hat{\phi}}(x, s), P_{\phi^*}(x, s)) \\
&\leq 2 R_n(\ell \circ \mathcal{G}_{\hat{\phi}, \Psi}) + 2L\sqrt{\frac{2\log(4/\delta)}{n}} + 12\kappa L\sqrt{\frac{1}{m} \log \frac{2N(\mathcal{P}_{\mathcal{X} \times \mathcal{S}}(\Phi), 1/m)}{\delta}} \\
&\leq 2 \max_{\phi \in \Phi} R_n(\ell \circ \mathcal{G}_{\phi, \Psi}) + 2L\sqrt{\frac{2\log(4/\delta)}{n}} + 12\kappa L\sqrt{\frac{1}{m} \log \frac{2N(\mathcal{P}_{\mathcal{X} \times \mathcal{S}}(\Phi), 1/m)}{\delta}}.
\end{aligned}
$$
(47)

Thus, we prove Theorem 3.4.

$\square$

## A.3   PROOFS FOR COROLLARY 3.5

In the following, we give the proof of Corollary 3.5, which is based on the analysis of Gaussian complexity.

*Proof.* By Theorem 3.4, we have

$$
\text{Error}_\ell(\hat{\phi}, \hat{\psi}) \leq 2 \max_{\phi \in \Phi} R_n(\ell \circ \mathcal{G}_{\phi, \Psi}) + 2L \cdot \sqrt{\frac{2}{n} \log \frac{4}{\delta}} + 12\kappa L \cdot \sqrt{\frac{1}{m} \log \frac{2N_{[\,]}(\mathcal{P}_{\mathcal{X} \times \mathcal{S}}(\Phi), 1/m)}{\delta}}.
$$
(48)

Therefore, it remains to bound the Rademacher complexity term. By Ledoux & Talagrand (2013), the Rademacher complexity is upper bounded by the Gaussian complexity, i.e.,

$$
R_n(\mathcal{F}) \leq c \cdot G_n(\mathcal{F}) = c \cdot \mathbb{E}\hat{G}_n(\mathcal{F}),
$$
(49)

where $c$ is some absolute constants. Here $G_n(\mathcal{F})$ is the Gaussian complexity, and it's empirical version is defined as

$$
\hat{G}_n(\mathcal{F}) := \mathbb{E}_{g_i}\left[ \sup_{f \in \mathcal{F}} \left| \frac{2}{n} \sum_{i=1}^n g_i f(x_i) \right| \, \middle| \, x_1, \cdots, x_n \right]
$$
(50)

where $g_1, \cdots, g_n$ are i.i.d. $\mathcal{N}(0, 1)$ random variables. By (5.36) in Wainwright (2019), we have

$$
\begin{aligned}
\hat{G}_n(\ell \circ \mathcal{G}_{\phi, \Psi}) &\leq \frac{1}{\sqrt{n}} \cdot \min_{\delta \in [0, L]} \left\{ \delta\sqrt{n} + 2L\sqrt{\log N(\ell \circ \mathcal{G}_{\phi, \Psi}, \delta, \|\cdot\|_\infty)} \right\} \\
&\leq \frac{1}{\sqrt{n}} \left( L + 2L\sqrt{\log N(\ell \circ \mathcal{G}_{\phi, \Psi}, L/\sqrt{n}, \|\cdot\|_\infty)} \right) \quad (\text{Take } \delta = L/\sqrt{n}) \\
&\leq 3L\sqrt{\frac{\log N(\ell \circ \mathcal{G}_{\phi, \Psi}, L/\sqrt{n}, \|\cdot\|_\infty)}{n}}.
\end{aligned}
$$
(51)

Combining (49) and (51), we obtain

$$
R_n(\ell \circ \mathcal{G}_{\phi, \Psi}) \leq 3cL\sqrt{\frac{\log N(\ell \circ \mathcal{G}_{\phi, \Psi}, L/\sqrt{n}, \|\cdot\|_\infty)}{n}}.
$$
(52)

By (48) and (52), we finish the proof. $\square$

## A.4   PROOFS FOR THEOREM 3.7

In this section, we first show the relation of Assumption 3.2 and Assumption 3.6. We then show that the MLE step in line 2 of Algorithm 1 guarantees an upper bound on the Hellinger distance $H(\mathbb{P}_{\hat{\phi}}(x, s), \mathbb{P}_{\phi^*}(x, s))$. Then, using the same techniques as that in the proof of Theorem 3.4, we prove Theorem 3.7.

### A.4.1 RELATION OF ASSUMPTION 3.2 AND ASSUMPTION 3.6

Assumption 3.6 is actually a relaxation of Assumption 3.2. To see this, by Assumption 3.2, for any $\phi \in \Phi$, we choose $T_1$ that satisfies (5) and $T_2$ that satisfies (6). Let $\psi = T_2^{-1} \circ \psi^*$. It then holds that

$$
\begin{aligned}
& d_{\mathrm{TV}}\big(\mathbb{P}_{\phi,\psi}(x,y), \mathbb{P}_{\phi^*,\psi^*}(x,y)\big) \\
&= d_{\mathrm{TV}}\big(\mathbb{P}_{T_1 \circ \phi, \psi^*}(x,y), \mathbb{P}_{\phi^*,\psi^*}(x,y)\big) \\
&\leq d_{\mathrm{TV}}\big(\mathbb{P}_{T_1 \circ \phi}(x,z), \mathbb{P}_{\phi^*}(x,z)\big) \\
&\leq \kappa \cdot d_{\mathrm{TV}}\big(\mathbb{P}_{\phi}(x,s), \mathbb{P}_{\phi^*}(x,s)\big).
\end{aligned}
$$

Note that the TV distance can be upper bounded by the Hellinger distance. Thus, Assumption 3.2 directly implies Assumption 3.6.

### A.4.2 HELLINGER DISTANCE GUARANTEE

Suppose that $\hat{\phi}$ is the output of the MLE step in Algorithm 1, which satisfies

$$
\hat{\phi} \leftarrow \arg\max_{\phi \in \Phi} \sum_{i=1}^{m} \log p_\phi(x_i, s_i). \tag{53}
$$

We have the following theoretical guarantee on the Hellinger distance between $\mathbb{P}_{\hat{\phi}}(x,s)$ and $\mathbb{P}_{\phi^*}(x,s)$.

**Lemma A.3.** *Let $\hat{\phi}$ be the output of Algorithm 1. It then holds that with probability at least $1 - \delta$ that*

$$
H\big(\mathbb{P}_{\hat{\phi}}(x,s), \mathbb{P}_{\phi^*}(x,s)\big) \leq \sqrt{\frac{2}{m} \log \frac{N_{[\,]}\big(\mathcal{P}_{\mathcal{X} \times \mathcal{S}}(\Phi), 1/m^2\big)}{\delta}}, \tag{54}
$$

*where we denote $\mathcal{P}_{\mathcal{X} \times \mathcal{S}}(\Phi) := \{p_\phi(x,s) \,|\, \phi \in \Phi\}$.*

*Proof of Lemma A.3.* For notation simplicity, we denote $\boldsymbol{x} := (x,s)$. Let $\epsilon > 0$. Similar to the proof of Theorem 3.3, we obtain with probability at least $1 - \delta$

$$
1 - \int \sqrt{\bar{p}_{\hat{\phi}}(\boldsymbol{x}) p_{\phi^*}(\boldsymbol{x})} \, d\boldsymbol{x} \leq \frac{1}{m} \log \frac{N_{[\,]}\big(\mathcal{P}_{\mathcal{X} \times \mathcal{S}}(\Phi), \epsilon\big)}{\delta}. \tag{55}
$$

Here $\bar{p}_{\hat{\phi}}(\boldsymbol{x}) \in \mathcal{N}_{[\,]}(\mathcal{P}_{\mathcal{X} \times \mathcal{S}}(\Phi), \epsilon)$ that satisfies $\bar{p}_{\hat{\phi}}(\boldsymbol{x}) \geq p_{\phi^*}(\boldsymbol{x})$ for any $\boldsymbol{x}$ and

$$
\int \bar{p}_{\hat{\phi}}(\boldsymbol{x}) - p_{\phi^*}(\boldsymbol{x}) \, d\boldsymbol{x} \leq \epsilon. \tag{56}
$$

Note that

$$
\begin{aligned}
& 1 - \int \sqrt{p_{\hat{\phi}}(\boldsymbol{x}) p_{\phi^*}(\boldsymbol{x})} \, d\boldsymbol{x} - \left(1 - \int \sqrt{\bar{p}_{\hat{\phi}}(\boldsymbol{x}) p_{\phi^*}(\boldsymbol{x})} \, d\boldsymbol{x}\right) \\
&= \int \left(\sqrt{\bar{p}_{\hat{\phi}}(\boldsymbol{x})} - \sqrt{p_{\hat{\phi}}(\boldsymbol{x})}\right) \sqrt{p_{\phi^*}(\boldsymbol{x})} \, d\boldsymbol{x} \\
&\leq \sqrt{\int \left(\sqrt{\bar{p}_{\hat{\phi}}(\boldsymbol{x})} - \sqrt{p_{\hat{\phi}}(\boldsymbol{x})}\right)^2 d\boldsymbol{x}} \\
&= \sqrt{\int \bar{p}_{\hat{\phi}}(\boldsymbol{x}) + p_{\hat{\phi}}(\boldsymbol{x}) - 2\sqrt{\bar{p}_{\hat{\phi}}(\boldsymbol{x}) p_{\hat{\phi}}(\boldsymbol{x})} \, d\boldsymbol{x}} \\
&\leq \sqrt{\int \bar{p}_{\hat{\phi}}(\boldsymbol{x}) - p_{\hat{\phi}}(\boldsymbol{x}) \, d\boldsymbol{x}} \\
&\leq \sqrt{\epsilon}. \tag{57}
\end{aligned}
$$

Here the first inequality follows from Cauchy-Schwarz inequality and the second follows from the fact that $\sqrt{\bar{p}_{\hat{\phi}}(\boldsymbol{x})p_{\hat{\phi}}(\boldsymbol{x})} \geq p_{\hat{\phi}}(\boldsymbol{x})$. By (55) and (57), we have

$$1 - \int \sqrt{p_{\hat{\phi}}(\boldsymbol{x})p_{\phi^*}(\boldsymbol{x})}\,d\boldsymbol{x} \leq \sqrt{\epsilon} + \frac{1}{m}\log\frac{N_{[\,]}\big(\mathcal{P}_{\mathcal{X}\times\mathcal{S}}(\Phi),\epsilon\big)}{\delta}, \tag{58}$$

which implies that

$$H^2\big(\mathbb{P}_{\hat{\phi}}(\boldsymbol{x}),\mathbb{P}_{\phi^*}(\boldsymbol{x})\big) = 1 - \int \sqrt{p_{\hat{\phi}}(\boldsymbol{x})p_{\phi^*}(\boldsymbol{x})}\,d\boldsymbol{x} \leq \sqrt{\epsilon} + \frac{1}{m}\log\frac{N_{[\,]}\big(\mathcal{P}_{\mathcal{X}\times\mathcal{S}}(\Phi),\epsilon\big)}{\delta}. \tag{59}$$

Set $\epsilon = 1/m^2$. We have

$$H^2\big(\mathbb{P}_{\hat{\phi}}(x,s),\mathbb{P}_{\phi^*}(x,s)\big) \leq \frac{2}{m}\log\frac{N_{[\,]}\big(\mathcal{P}_{\mathcal{X}\times\mathcal{S}}(\Phi),1/m^2\big)}{\delta}. \tag{60}$$

$\square$

### A.4.3 PROOF OF THEOREM 3.7

With Lemma A.3 in hand, we are ready to prove Theorem 3.7.

*Proof of Theorem 3.7.* Let $\hat{\phi}$ be the output of the MLE step in Algorithm 1. And for any $\phi \in \Phi, \psi \in \Psi$, we define

$$\Delta_{\phi,\psi} := \mathbb{E}_{\phi^*,\psi^*}[\ell(g_{\phi,\psi}(x),y)] - \frac{1}{n}\sum_{j=1}^{n}\ell(g_{\phi,\psi}(x_j),y_j). \tag{61}$$

Following the same arguments as that in the proof of Theorem 3.4, we have with probability at least $1-\delta$,

$$H\big(\mathbb{P}_{\hat{\phi}}(x,s),\mathbb{P}_{\phi^*}(x,s)\big) \leq \sqrt{\frac{2}{m}\log\frac{2N(\mathcal{P}_{\mathcal{X}\times\mathcal{S}}(\Phi),1/m^2)}{\delta}} \tag{62}$$

and

$$\sup_{\psi\in\Psi}|\Delta_{\hat{\phi},\psi}| \leq R_n(\ell\circ\mathcal{G}_{\hat{\phi},\Psi}) + L\sqrt{\frac{2\log(4/\delta)}{n}}. \tag{63}$$

Moreover, as mentioned in (35), we have

$$\mathrm{Error}_\ell(\hat{\phi},\hat{\psi}) \leq \Delta_{\hat{\phi},\hat{\psi}} - \Delta_{\hat{\phi},\tilde{\psi}} + \mathbb{E}_{\phi^*,\psi^*}[\ell(g_{\hat{\phi},\tilde{\psi}}(x),y)] - \mathbb{E}_{\phi^*,\psi^*}[\ell(g_{\phi^*,\psi^*}(x),y)]$$

$$\leq 2R_n(\ell\circ\mathcal{G}_{\hat{\phi},\Psi}) + 2L\sqrt{\frac{2\log(4/\delta)}{n}}$$

$$+ \mathbb{E}_{\phi^*,\psi^*}[\ell(g_{\phi^*,\psi^*}(x),y)] - \mathbb{E}_{\phi^*,\psi^*}[\ell(g_{\phi^*,\psi^*}(x),y)], \tag{64}$$

where $\tilde{\psi} := \arg\min_{\psi\in\Psi} d_{\mathrm{TV}}(\mathbb{P}_{\hat{\phi},\psi}(x,y),\mathbb{P}_{\phi^*,\psi^*}(x,y))$ and the second inequality follows from (63). By lemma A.2, we have

$$\mathbb{E}_{\phi^*,\psi^*}[\ell(g_{\hat{\phi},\tilde{\psi}}(x),y)] - \mathbb{E}_{\phi^*,\psi^*}[\ell(g_{\phi^*,\psi^*}(x),y)]$$

$$\leq 4L\cdot d_{\mathrm{TV}}(\mathbb{P}_{\hat{\phi},\tilde{\psi}}(x,y),\mathbb{P}_{\phi^*,\psi^*}(x,y))$$

$$= 4L\cdot\min_{\psi\in\Psi} d_{\mathrm{TV}}(\mathbb{P}_{\hat{\phi},\psi}(x,y),\mathbb{P}_{\phi^*,\psi^*}(x,y)) \quad\text{(by definition of }\tilde{\psi})$$

$$\leq_{1)} 4\kappa L\cdot H(\mathbb{P}_{\hat{\phi}}(x,s),\mathbb{P}_{\phi^*}(x,s))$$

$$\leq_{2)} 4\kappa L\sqrt{\frac{2}{m}\log\frac{2N(\mathcal{P}_{\mathcal{X}\times\mathcal{S}}(\Phi),1/m^2)}{\delta}}, \tag{65}$$

where 1) follows from Assumption 3.6 and 2) follows from (62). Combining (64) and (65), we have

$$\mathrm{Error}_\ell(\hat{\phi},\hat{\psi}) \leq 2R_n(\ell\circ\mathcal{G}_{\hat{\phi},\Psi}) + 2L\sqrt{\frac{2\log(4/\delta)}{n}} + 4\kappa L\sqrt{\frac{2}{m}\log\frac{2N(\mathcal{P}_{\mathcal{X}\times\mathcal{S}}(\Phi),1/m^2)}{\delta}}$$

$$\leq 2\max_{\phi\in\Phi} R_n(\ell\circ\mathcal{G}_{\phi,\Psi}) + 2L\sqrt{\frac{2\log(4/\delta)}{n}} + 4\kappa L\sqrt{\frac{2}{m}\log\frac{2N(\mathcal{P}_{\mathcal{X}\times\mathcal{S}}(\Phi),1/m^2)}{\delta}}. \tag{66}$$

$\square$

## B  ADDITIONAL RESULTS AND PROOFS FOR SECTION 4

In Section B.1, by analysing the total variation distance between two high-dimensional Gaussians and applying the Davis-Kahan theorem, we show that factor model with linear regression as downstream tasks has $\kappa$-transferability (Lemma 4.2), where $\kappa$ depends on the largest and smallest singular value of the ground truth parameter $B^*$. In Section B.2 and Section B.3, we prove two lemmas that will be used in the proof of Theorem 4.3. To be specific, in Section B.2, we upper bound the bracketing number of the set $\mathcal{P}(\mathcal{B})$ by using $\epsilon$-discretization (Lemma B.5). In Section B.3, we prove Lemma B.6, which will be used to upper bound the Rademacher complexity of the function class $\ell \circ \mathcal{G}_{B,\mathcal{C}}$. In Section B.4, we prove Theorem 4.3. Finally, in Section B.5, we provide a refined analysis for proving Theorem B.9.

### B.1  PROOFS FOR LEMMA 4.2

First of all, we present some useful lemmas that will be used in the proof of Lemma 4.2. Given two high-dimensional Gaussians, we can bound their total variation distance as follows.

**Lemma B.1** (Theorem 1.2 and Proposition 2.1 in Devroye et al. (2018)). *Suppose that $d > 1$. Let $\mu_1 \neq \mu_2 \in \mathbb{R}^d$. Then, we have*

$$\frac{1}{200} \leq \frac{d_{\mathrm{TV}}\big(\mathcal{N}(\mu_1, I_d), \mathcal{N}(\mu_2, I_d)\big)}{\min\{1, \|\mu_1 - \mu_2\|_2\}} \leq 1.$$

**Lemma B.2** (Theorem 1.1 in Devroye et al. (2018)). *Suppose that $d > 1$. Let $\mu \in \mathbb{R}^d$ and $\Sigma_1 \neq \Sigma_2$ be positive definite $d \times d$ matrices. Then, we have*

$$\frac{1}{100} \leq \frac{d_{\mathrm{TV}}\big(\mathcal{N}(\mu, \Sigma_1), \mathcal{N}(\mu, \Sigma_2)\big)}{\min\{1, \|\Sigma_1^{-1/2}\Sigma_2\Sigma_1^{-1/2} - I_d\|_{\mathrm{F}}\}} \leq \frac{3}{2}.$$

Recall that we define $\mathcal{B} := \{B \in \mathbb{R}^{d \times r} \mid \|B\|_2 \leq D\}$. Let $B \in \mathcal{B}$ and $B^*$ be the ground truth parameter. We denote by $\sigma_{\max}^*$ and $\sigma_{\min}^*$ the largest and smallest singular value of $B^*$, respectively. Moreover, we denote the singular value decomposition of $B$ and $B^*$ by $B = U\Sigma V$ and $B^* = U^*\Sigma^*V^*$, respectively. Here $\Sigma, \Sigma^* \in \mathbb{R}^{r \times r}$ are diagonal matrices and $U, U^* \in \mathbb{R}^{d \times r}$, $V, V^* \in \mathbb{R}^{r \times d}$ are matrices with orthogonal columns. Let

$$M := BB^T = U\Lambda U^T, \quad M^* := B^*B^{*T} = U^*\Lambda^*U^{*T}, \tag{67}$$

where $\Lambda := \Sigma\Sigma^T$ and $\Lambda^* := \Sigma^*\Sigma^{*T}$. We define

$$O := \underset{O \in \mathcal{O}^{r \times r}}{\arg\min} \|UO - U^*\|_{\mathrm{F}}. \tag{68}$$

Then, we have the following lemmas.

**Lemma B.3.** *For $M, M^*$ defined in (67) and $O$ defined in (68), there exists some absolute constants $c > 1$ such that*

$$\|UO - U^*\|_{\mathrm{F}} \leq \frac{c}{(\sigma_{\min}^*)^2}\|M - M^*\|_{\mathrm{F}}.$$

*Here $\sigma_{\min}^*$ is the smallest singular value of the true parameter $B^*$.*

*Proof.* An application of Davis-Kahan Theorem (Davis & Kahan, 1970). □

**Lemma B.4.** *For $M, M^*$ defined in (67) and $O$ defined in (68), there exists some absolute constants $c$ such that*

$$\|\Lambda^{1/2}O - O\Lambda^{*1/2}\|_{\mathrm{F}} \leq \frac{4c(\sigma_{\max}^*)^2}{(\sigma_{\min}^*)^3}\|M - M^*\|_{\mathrm{F}}.$$

*Here $\sigma_{\min}^*$ is the smallest singular value of the true parameter $B^*$.*

*Proof of Lemma B.4.* Our proof is inspired by Ma et al. (2018). By Lemma 2.1 in Schmitt (1992), we have

$$\|\Lambda^{1/2}O - O\Lambda^{*1/2}\|_{\mathrm{F}} \leq \frac{1}{\sqrt{\sigma_{\min}(M^*)}}\|O^T\Lambda O - \Lambda^*\|_{\mathrm{F}} = \frac{1}{\sigma_{\min}^*}\|O^T\Lambda O - \Lambda^*\|_{\mathrm{F}}. \quad (69)$$

Note that $\Lambda = U^T M U$ and $\Lambda^* = U^{*T}M^*U^*$. Thus, we have

$$
\begin{aligned}
\|O^T\Lambda O - \Lambda^*\|_{\mathrm{F}} &= \|O^T U^T M U O - U^{*T}M^*U^*\|_{\mathrm{F}} \\
&\leq \|O^T U^T M U O - O^T U^T M^* U O\|_{\mathrm{F}} + \|O^T U^T M^* U O - U^{*T}M^*U O\|_{\mathrm{F}} \\
&\quad + \|U^{*T}M^*U O - U^{*T}M^*U^*\|_{\mathrm{F}} \\
&\leq \|M - M^*\|_{\mathrm{F}} + 2\|M^*\|_2\|UO - U^*\|_{\mathrm{F}} \\
&\leq \|M - M^*\|_{\mathrm{F}} + 2c\left(\frac{\sigma_{\max}^*}{\sigma_{\min}^*}\right)^2\|M - M^*\|_{\mathrm{F}} \\
&\leq 4c\left(\frac{\sigma_{\max}^*}{\sigma_{\min}^*}\right)^2\|M - M^*\|_{\mathrm{F}}, \quad (70)
\end{aligned}
$$

where the third inequality follows from Lemma B.3. Combing (69) and (70), we have

$$\|\Lambda^{1/2}O - O\Lambda^{*1/2}\|_{\mathrm{F}} \leq \frac{4c(\sigma_{\max}^*)^2}{(\sigma_{\min}^*)^3}\|M - M^*\|_{\mathrm{F}}.$$

$\square$

Now we are ready to prove Lemma 4.2.

*Proof of Lemma 4.2.* Let $\mathcal{O}^{r\times r} := \{O \in \mathbb{R}^{r\times r} \,|\, OO^T = O^T O = I_r\}$. First of all, we show that for any $(B, \beta, O) \in \mathcal{B} \times \mathcal{C} \times \mathcal{O}$, it holds that $\mathbb{P}_{B,\beta}(x, y) = \mathbb{P}_{BO,O^T\beta}(x, y)$. This can be easily seen by the following observation,

$$\mathbb{P}_{BO,O^T\beta} \sim \mathcal{N}\left(0, \begin{bmatrix} BO(BO)^T & BOO^T\beta \\ \beta^T OO^T B^T & (O^T\beta)^T O^T\beta \end{bmatrix}\right) = \mathcal{N}\left(0, \begin{bmatrix} BB^T & B\beta \\ \beta^T B^T & \beta^T\beta \end{bmatrix}\right) \sim \mathbb{P}_{B,\beta}.$$

By Lemma B.3, it holds for some constant $c > 1$ that

$$\|UO - U^*\|_{\mathrm{F}} \leq \frac{c}{(\sigma_{\min}^*)^2}\|BB^T - B^*B^{*T}\|_{\mathrm{F}}. \quad (71)$$

By Lemma B.4, it holds for some constant $c > 1$ that

$$\|\Sigma O - O\Sigma^*\|_{\mathrm{F}} \leq \frac{4c(\sigma_{\max}^*)^2}{(\sigma_{\min}^*)^3}\|BB^T - B^*B^{*T}\|_{\mathrm{F}}. \quad (72)$$

Let $\hat{O} := V^{-1}OV^* \in \mathcal{O}^{r\times r}$. By (71) and (72), we have

$$
\begin{aligned}
\|B\hat{O} - B^*\|_{\mathrm{F}} &= \|U\Sigma O V^* - U^*\Sigma^*V^*\|_{\mathrm{F}} \\
&\leq \|U\Sigma O - U^*\Sigma^*\|_{\mathrm{F}} \\
&\leq \|U\Sigma O - UO\Sigma^*\|_{\mathrm{F}} + \|UO\Sigma^* - U^*\Sigma^*\|_{\mathrm{F}} \\
&\leq \|\Sigma O - O\Sigma^*\|_{\mathrm{F}} + \|UO - U^*\|_{\mathrm{F}}\|\Sigma^*\|_2 \\
&\leq c\cdot\left(\frac{4(\sigma_{\max}^*)^2}{(\sigma_{\min}^*)^3} + \frac{\sigma_{\max}^*}{(\sigma_{\min}^*)^2}\right)\cdot\|BB^T - B^*B^{*T}\|_{\mathrm{F}} \\
&\leq \frac{5c(\sigma_{\max}^*)^2}{(\sigma_{\min}^*)^3}\cdot\|BB^T - B^*B^{*T}\|_{\mathrm{F}} \quad (73)
\end{aligned}
$$

Note that

$$
\begin{aligned}
d_{\mathrm{TV}}\big(\mathbb{P}_{B\hat{O}}(x,z),\mathbb{P}_{B^*}(x,z)\big) &= \int |p_{B\hat{O}}(x\,|\,z) - p_{B^*}(x\,|\,z)|p(z)\,dxdz \\
&= \int d_{\mathrm{TV}}\big(\mathcal{N}(B\hat{O}z,I_d),\mathcal{N}(B^*z,I_d)\big)p(z)\,dz \\
&\le \int \min\{1,\|B\hat{O}z - B^*z\|_2\}p(z)\,dz \\
&\le \min\big\{1,\mathbb{E}[\|B\hat{O}z - B^*z\|_2]\big\},
\end{aligned}
\tag{74}
$$

where the first inequality follows from Lemma B.1. We can show that

$$
\begin{aligned}
\mathbb{E}[\|B\hat{O}z - B^*z\|_2] &\le \Big(\mathbb{E}\big[\|B\hat{O}z - B^*z\|_2^2\big]\Big)^{1/2} \\
&= \Big(\mathbb{E}\big[z^T(B\hat{O} - B^*)^T(B\hat{O} - B^*)z\big]\Big)^{1/2} \\
&= \Big(\mathbb{E}\big[\mathrm{Tr}\big((B\hat{O} - B^*)^T(B\hat{O} - B^*)zz^T\big)\big]\Big)^{1/2} \\
&= \Big(\mathrm{Tr}\big((B\hat{O} - B^*)^T(B\hat{O} - B^*)\big)\Big)^{1/2} \\
&= \|B\hat{O} - B^*\|_{\mathrm{F}}.
\end{aligned}
\tag{75}
$$

By (73), (74) and (75), it holds that

$$
\begin{aligned}
d_{\mathrm{TV}}&\big(\mathbb{P}_{B\hat{O}}(x,z),\mathbb{P}_{B^*}(x,z)\big) \\
&\le \min\big\{1,\|B\hat{O} - B^*\|_{\mathrm{F}}\big\} \\
&\le \min\left\{1,\frac{5c(\sigma_{\max}^*)^2}{(\sigma_{\min}^*)^3}\cdot\|BB^T - B^*B^{*T}\|_{\mathrm{F}}\right\} \\
&\le \frac{5c(\sigma_{\max}^*)^2}{(\sigma_{\min}^*)^3}\cdot\big((\sigma_{\max}^*)^2 + 1\big)\cdot\min\left\{1,\frac{\|BB^T - B^*B^{*T}\|_{\mathrm{F}}}{(\sigma_{\max}^*)^2 + 1}\right\},
\end{aligned}
\tag{76}
$$

where the last inequality follows from $c > 1$ and

$$
\frac{(\sigma_{\max}^*)^2 + 1}{\sigma_{\min}^*} \ge \frac{2\sigma_{\max}^*}{\sigma_{\min}^*} > 1.
$$

By Lemma B.2, we have

$$
\begin{aligned}
d_{\mathrm{TV}}&(p_B(x),p_{B^*}(x)) \\
&\ge \frac{1}{100}\min\big\{1,\|(B^*B^{*T} + I_d)^{-1/2}(BB^T - B^*B^{*T})(B^*B^{*T} + I_d)^{-1/2}\|_{\mathrm{F}}\big\}.
\end{aligned}
\tag{77}
$$

Note that

$$
\begin{aligned}
\|(B^*B^{*T} &+ I_d)^{-1/2}(BB^T - B^*B^{*T})(B^*B^{*T} + I_d)^{-1/2}\|_{\mathrm{F}} \\
&\ge \frac{\|BB^T - B^*B^{*T}\|_{\mathrm{F}}}{\|B^*B^{*T} + I_d\|_2} \ge \frac{\|BB^T - B^*B^{*T}\|_{\mathrm{F}}}{(\sigma_{\max}^*)^2 + 1}.
\end{aligned}
\tag{78}
$$

Thus, by (77) and (78), it holds that

$$
d_{\mathrm{TV}}(p_B(x),p_{B^*}(x)) \ge \frac{1}{100}\min\left\{1,\frac{\|BB^T - B^*B^{*T}\|_{\mathrm{F}}}{(\sigma_{\max}^*)^2 + 1}\right\}
\tag{79}
$$

Finally, by (76) and (79), we have

$$
\begin{aligned}
d_{\mathrm{TV}}\big(\mathbb{P}_{B\hat{O}}(x,z),\mathbb{P}_{B^*}(x,z)\big) &\le \frac{500c(\sigma_{\max}^*)^2\big((\sigma_{\max}^*)^2 + 1\big)}{(\sigma_{\min}^*)^3}\cdot d_{\mathrm{TV}}(p_B(x),p_{B^*}(x)) \\
&\le \frac{500c(\sigma_{\max}^* + 1)^4}{(\sigma_{\min}^*)^3}\cdot d_{\mathrm{TV}}(p_B(x),p_{B^*}(x)).
\end{aligned}
$$

$\square$

## B.2 BRACKETING NUMBER

By an application of $\epsilon$-discretization technique, we upper bound the bracketing number of $\mathcal{P}(\mathcal{B})$ as follows.

**Lemma B.5.** *Let* $\mathcal{P}_\mathcal{X}(\mathcal{B}) := \{\mathcal{N}(0, BB^T + I_d) \,|\, B \in \mathcal{B}\}$, *where* $\mathcal{B} = \{B \in \mathbb{R}^{d \times r} \,|\, \|B\|_2 \le D\}$ *for some* $D > 0$. *Then the entropy can be bounded as follows,*

$$\log N_{[\,]}(\mathcal{P}_\mathcal{X}(\mathcal{B}), 1/m) \le 4dr \log\big(24mdr(D^2 + 1)\big).$$

*Proof of Lemma B.5.* We consider a set of Gaussian distribution

$$\mathcal{P}_\mathcal{X}(\mathcal{B}) := \left\{ p_\Sigma(x) = \frac{1}{\sqrt{(2\pi)^d |\Sigma|}} e^{-\frac{1}{2} x^T \Sigma^{-1} x} \,\middle|\, \Sigma = BB^T + I_d, B \in \mathcal{B} \right\},$$

where $\mathcal{B} = \{B \in \mathbb{R}^{d \times r} \,|\, \|B\|_2 \le D\}$. Note that

$$\lambda_{\max}(\Sigma^{-1}) = \big(\lambda_{\min}(\Sigma)\big)^{-1} = 1, \ \lambda_{\min}(\Sigma^{-1}) = \big(\lambda_{\max}(\Sigma)\big)^{-1} \ge \frac{1}{D^2 + 1}. \tag{80}$$

Here we denote by $\lambda_{\max}(\Sigma^{-1})$ and $\lambda_{\min}(\Sigma^{-1})$ the largest eigenvalue and the smallest eigenvalue of $\Sigma^{-1}$, respectively. Our goal is to find a $1/m$-bracket $\mathcal{N}_{[\,]}(\mathcal{P}_\mathcal{X}(\mathcal{B}), 1/m)$ of $\mathcal{P}_\mathcal{X}(\mathcal{B})$. In other words, for any $p_\Sigma(x) \in \mathcal{P}_\mathcal{X}(\mathcal{B})$, we need to define $\bar{p}_\Sigma(x) \in \mathcal{N}_{[\,]}(\mathcal{P}_\mathcal{X}(\mathcal{B}), 1/m)$ such that

- $\bar{p}_\Sigma(x) \ge p_\Sigma(x), \ \forall x \in \mathbb{R}^d$
- $\int |\bar{p}_\Sigma(x) - p_\Sigma(x)| \, dx \le 1/m$.

Note that $\text{rank}(BB^T) = r < d$ and $\Sigma = BB^T + I_d$. Thus, the eigendecomposition of $\Sigma^{-1}$ has the following form

$$\Sigma^{-1} = V \begin{bmatrix} \lambda_1 & & & & & \\ & \ddots & & & & \\ & & \lambda_r & & & \\ & & & 1 & & \\ & & & & \ddots & \\ & & & & & 1 \end{bmatrix} V^T = U \begin{bmatrix} \lambda_1 - 1 & & \\ & \ddots & \\ & & \lambda_r - 1 \end{bmatrix} U^T + I_d, \tag{81}$$

where $VV^T = V^TV = I_d$ and $U \in \mathbb{R}^{d \times r}$ is the first $r$ columns of $V$. For notation simplicity, we denote

$$\Lambda := \begin{bmatrix} \lambda_1 - 1 & & \\ & \ddots & \\ & & \lambda_r - 1 \end{bmatrix}.$$

Thus, we have $\Sigma^{-1} = U\Lambda U^T + I_d$. For some fixed $0 < \epsilon \le (D^2 + 1)^{-1}/2$ (which we will choose later), if $\lambda_i \in [k\epsilon, (k + 1)\epsilon)$ for some $k \in \mathbb{Z}$, we define $\bar{\lambda}_i := (k - 1)\epsilon$. Note that $\lambda_i \ge \lambda_{\min}(\Sigma^{-1}) \ge (D^2 + 1)^{-1}$. Thus, it holds that $k \ge 2$ and $\bar{\lambda}_i = (k - 1)\epsilon \ge \epsilon > 0$. Moreover, we have $\epsilon \le \lambda_i - \bar{\lambda}_i \le 2\epsilon$. We define

$$\bar{\Lambda} := \begin{bmatrix} \bar{\lambda}_1 - 1 & & \\ & \ddots & \\ & & \bar{\lambda}_r - 1 \end{bmatrix}.$$

For the matrix $U = (u_{i,j}) \in \mathbb{R}^{d \times r}$, if $u_{i,j} \in [\frac{k\epsilon}{3\sqrt{dr}}, \frac{(k+1)\epsilon}{3\sqrt{dr}})$ for some $k \in \mathbb{Z}$, we define $\bar{u}_{i,j} := \frac{k\epsilon}{3\sqrt{dr}}$ and $\bar{U} := (\bar{u}_{i,j}) \in \mathbb{R}^{d \times r}$. It then holds that

$$\|U - \bar{U}\|_2 \le \|U - \bar{U}\|_F = \sqrt{\sum_{i,j} |u_{i,j} - \bar{u}_{i,j}|^2} \le \sqrt{dr} \cdot \frac{\epsilon}{3\sqrt{dr}} = \frac{\epsilon}{3}. \tag{82}$$

We define

$$\overline{\Sigma^{-1}} := \bar{U}\bar{\Lambda}\bar{U}^T + I_d. \tag{83}$$

Note that $(D^2 + 1)^{-1} \le \lambda_i \le 1$ and $|u_{i,j}| \le 1$. Thus, we totally have

$$\left(\frac{1 - (D^2 + 1)^{-1}}{\epsilon}\right)^r \cdot \left(\frac{6\sqrt{dr}}{\epsilon}\right)^{dr} = \left(\frac{D^2}{(D^2 + 1)\epsilon}\right)^r \cdot \left(\frac{6\sqrt{dr}}{\epsilon}\right)^{dr} \tag{84}$$

many $\bar{\Sigma}^{-1}$. Note that for any $\|x\|_2 = 1$, we have

$$
\begin{aligned}
x^T(\Sigma^{-1} - \overline{\Sigma^{-1}})x &= x^T(U^T\Lambda U - \bar{U}\bar{\Lambda}\bar{U}^T)x \\
&= x^T U^T(\Lambda - \bar{\Lambda})Ux + x^T(U - \bar{U})^T\bar{\Lambda}(U + \bar{U})x \\
&\ge \lambda_{\min}(\Lambda - \bar{\Lambda}) - \|(U - \bar{U})^T\bar{\Lambda}(U + \bar{U})\|_2 \\
&\ge \lambda_{\min}(\Lambda - \bar{\Lambda}) - \|U - \bar{U}\|_2 \cdot \|\bar{\Lambda}(U + \bar{U})\|_2 \\
&\ge \epsilon - 3\left(2\epsilon + \frac{D^2}{D^2 + 1}\right)\|U - \bar{U}\|_2 \\
&\ge \epsilon - 3\left(2\epsilon + \frac{D^2}{D^2 + 1}\right) \cdot \frac{\epsilon}{3} \ge 0,
\end{aligned}
$$

where the third inequality follows from

$$\|\bar{\Lambda}(U + \bar{U})\|_2 \le \|\bar{\Lambda}\|_2\|U + \bar{U}\|_2 \le \left(2\epsilon + 1 - \frac{1}{D^2 + 1}\right) \cdot \left(2 + \frac{\epsilon}{3}\right) \le 3\left(2\epsilon + \frac{D^2}{D^2 + 1}\right).$$

and the last inequality follows from our assumption $\epsilon \le (D^2 + 1)^{-1}/2$. Thus, for any $x \in \mathbb{R}^d$, it holds that

$$x^T(\Sigma^{-1} - \overline{\Sigma^{-1}})x \ge 0. \tag{85}$$

We consider $\bar{p}_\Sigma(x)$ of the following form

$$\bar{p}_\Sigma(x) = c\sqrt{\frac{|\overline{\Sigma^{-1}}|}{(2\pi)^d}}e^{-\frac{1}{2}x^T\overline{\Sigma^{-1}}x}.$$

By (85), we have: $\bar{p}_\Sigma(x) \ge p_\Sigma(x)$ holds for any $x \in \mathbb{R}^d$ if and only if

$$c \ge \sqrt{\frac{|\Sigma^{-1}|}{|\overline{\Sigma^{-1}}|}} = \sqrt{\frac{\lambda_1 \dots \lambda_r}{\bar{\lambda}_1 \dots \bar{\lambda}_r}}.$$

Note that

$$\frac{\lambda_i}{\bar{\lambda}_i} \le \frac{(k + 1)\epsilon}{(k - 1)\epsilon} = 1 + \frac{2}{k - 1} \le 1 + \frac{4}{k} \le 1 + 4(D^2 + 1)\epsilon,$$

where the second inequality follows from $k \ge 2$ and the last inequality follows from $k\epsilon \ge (D^2 + \sigma^2)^{-1}$. We then obtain that

$$\sqrt{\frac{\lambda_1 \dots \lambda_r}{\bar{\lambda}_1 \dots \bar{\lambda}_r}} \le \left(1 + 4(D^2 + 1)\epsilon\right)^{r/2}.$$

Let $c = (1 + 4(D^2 + 1)\epsilon)^{r/2}$. It then holds that

$$c \ge \sqrt{\frac{\lambda_1 \dots \lambda_r}{\bar{\lambda}_1 \dots \bar{\lambda}_r}},$$

which implies $\bar{p}_\Sigma(x) \ge p_\Sigma(x)$ holds for any $x \in \mathbb{R}^d$. Note that

$$\int |\bar{p}_\Sigma(x) - p_\Sigma(x)| \, dx = c - 1 = (1 + 4(D^2 + 1)\epsilon)^{r/2} - 1 \le 4(D^2 + 1)\epsilon r,$$

where the last inequality follow from $(1+x)^{r/2} \le 1 + rx$ for $x \le r^{-1}$. Let

$$\epsilon = \frac{1}{4(D^2+1)mr}. \tag{86}$$

We have

$$\int |\bar{p}_\Sigma(x) - p_\Sigma(x)|\, dx \le 4(D^2+1)\epsilon r = \frac{1}{m}.$$

By (84) and (86), we show that

$$N_{[\,]}(\mathcal{P}_{\mathcal{X}}(\mathcal{B}), 1/m) \le (4rmD^2)^r \cdot \left(24rm(D^2+1)\sqrt{dr}\right)^{dr},$$

which implies

$$\log N_{[\,]}(\mathcal{P}_{\mathcal{X}}(\mathcal{B}), 1/m) \le 4dr \log\left(24mdr(D^2+1)\right).$$

$\square$

## B.3 RADEMACHER COMPLEXITY

Note that for fixed $B$ the prediction function class

$$\mathcal{G}_{B,\mathcal{C}} := \left\{ g_{B,\beta}(x) = \beta^T B^T (BB^T + \sigma^2 I_d)^{-1} x \,\middle|\, \beta \in \mathcal{C} \right\}$$

belongs to a linear hypothesis class. For a linear hypothesis class $\mathcal{H}$, we can bound its empirical Rademacher complexity as follows.

**Lemma B.6.** *For a linear hypothesis class $\mathcal{H} = \{h_\beta(x) = \beta^T x \,|\, \beta \in \mathbb{R}^r, \|\beta\|_2 \le D\}$, where $x \in \mathbb{R}^r$ and $\|x\|_2 \le X$, the empirical Rademacher complexity can be bounded as follows,*

$$\hat{R}_n(\mathcal{H}) \le \frac{2DX}{\sqrt{n}}.$$

*Proof of Lemma B.6.* Note that

$$\hat{R}_n(\mathcal{H}) = \frac{2}{n}\mathbb{E}_{\sigma_i}\left[\sup_{\|\beta\|_2 \le D} \sum_{i=1}^n \sigma_i \cdot \beta^T x_i\right] = \frac{2}{n}\mathbb{E}_{\sigma_i}\left[\sup_{\|\beta\|_2 \le D} \beta^T\left(\sum_{i=1}^n \sigma_i x_i\right)\right]$$

$$\le \frac{2}{n}\mathbb{E}_{\sigma_i}\left[\sup_{\|\beta\|_2 \le D} \|\beta\|_2 \left\|\sum_{i=1}^n \sigma_i x_i\right\|_2\right] \le \frac{2D}{n}\mathbb{E}_{\sigma_i}\left[\sqrt{\sum_{i,j} \sigma_i \sigma_j x_i^T x_j}\right].$$

By Jensen's inequality, we then have

$$\hat{R}_n(\mathcal{H}) \le \frac{2D}{n}\mathbb{E}_{\sigma_i}\left[\sqrt{\sum_{i,j} \sigma_i \sigma_j x_i^T x_j}\right] \le \frac{2D}{n}\sqrt{E_{\sigma_i}\left[\sum_{i,j} \sigma_i \sigma_j x_i^T x_j\right]} = \frac{2D}{n}\sqrt{\sum_{i=1}^n \|x_i\|^2} \le \frac{2DX}{\sqrt{n}}.$$

$\square$

## B.4 PROOFS FOR THEOREM 4.3

In this section, we verify the utility of Algorithm 1 by proving Theorem 4.3. Recall that the truncated squared loss is defined as

$$\tilde{\ell}(x,y) := (y-x)^2 \mathbb{I}_{\{(y-x)^2 \le L\}} + L \cdot \mathbb{I}_{\{(y-x)^2 > L\}}, \tag{87}$$

which is $L-$bounded and $2\sqrt{L}-$Lipschitz w.r.t. the first argument. Before proving Theorem 4.3, we need to state some core lemmas. Recall the definition of $g_{B,\beta}(x)$:

$$g_{B,\beta}(x) := \arg\min_g \mathbb{E}_{B,\beta}[\ell(g(x),y)]. \tag{88}$$

Since $\ell$ is the squared loss, it's obvious that

$$g_{B,\beta}(x) := \arg\min_g \mathbb{E}_{B,\beta}[\ell(g(x),y)] = \mathbb{E}_{\mathbb{P}_{B,\beta}(x,y)}[y \mid x] = \beta^T B^T (BB^T + I_d)^{-1} x. \tag{89}$$

The next lemma shows that the optimal predictor under the squared loss $\ell$ and the truncated squared loss $\tilde{\ell}$ stays the same.

**Lemma B.7.** *We denote by $\tilde{g}_{B,\beta}$ the optimal predictor under truncated squared loss, i.e.,*

$$\tilde{g}_{B,\beta} \leftarrow \arg\min_{g} \mathbb{E}_{B,\beta}[\tilde{\ell}(g(x), y)]. \tag{90}$$

*It then holds that*

$$\tilde{g}_{B,\beta}(x) = \mathbb{E}_{\mathbb{P}_{B,\beta}(x,y)}[y \mid x] = g_{B,\beta}(x). \tag{91}$$

*Proof of Lemma B.7.* Notice that, the distribution (under parameter $B, \beta$) of $y$ given $x$ is a Gaussian distribution with mean $\mu = \mathbb{E}_{P_{B,\beta}(x,y)}[y \mid x]$ and variance $v^2$ (which is of no importance). We define function $f$ as

$$
\begin{aligned}
f(a) &:= \mathbb{E}_{B,\beta}[\tilde{\ell}(a, y) \mid x] \\
&= \int_{a-\sqrt{L}}^{a+\sqrt{L}} (y-a)^2 \frac{1}{v\sqrt{2\pi}} e^{-\frac{(y-\mu)^2}{2v^2}} \mathrm{d}y + \int_{a+\sqrt{L}}^{+\infty} L \frac{1}{v\sqrt{2\pi}} e^{-\frac{(y-\mu)^2}{2v^2}} \mathrm{d}y \\
&\quad + \int_{-\infty}^{a-\sqrt{L}} L \frac{1}{v\sqrt{2\pi}} e^{-\frac{(y-\mu)^2}{2v^2}} \mathrm{d}y.
\end{aligned} \tag{92}
$$

Then, it holds that

$$
\begin{aligned}
f'(a) &= \frac{L}{v\sqrt{2\pi}} e^{-\frac{(a-\mu+\sqrt{L})^2}{2v^2}} - \frac{L}{v\sqrt{2\pi}} e^{-\frac{(a-\mu-\sqrt{L})^2}{2v^2}} + \int_{a-\sqrt{L}}^{a+\sqrt{L}} 2(a-y) \frac{1}{v\sqrt{2\pi}} e^{-\frac{(y-\mu)^2}{2v^2}} \mathrm{d}y \\
&\quad - \frac{L}{v\sqrt{2\pi}} e^{-\frac{(a-\mu+\sqrt{L})^2}{2v^2}} + \frac{L}{v\sqrt{2\pi}} e^{-\frac{(a-\mu-\sqrt{L})^2}{2v^2}} \\
&= \int_{a-\sqrt{L}}^{a+\sqrt{L}} 2(a-y) \frac{1}{v\sqrt{2\pi}} e^{-\frac{(y-\mu)^2}{2v^2}} \mathrm{d}y \\
&= \int_{a-\sqrt{L}}^{a} 2(a-y) \frac{1}{v\sqrt{2\pi}} e^{-\frac{(y-\mu)^2}{2v^2}} \mathrm{d}y + \int_{a}^{a+\sqrt{L}} 2(a-y) \frac{1}{v\sqrt{2\pi}} e^{-\frac{(y-\mu)^2}{2v^2}} \mathrm{d}y \\
&= \int_{0}^{\sqrt{L}} 2z \frac{1}{v\sqrt{2\pi}} e^{-\frac{(a-z-\mu)^2}{2v^2}} \mathrm{d}z - \int_{0}^{\sqrt{L}} 2z \frac{1}{v\sqrt{2\pi}} e^{-\frac{(a+z-\mu)^2}{2v^2}} \mathrm{d}z \\
&= \int_{0}^{\sqrt{L}} \frac{2z}{v\sqrt{2\pi}} (e^{-\frac{(a-z-\mu)^2}{2v^2}} - e^{-\frac{(a+z-\mu)^2}{2v^2}}) \mathrm{d}z.
\end{aligned} \tag{93}
$$

Notice that for $z \in [0, \sqrt{L}]$,

$$e^{-\frac{(a-z-\mu)^2}{2v^2}} - e^{-\frac{(a+z-\mu)^2}{2v^2}} > 0 \text{ when } a > \mu, \tag{94}$$

$$e^{-\frac{(a-z-\mu)^2}{2v^2}} - e^{-\frac{(a+z-\mu)^2}{2v^2}} < 0 \text{ when } a < \mu. \tag{95}$$

Therefore, we have $f'(a) < 0$ when $a < \mu$, $f'(a) > 0$ when $a > \mu$, which implies that $a = \mu$ is the unique minimizer of $f(a)$, i.e.,

$$\tilde{g}_{B,\beta}(x) = \mathbb{E}_{\mathbb{P}_{B,\beta}(x,y)}[y \mid x] = g_{B,\beta}(x). \tag{96}$$

$\square$

The following lemma shows that the truncation has no significant influence on the excess risk.

**Lemma B.8.** *There exist $c_2 = (D^2 + 1)^3$, such that*

$$\mathrm{Error}_{\ell}(\hat{B}, \hat{\beta}) \leq \mathbb{E}_{B^*,\beta^*}[\tilde{\ell}(g_{\hat{B},\hat{\beta}}(x), y)] - \mathbb{E}_{B^*,\beta^*}[\tilde{\ell}(g_{B^*,\beta^*}(x), y)] + \sqrt{\frac{2Lc_2}{\pi}} e^{-\frac{L}{2c_2}}. \tag{97}$$

*Proof of Lemma B.8.*

$$\text{Error}_\ell(\hat{B}, \hat{\beta}) = \mathbb{E}_{B^*,\beta^*}[\ell(g_{\hat{B},\hat{\beta}}(x), y)] - \mathbb{E}_{B^*,\beta^*}[\ell(g_{B^*,\beta^*}(x), y)]$$

$$= \mathbb{E}_{B^*,\beta^*}[\ell(g_{\hat{B},\hat{\beta}}(x), y)] - \mathbb{E}_{B^*,\beta^*}[\tilde{\ell}(g_{\hat{B},\hat{\beta}}(x), y)]$$

$$+ \mathbb{E}_{B^*,\beta^*}[\tilde{\ell}(g_{\hat{B},\hat{\beta}}(x), y)] - \mathbb{E}_{B^*,\beta^*}[\tilde{\ell}(g_{B^*,\beta^*}(x), y)]$$

$$+ \mathbb{E}_{B^*,\beta^*}[\tilde{\ell}(g_{B^*,\beta^*}(x), y)] - \mathbb{E}_{B^*,\beta^*}[\ell(g_{B^*,\beta^*}(x), y)] \quad (\leq 0 \text{ since } \tilde{\ell} \leq \ell)$$

$$\leq \sup_{B,\beta}\{\mathbb{E}_{B^*,\beta^*}[\ell(g_{B,\beta}(x), y)] - \mathbb{E}_{B^*,\beta^*}[\tilde{\ell}(g_{B,\beta}(x), y)]\}$$

$$+ \mathbb{E}_{B^*,\beta^*}[\tilde{\ell}(g_{\hat{B},\hat{\beta}}(x), y)] - \mathbb{E}_{B^*,\beta^*}[\tilde{\ell}(g_{B^*,\beta^*}(x), y)] \tag{98}$$

For the first term, we have

$$\sup_{B,\beta}\{\mathbb{E}_{B^*,\beta^*}[\ell(g_{B,\beta}(x), y)] - \mathbb{E}_{B^*,\beta^*}[\tilde{\ell}(g_{B,\beta}(x), y)]\}$$

$$= \sup_{B,\beta}\{\mathbb{E}_{B^*,\beta^*}((g_{B,\beta}(x) - y)^2 - L)\mathbb{1}_{\{(g_{B,\beta}(x)-y)^2 \geq L\}}\}. \tag{99}$$

Notice that

$$g_{B,\beta}(x) - y = \beta^T B^T (BB^T + I_d)^{-1} x - y \sim \mathcal{N}(0, \lambda^2), \tag{100}$$

where

$$\lambda^2 = Var_{B^*,\beta^*}[g_{B,\beta}(x) - y]$$

$$= \mathbb{E}_{B^*,\beta^*}(\beta^T B^T (BB^T + I_d)^{-1} x - y)^2$$

$$= \epsilon^2 + \beta^T B^T (BB^T + I_d)^{-1} (B^* B^{*T} + I_d)(BB^T + I_d)^{-1} B\beta$$

$$+ \beta^{*T}\beta^* - 2\beta^T B^T (BB^T + I_d)^{-1} B^*\beta^*$$

$$\leq \epsilon^2 + \beta^{*T}\beta^* + \|(BB^T + I_d)^{-1}\|_2^2 \cdot \|B^* B^{*T} + I_d\|_2 \cdot \|B\beta\|_2^2$$

$$+ 2\|(BB^T + I_d)^{-1}\|_2 \cdot \|B^*\beta^*\|_2 \cdot \|B\beta\|_2$$

$$\leq \epsilon^2 + \beta^{*T}\beta^* + D^4\|B^* B^{*T} + I_d\|_2 + 2D^2\|B^*\beta^*\|_2$$

$$\leq 1 + D^2 + D^4(D^2 + 1) + 2D^4$$

$$\leq c_2. \tag{101}$$

Therefore

$$\sup_{B,\beta}\{\mathbb{E}_{B^*,\beta^*}((g_{B,\beta}(x) - y)^2 - L)\mathbb{1}_{\{(g_{B,\beta}(x)-y)^2 \geq L\}}\}$$

$$= \sup_\lambda 2 \int_{\sqrt{L}}^{+\infty} \frac{1}{\lambda\sqrt{2\pi}}(x^2 - L)e^{-\frac{x^2}{2\lambda^2}} \, dx$$

$$= 2\sup_\lambda \left\{ -\frac{\lambda}{\sqrt{2\pi}}xe^{-\frac{x^2}{2\lambda^2}}\Big|_{\sqrt{L}}^{+\infty} + (\lambda^2 - L)\int_{\sqrt{L}}^{+\infty} \frac{1}{\lambda\sqrt{2\pi}}e^{-\frac{x^2}{2\lambda^2}} \, dx \right\}$$

$$= 2\sup_\lambda \left\{ \sqrt{\frac{L}{2\pi}}\lambda e^{-\frac{L}{2\lambda^2}} + (\lambda^2 - L)\int_{\sqrt{L}}^{+\infty} \frac{1}{\lambda\sqrt{2\pi}}e^{-\frac{x^2}{2\lambda^2}} \, dx \right\}$$

$$\leq 2\sup_\lambda \left\{ \sqrt{\frac{L}{2\pi}}\lambda e^{-\frac{L}{2\lambda^2}} \right\} \quad (\text{since } L \geq c_2 \geq \lambda^2)$$

$$= \sqrt{\frac{2Lc_2}{\pi}}e^{-\frac{L}{2c_2}}. \tag{102}$$

The last equation holds since $\lambda e^{-\frac{L}{2\lambda^2}}$ monotone increases w.r.t. $\lambda$, and $\lambda \leq \sqrt{c_1}$. Combining (98), (99) and (102), we finish the proof. $\square$

Now we are ready to prove Theorem 4.3.

*Proof of Theorem 4.3.* Note that $\tilde{l}$ is $L-$bounded. By Lemma B.7, we can apply Theorem 3.4 to $\tilde{l}$, which gives

$$\mathbb{E}_{B^*,\beta^*}[\tilde{\ell}(g_{\hat{B},\hat{\beta}}(x), y)] - \mathbb{E}_{B^*,\beta^*}[\tilde{\ell}(g_{B^*,\beta^*}(x), y)]$$

$$\leq 2 \max_{B \in \mathcal{B}} R_n \left( \tilde{\ell} \circ \mathcal{G}_{B,\mathcal{C}} \right) + L \cdot \sqrt{\frac{2}{n} \log \frac{4}{\delta}} + 12\kappa L \cdot \sqrt{\frac{1}{m} \log \frac{2N_{[\,]}(\mathcal{P}_{\mathcal{X}}(\mathcal{B}), 1/m)}{\delta}}. \tag{103}$$

Here $\kappa = c_1 (\sigma_{max}^* + 1)^4 / \sigma_{min}^{*3}$ is the transferability defined in Lemma 4.2.

By Lemma B.5, we have

$$\log N_{[\,]}(\mathcal{P}(\mathcal{B}), 1/m) \leq 4dr \log(24mdr(D^2 + 1)). \tag{104}$$

Since $\tilde{l}$ is $2\sqrt{L}-$Lipschitz w.r.t. the first argument, the contraction principle (Theoerem 4.12 in Ledoux & Talagrand (2013)) gives

$$R_n \left( \tilde{\ell} \circ \mathcal{G}_{B,\mathcal{C}} \right) \leq 2\sqrt{L} R_n \left( \mathcal{G}_{B,\mathcal{C}} \right). \tag{105}$$

Therefore it remains to bound $R_n \left( \mathcal{G}_{B,\mathcal{C}} \right)$. By Lemma B.6, for fixed $B$,

$$R_n \left( \mathcal{G}_{B,\mathcal{C}} \right) = \mathbb{E}_{\{x_j\}_{j=1}^n} \mathbb{E}_{\{\sigma_j\}_{j=1}^n} [\sup_\beta \frac{2}{n} \sum_{j=1}^n \sigma_j g_{B,\beta}(x_j)]$$

$$= \mathbb{E}_{\{x_j\}_{j=1}^n} \mathbb{E}_{\{\sigma_j\}_{j=1}^n} [\sup_\beta \frac{2}{n} \sum_{j=1}^n \sigma_j \beta^T B^T (BB^T + I_d)^{-1} x_j]$$

$$\leq \mathbb{E}_{\{x_j\}_{j=1}^n} [\frac{2D}{\sqrt{n}} \sup_j \|B^T (BB^T + I_d)^{-1} x_j\|_2] \quad \text{(By Lemma B.6, since } \|\beta\|_2 \leq D)$$

$$= \frac{2D}{\sqrt{n}} \mathbb{E}_{\{x_j\}_{j=1}^n} [\sup_j \|B^T (BB^T + I_d)^{-1} x_j\|_2]. \tag{106}$$

Note that $x_j \sim \mathcal{N}(0, B^* B^{*T} + I_d)$. Therefore $B^T (BB^T + I_d)^{-1} x_j \sim \mathcal{N}(0, \Sigma)$, where

$$\Sigma := B^T (BB^T + I_d)^{-1} (B^* B^{*T} + I_d)(BB^T + I_d)^{-1} B. \tag{107}$$

Thus, we have

$$\Sigma^{-\frac{1}{2}} B^T (BB^T + I_d)^{-1} x_j \sim \mathcal{N}(0, I_r). \tag{108}$$

Let $u_j := \Sigma^{-\frac{1}{2}} B^T (BB^T + I_d)^{-1} x_j$, then

$$\mathbb{E}_{\{x_j\}_{j=1}^n} [\sup_j \|B^T (BB^T + I_d)^{-1} x_j\|_2]$$

$$= \mathbb{E}_{\{x_j\}_{j=1}^n} [\sup_j \|\Sigma^{\frac{1}{2}} u_j\|_2]$$

$$\leq \mathbb{E}_{\{x_j\}_{j=1}^n} [\sup_j \|\Sigma^{\frac{1}{2}}\|_2 \|u_j\|_2]$$

$$\leq \sup \|\Sigma^{\frac{1}{2}}\|_2 \mathbb{E}_{\{x_j\}_{j=1}^n} [\sup_j \|u_j\|_2]. \tag{109}$$

By the Theorem 3.1.1 in Vershynin (2018), $\|u_j\| - \sqrt{r}$ is $c_4-$subGaussian for some absolute constant $c_4$. Therefore, for any $t > 0$,

$$e^{\mathbb{E}[t \sup_j \|u_j\|_2]} \leq \mathbb{E}[e^{t \sup_j \|u_j\|_2}] \quad \text{(by Jensen's inequality)}$$

$$\leq \sum_{j=1}^n \mathbb{E}[e^{t\|u_j\|_2}]$$

$$= \sum_{j=1}^n \mathbb{E}[e^{t\|u_j\|_2 - \sqrt{r}}] e^{t\sqrt{r}}$$

$$\leq \sum_{j=1}^n e^{\frac{t^2}{2} c_4} e^{t\sqrt{r}}$$

$$= n e^{t\sqrt{r} + \frac{t^2}{2} c_4}. \tag{110}$$

Taking log on both sides, we have

$$\mathbb{E}[\sup_j \|u_j\|_2] \leq \frac{\log n}{t} + \sqrt{r} + \frac{t}{2}c_4, \tag{111}$$

which holds for any $t > 0$. Take $t = \sqrt{\frac{2 \log n}{c_4}}$, we get

$$\mathbb{E}[\sup_j \|u_j\|_2] \leq \sqrt{2c_4 \log n} + \sqrt{r}. \tag{112}$$

Note that

$$\begin{aligned}
\|\Sigma\|_2 &= \|B^T(BB^T + I_d)^{-1}(B^*B^{*T} + I_d)(BB^T + I_d)^{-1}B\|_2 \\
&\leq \|B\|_2^2 \cdot \|(BB^T + I_d)^{-1}\|_2^2 \cdot \|B^*B^{*T} + I_d\| \\
&\leq (D^2 + 1)^2,
\end{aligned} \tag{113}$$

i.e., $\sup \|\Sigma^{\frac{1}{2}}\|_2 \leq (D^2 + 1)$. Combining (106), (109), (112) and (113), we have

$$\begin{aligned}
R_n(\mathcal{G}_{\phi,\Psi}) &\leq \frac{2D}{\sqrt{n}} \mathbb{E}_{\{x_j\}_{j=1}^n}[\sup_j \|B^T(BB^T + I_d)^{-1}x_j\|_2] \\
&\leq \frac{2D}{\sqrt{n}} \sup \|\Sigma^{\frac{1}{2}}\|_2 \mathbb{E}_{\{x_j\}_{j=1}^n}[\sup_j \|u_j\|_2] \\
&\leq \frac{2D}{\sqrt{n}}(D^2 + 1)(\sqrt{2c_4 \log n} + \sqrt{r}),
\end{aligned} \tag{114}$$

which implies

$$\max_{\phi \in \Phi} R_n\left(\tilde{\ell} \circ \mathcal{G}_{\phi,\Psi}\right) \leq 2\sqrt{L} \max_{\phi in \Phi} R_n(\mathcal{G}_{\phi,\Psi}) \leq 2\sqrt{L}\frac{2D}{\sqrt{n}}(D^2 + 1)(\sqrt{2c_4 \log n} + \sqrt{r}) \tag{115}$$

We are now ready to bound the excess risk. By Lemma B.8, we have

$$\begin{aligned}
\text{Error}_\ell(\hat{B}, \hat{\beta}) &\leq \mathbb{E}_{B^*,\beta^*}[\tilde{\ell}(g_{\hat{B},\hat{\beta}}(x), y)] - \mathbb{E}_{B^*,\beta^*}[\tilde{\ell}(g_{B^*,\beta^*}(x), y)] + \sqrt{\frac{2Lc_2}{\pi}}e^{-\frac{L}{2c_2}} \\
&\leq 2\max_{\phi \in \Phi} R_n\left(\tilde{\ell} \circ \mathcal{G}_{\phi,\Psi}\right) + L \cdot \sqrt{\frac{2}{n}\log\frac{4}{\delta}} \\
&\quad + 12\kappa L \cdot \sqrt{\frac{1}{m}\log\frac{2N_{[\,]}(\mathcal{P}_\mathcal{X}(\mathcal{B}), 1/m)}{\delta}} + \sqrt{\frac{2Lc_2}{\pi}}e^{-\frac{L}{2c_2}} \\
&\leq 4\sqrt{L}\frac{2D}{\sqrt{n}}(D^2 + 1)(\sqrt{2c_4 \log n} + \sqrt{r}) + L \cdot \sqrt{\frac{2}{n}\log\frac{4}{\delta}} \\
&\quad + 12\kappa L\sqrt{\frac{1}{m}(4dr\log(24mdr(D^2 + 1)) + \log(2/\delta))} + \sqrt{\frac{2Lc_2}{\pi}}e^{-\frac{L}{2c_2}}, \tag{116}
\end{aligned}$$

where the second inequality follows from (103) and the last inequality follows from (104), (115). Here $c_4$ is an absolute constant. Note that $c_2 = (D^2 + 1)^3$ and $L = c_2 \log n$. Thus, we have

$$\begin{aligned}
\text{Error}_\ell(\hat{B}, \hat{\beta}) &\leq 8\sqrt{2c_4}L\sqrt{\frac{1}{n}} + 8L\sqrt{\frac{r}{n}} + L \cdot \sqrt{\frac{2}{n}\log\frac{4}{\delta}} \\
&\quad + 12\kappa L\sqrt{\frac{1}{m}(4dr\log(24mdr(D^2 + 1)) + \log(2/\delta))} + L\sqrt{\frac{2}{\pi n}} \\
&\leq \tilde{\mathcal{O}}\left(\kappa L\sqrt{\frac{dr}{m}} + L\sqrt{\frac{r}{n}}\right), \tag{117}
\end{aligned}$$

where $L = (D^2 + 1)^3 \log n$ and $\kappa = c_1(\sigma_{max}^* + 1)^4/\sigma_{min}^{*3}$ for some absolute constants $c_1$.

$\square$

## B.5 FAST RATE FOR FACTOR MODELS WITH LINEAR REGRESSION AS DOWNSTREAM TASK

In this section, we provide a refined analysis for factor model, which implies a faster rate.

**Theorem B.9** (Fast rate). *Let $\hat{B}, \hat{\beta}$ be the outputs of Algorithm 1. Then, if $m \gtrsim (D^2+1)^2 d \log(1/\delta)$, $n \gtrsim (D^2+1)^2 r \log(1/\delta)$, for factor models with linear regression as downstream tasks, with probability at least $1 - \delta$, the excess risk can be bounded as follows,*

$$\text{Error}_\ell(\hat{B}, \hat{\beta}) \leq \mathcal{O}\left((D^2+1)^6(D^4 + \sigma_{\min}^{*-4})\frac{d\log(1/\delta)}{m} + (D^2+1)^2\frac{r\log(4/\delta)}{n}\right).$$

*Here $\mathcal{O}(\cdot)$ omits some absolute constants.*

*Proof of Theorem B.9.* First notice that we can rewrite our model (without $z$) as

$$y = \beta^{*T}C^*x + w, \tag{118}$$

where $\beta^* \in \mathbb{R}^{r \times 1}$, $C^* = B^{*T}(B^*B^{*T} + I_d)^{-1} \in \mathbb{R}^{r \times d}$, $x \sim N(0, B^*B^{*T} + I_d)$, $w \sim N(0, \epsilon^2 + \|\beta^*\|_2^2 - \beta^{*T}B^{*T}(B^*B^{*T} + I_d)^{-1}B^*\beta^*)$. Here $w$ and $x$ are independent. Therefore we can write our data as

$$Y = XC^{*T}\beta^* + W, \tag{119}$$

where $Y = (y_1, \cdots, y_n)^T \in \mathbb{R}^{n \times 1}$, $X = (x_1, \cdots, x_n)^T \in \mathbb{R}^{n \times d}$, $W = (w_1, \cdots, w_n)^T \in \mathbb{R}^{n \times 1}$.

In the first step (MLE), we obtain an estimator $\hat{B}$ and the corresponding estimator $\hat{C} = \hat{B}^T(\hat{B}\hat{B}^T + I_d)^{-1}$. Then our estimator $\hat{\beta}$ for the second step (ERM) is given by

$$\begin{aligned}
\hat{\beta} &= \arg\min_\beta \|Y - X\hat{C}^T\beta\|_2^2 \\
&= ((X\hat{C}^T)^T(X\hat{C}^T))^{-1}(X\hat{C}^T)^TY \\
&= (\hat{C}X^TX\hat{C}^T)^{-1}\hat{C}X^TY. \tag{120}
\end{aligned}$$

Then our risk is given by

$$\begin{aligned}
\text{Error}_\ell(\hat{B}, \hat{\beta}) &= \mathbb{E}_{\mathbb{P}_{B^*,\beta^*}(x,y)}\left[(y - g_{\hat{B},\hat{\beta}}(x))^2\right] - \mathbb{E}_{\mathbb{P}_{B^*,\beta^*}(x,y)}\left[(y - g_{B^*,\beta^*}(x))^2\right] \\
&= \mathbb{E}[(\beta^{*T}C^*x + w - \hat{\beta}^T\hat{B}^T(\hat{B}\hat{B}^T + I_d)^{-1}x)^2] - \mathbb{E}[w^2] \\
&= \mathbb{E}[(\beta^{*T}C^*x - \hat{\beta}^T\hat{C}x)^2] \\
&= (\beta^{*T}C^* - \hat{\beta}^T\hat{C})(B^*B^{*T} + I_d)(\beta^{*T}C^* - \hat{\beta}^T\hat{C})^T \\
&\leq \|B^*B^{*T} + I_d\|_2\|\hat{C}^T\hat{\beta} - C^{*T}\beta^*\|_2^2 \tag{121}
\end{aligned}$$

Our goal is to bound $\|\hat{C}^T\hat{\beta} - C^{*T}\beta^*\|_2^2$. Consider the SVD of $C^{*T}$ and $\hat{C}^T$, i.e., $C^{*T} = U^*\Lambda^*V^{*T}$, $\hat{C}^T = \hat{U}\hat{\Lambda}\hat{V}^T$. Then, we have

$$\begin{aligned}
&\hat{C}^T\hat{\beta} - C^{*T}\beta^* \\
&= \hat{C}^T(\hat{C}X^TX\hat{C}^T)^{-1}\hat{C}X^TY - C^{*T}\beta^* \\
&= \hat{C}^T(\hat{C}X^TX\hat{C}^T)^{-1}\hat{C}X^T(XC^{*T}\beta^* + W) - C^{*T}\beta^* \\
&= (\hat{C}^T(\hat{C}X^TX\hat{C}^T)^{-1}\hat{C}X^TXC^{*T} - C^{*T})\beta^* + \hat{C}^T(\hat{C}X^TX\hat{C}^T)^{-1}\hat{C}X^TW \\
&= (\hat{U}(\hat{U}^TX^TX\hat{U})^{-1}\hat{U}^TX^TXU^* - U^*)\Lambda^*V^{*T}\beta^* + \hat{U}(\hat{U}^TX^TX\hat{U})^{-1}\hat{U}^TX^TW. \tag{122}
\end{aligned}$$

Therefore

$$\begin{aligned}
\|\hat{C}^T\hat{\beta} - C^{*T}\beta^*\|_2^2 &\leq 2\|(\hat{U}(\hat{U}^TX^TX\hat{U})^{-1}\hat{U}^TX^TXU^* - U^*)\|_2^2\|\Lambda^*\|_2^2\|\beta^*\|_2^2 \\
&\quad + 2\|\hat{U}(\hat{U}^TX^TX\hat{U})^{-1}\hat{U}^TX^TW\|_2^2 \tag{123}
\end{aligned}$$

We give two lemmas for bounding the related terms. The first lemma considers the bias term:

**Lemma B.10.** *Let $\Sigma := B^*B^{*T} + I_d$. If $n \gtrsim \|\Sigma\|^2 r \log(1/\delta)$, then with probability at least $1 - \delta$,*

$$\|(\hat{U}(\hat{U}^TX^TX\hat{U})^{-1}\hat{U}^TX^TXU^* - U^*)\|_2^2 \leq \mathcal{O}(\|\Sigma\|^2\Delta^2), \tag{124}$$

*where $\Delta = dist(\hat{U}, U^*) := \|\hat{U}\hat{U}^T - U^*U^{*T}\|$.*

The second lemma considers the variance term:

**Lemma B.11.** *Let $\Sigma := B^* B^{*T} + I_d$. If $n \gtrsim \|\Sigma\|^2 r \log(1/\delta)$, then with probability at least $1 - \delta$,*

$$\|\hat{U}(\hat{U}^T X^T X \hat{U})^{-1} \hat{U}^T X^T W\|_2^2 \leq \mathcal{O}\left(\frac{\sigma^2 r \log(4/\delta)}{n}\right), \tag{125}$$

*where $\sigma^2 := \mathbb{E}(w^2) = \epsilon^2 + \|\beta^*\|_2^2 - \beta^{*T} B^{*T}(B^* B^{*T} + I_d)^{-1} B^* \beta^*$ is the variance of $w$.*

Using this two lemmas together with the decomposition (123), we have

$$\|\hat{C}^T \hat{\beta} - C^{*T} \beta^*\|_2^2 \leq \mathcal{O}\left(\|\beta^*\|^2 \|\Lambda^*\|^2 \|\Sigma\|^2 \Delta^2 + \frac{\sigma^2 r \log(4/\delta)}{n}\right). \tag{126}$$

Now it remains to control $\Delta$, which is related to the estimation error of the first step (MLE). The following lemma gives an upper bound for $\Delta$.

**Lemma B.12.** *If $m \gtrsim \|\Sigma\|^2 d \log(1/\delta)$, then with probability at least $1 - \delta$,*

$$\Delta^2 \leq \mathcal{O}\left(\|\Sigma\|^2 \frac{d \log(1/\delta)}{m} \lambda_r^{-2}(C^{*T} C^*)\right), \tag{127}$$

*where $\lambda_r(C^{*T} C^*)$ is the $r$-th (smallest) nonzero eigenvalue of $C^{*T} C^*$.*

By Lemma B.10, B.11, B.12, we have

$$\begin{aligned}
\text{Error}_\ell(\hat{B}, \hat{\beta}) &\leq \|\Sigma\| \|\hat{C}^T \hat{\beta} - C^{*T} \beta^*\|_2^2 \\
&\leq \mathcal{O}(\|\beta^*\|^2 \|\Lambda^*\|^2 \|\Sigma\|^3 \Delta^2 + \|\Sigma\| \frac{\sigma^2 r \log(4/\delta)}{n}). \\
&\leq \mathcal{O}(\|\beta^*\|^2 \|\Lambda^*\|^2 \|\Sigma\|^5 \lambda_r^{-2}(C^{*T} C^*) \frac{d \log(1/\delta)}{m} + \|\Sigma\| \frac{\sigma^2 r \log(4/\delta)}{n}).
\end{aligned} \tag{128}$$

Using the assumptions that $\|\beta^*\| \leq D$ and $\|B^*\| \leq D$, we can bound these terms by $D$ and quantities related to ground truth. First notice that $\Sigma$ have eigenvalues $\sigma_1^{*2} + 1 \geq \sigma_2^{*2} + 1 \geq \cdots \geq \sigma_r^{*2} + 1 \geq 1 = \cdots = 1$, where $\sigma_i^*$ are singular values of $B^*$, therefore $\|\Sigma\| \leq D^2 + 1$. Also, since

$$\begin{aligned}
C^{*T} C^* &= (B^* B^{*T} + I_d)^{-1} B^* B^{*T} (B^* B^{*T} + I_d)^{-1} \\
&= (B^* B^{*T} + I_d)^{-1} - (B^* B^{*T} + I_d)^{-2} \\
&= \Sigma^{-1} - \Sigma^{-2}, 
\end{aligned} \tag{129}$$

we know that $C^{*T} C^*$ has $r$ nonzero eigenvalues $\{(\sigma_i^* + \sigma_i^{*-1})^{-2}\}_{i=1}^r$. Therefore $\|\Lambda^*\|^2 = \|C^{*T} C^*\| \leq 1/4$,

$$\begin{aligned}
\lambda_r^{-2}(C^{*T} C^*) &\leq \max((\sigma_1^* + \sigma_1^{*-1})^4, (\sigma_r^* + \sigma_r^{*-1})^4) \\
&\leq \mathcal{O}(D^4 + \sigma_r^{*-4}).
\end{aligned} \tag{130}$$

For $\sigma^2$, we have

$$\begin{aligned}
\sigma^2 &= \epsilon^2 + \|\beta^*\|_2^2 - \beta^{*T} B^{*T}(B^* B^{*T} + I_d)^{-1} B^* \beta^* \\
&\leq 1 + \|\beta^*\|^2 \|I_r - B^{*T}(B^* B^{*T} + I_d)^{-1} B^*\| \\
&\leq 1 + D^2.
\end{aligned} \tag{131}$$

Combine all this bounds, we have

$$\begin{aligned}
\text{Error}_\ell(\hat{B}, \hat{\beta}) &\leq \mathcal{O}(\|\beta^*\|^2 \|\Lambda^*\|^2 \|\Sigma\|^5 \lambda_r^{-2}(C^{*T} C^*) \frac{d \log(1/\delta)}{m} + \|\Sigma\| \frac{\sigma^2 r \log(4/\delta)}{n}). \\
&\leq \mathcal{O}((D^2 + 1)^6 (D^4 + \sigma_{min}^{*-4}) \frac{d \log(1/\delta)}{m} + (D^2 + 1)^2 \frac{r \log(4/\delta)}{n}).
\end{aligned} \tag{132}$$

$\square$

In the sequel, we give the proofs of Lemma B.10, B.11 and B.12. We first prove some additional technical lemmas. The following lemma, which is a simple corollary of Tripuraneni et al. (2021) Lemma 20, shows the concentration property of empirical covariance matrix.

**Lemma B.13.** *Let $\Sigma \in \mathbb{R}^d$ be a positive definite matrix. Let $\{x_i\}_{i=1}^n$ be $d-$dimensional Gaussian random vectors i.i.d. sample from $N(0, \Sigma)$, $X = (x_1, \cdots, x_n)^T \in \mathbb{R}^{n \times d}$. Then for any $A, B \in \mathbb{R}^{d \times r}$, we have with probability at least $1 - \delta$*

$$\|A^T(\frac{X^T X}{n})B - A^T \Sigma B\|_2 \leq \mathcal{O}(\|A\|\|B\|\|\Sigma\|(\sqrt{\frac{r}{n}} + \frac{r}{n} + \sqrt{\frac{\log(1/\delta)}{n}} + \frac{\log(1/\delta)}{n}). \quad (133)$$

*Proof.* We write the SVD of $A$ and $B$: $A = U_1 \Lambda_1 V_1^T$, $B = U_2 \Lambda_2 V_2^T$, where $U_1, U_2 \in \mathbb{R}^{d \times r}$, $\Lambda_1, \Lambda_2, V_1, V_2 \in \mathbb{R}^{r \times r}$. Then

$$\|A^T(\frac{X^T X}{n})B - A^T \Sigma B\|_2 = \|V_1 \Lambda_1 U_1^T(\frac{X^T X}{n})U_2 \Lambda_2 V_2^T - V_1 \Lambda_1 U_1^T \Sigma U_2 \Lambda_2 V_2^T\|_2$$

$$\leq \|V_1 \Lambda_1\|\|U_1^T(\frac{X^T X}{n})U_2 - U_1^T \Sigma U_2\|\|\Lambda_2 V_2^T\|$$

$$\leq \|A\|\|B\|\|U_1^T(\frac{X^T X}{n})U_2 - U_1^T \Sigma U_2\|. \quad (134)$$

Now since $U_1, U_2 \in \mathbb{R}^{d \times r}$ are projection matrices, we can apply Tripuraneni et al. (2021) Lemma 20, therefore

$$\|U_1^T(\frac{X^T X}{n})U_2 - U_1^T \Sigma U_2\| \leq \mathcal{O}(\|\Sigma\|(\sqrt{\frac{r}{n}} + \frac{r}{n} + \sqrt{\frac{\log(1/\delta)}{n}} + \frac{\log(1/\delta)}{n})) \quad (135)$$

which gives what we want. $\square$

The following lemma is a basic matrix perturbation result (see Tripuraneni et al. (2021) Lemma 25).

**Lemma B.14.** *Let $A$ be a positive definite matrix and $E$ another matrix which satisfies $\|EA^{-1}\| \leq \frac{1}{4}$, then $F := (A + E)^{-1} - A^{-1}$ satisfies $\|F\| \leq \frac{4}{3}\|A^{-1}\|\|EA^{-1}\|$.*

With these two technical lemmas, we are able to prove Lemma B.10, B.11.

*Proof of Lemma B.10.* We consider $\hat{U} \in \mathbb{R}^{d \times r}$ and $\hat{U}_\perp^T \in \mathbb{R}^{d \times (d-r)}$ be orthonormal projection matrices spanning orthogonal subspaces which are rank $r$ and rank $d - r$ respectively, so that $\text{range}(\hat{U}) \oplus \text{range}(\hat{U}_\perp) = \mathbb{R}^d$. Then $\Delta = dist(\hat{U}, U^*) = \|\hat{U}_\perp^T U^*\|_2$ (see Chen et al. (2021) Lemma 2.5). Notice that $I_d = \hat{U}\hat{U}^T + \hat{U}_\perp \hat{U}_\perp^T$, we have

$$\hat{U}(\hat{U}^T X^T X \hat{U})^{-1}\hat{U}^T X^T X U^* - U^*$$

$$= \hat{U}(\hat{U}^T X^T X \hat{U})^{-1}\hat{U}^T X^T X (\hat{U}\hat{U}^T + \hat{U}_\perp \hat{U}_\perp^T)U^* - U^*$$

$$= \hat{U}(\hat{U}^T X^T X \hat{U})^{-1}\hat{U}^T X^T X \hat{U}\hat{U}^T U^* + \hat{U}(\hat{U}^T X^T X \hat{U})^{-1}\hat{U}^T X^T X \hat{U}_\perp \hat{U}_\perp^T U^* - U^*$$

$$= \hat{U}(\hat{U}^T X^T X \hat{U})^{-1}\hat{U}^T X^T X \hat{U}_\perp \hat{U}_\perp^T U^* + \hat{U}\hat{U}^T U^* - U^*$$

$$= \hat{U}(\hat{U}^T X^T X \hat{U})^{-1}\hat{U}^T X^T X \hat{U}_\perp \hat{U}_\perp^T U^* - \hat{U}_\perp \hat{U}_\perp^T U^* \quad (136)$$

Therefore

$$\|\hat{U}(\hat{U}^T X^T X \hat{U})^{-1}\hat{U}^T X^T X U^* - U^*\|_2^2 \leq 2\|\hat{U}(\hat{U}^T X^T X \hat{U})^{-1}\hat{U}^T X^T X \hat{U}_\perp \hat{U}_\perp^T U^*\|_2^2 + 2\|\hat{U}_\perp \hat{U}_\perp^T U^*\|_2^2. \quad (137)$$

For the second term,

$$\|\hat{U}_\perp \hat{U}_\perp^T U^*\|_2^2 \leq \|\hat{U}_\perp\|^2 \|\hat{U}_\perp^T U^*\|^2 \leq \Delta^2. \quad (138)$$

For the first term,

$$\|\hat{U}(\hat{U}^T X^T X \hat{U})^{-1}\hat{U}^T X^T X \hat{U}_\perp \hat{U}_\perp^T U^*\|$$

$$= \|\hat{U}(\hat{U}^T \frac{X^T X}{n} \hat{U})^{-1}\hat{U}^T \frac{X^T X}{n} \hat{U}_\perp \hat{U}_\perp^T U^*\|$$

$$= \|\hat{U}((\hat{U}^T \Sigma \hat{U})^{-1} + F)(\hat{U}^T \Sigma \hat{U}_\perp \hat{U}_\perp^T U^* + E_1)\|$$

$$\leq \|(\hat{U}^T \Sigma \hat{U})^{-1}(\hat{U}^T \Sigma \hat{U}_\perp \hat{U}_\perp^T U^*)\| + \|(\hat{U}^T \Sigma \hat{U})^{-1} E_1\| + \|F\hat{U}^T \Sigma \hat{U}_\perp \hat{U}_\perp^T U^*\| + \|FE_1\|, \quad (139)$$

where $E_1 = \hat{U}^T \frac{X^T X}{n} \hat{U}_\perp \hat{U}_\perp^T U^* - \hat{U}^T \Sigma \hat{U}_\perp \hat{U}_\perp^T U^*$, $F = (\hat{U}^T \frac{X^T X}{n} \hat{U})^{-1} - (\hat{U}^T \Sigma \hat{U})^{-1}$. In order to bound $\|F\|$, let $E = \hat{U}^T \frac{X^T X}{n} \hat{U} - \hat{U}^T \Sigma \hat{U}$, then by Lemma B.13, with probability at least $1 - \delta$,

$$\|E\| \leq \mathcal{O}(\|\Sigma\|(\sqrt{\frac{r}{n}} + \frac{r}{n} + \sqrt{\frac{\log(1/\delta)}{n}} + \frac{\log(1/\delta)}{n})). \tag{140}$$

Therefore, since $\lambda_{min}(\Sigma) = 1$,

$$\begin{aligned}
\|E(\hat{U}^T \Sigma \hat{U})^{-1}\| &\leq \|E\| \|(\hat{U}^T \Sigma \hat{U})^{-1}\| \\
&\leq \|E\| \lambda_{min}(\Sigma)^{-1} \\
&\leq \mathcal{O}(\|\Sigma\|(\sqrt{\frac{r}{n}} + \frac{r}{n} + \sqrt{\frac{\log(1/\delta)}{n}} + \frac{\log(1/\delta)}{n}))
\end{aligned} \tag{141}$$

Notice that $n \gtrsim \|\Sigma\|^2 r \log(1/\delta)$ implies $\sqrt{\frac{r}{n}} + \frac{r}{n} + \sqrt{\frac{\log(1/\delta)}{n}} + \frac{\log(1/\delta)}{n} \lesssim \|\Sigma\|^{-1}$. Thus, we show that when $n$ is large enough, we have $\|E(\hat{U}^T \Sigma \hat{U})^{-1}\| \leq \frac{1}{4}$. Therefore we can apply Lemma B.14, which gives

$$\begin{aligned}
\|F\| &\leq \frac{4}{3} \|E(\hat{U}^T \Sigma \hat{U})^{-1}\| \|(\hat{U}^T \Sigma \hat{U})^{-1}\| \\
&\leq \frac{4}{3} \times \frac{1}{4} \|(\hat{U}^T \Sigma \hat{U})^{-1}\| \\
&\leq \frac{1}{3}.
\end{aligned} \tag{142}$$

As for $\|E_1\|$, directly applying Lemma B.13, using $n \gtrsim \|\Sigma\|^2 r \log(1/\delta)$, we get

$$\begin{aligned}
\|E_1\| &\leq \mathcal{O}(\|\Sigma\| \|\hat{U}_\perp \hat{U}_\perp^T U^*\|(\sqrt{\frac{r}{n}} + \frac{r}{n} + \sqrt{\frac{\log(1/\delta)}{n}} + \frac{\log(1/\delta)}{n})) \\
&\leq \mathcal{O}(\|\Sigma\| \Delta \|\Sigma\|^{-1}) \\
&\leq \mathcal{O}(\Delta)
\end{aligned} \tag{143}$$

Combining (139),(142)and(143), we have

$$\begin{aligned}
&\|\hat{U}(\hat{U}^T X^T X \hat{U})^{-1} \hat{U}^T X^T X \hat{U}_\perp \hat{U}_\perp^T U^*\| \\
&\leq \|(\hat{U}^T \Sigma \hat{U})^{-1}(\hat{U}^T \Sigma \hat{U}_\perp \hat{U}_\perp^T U^*)\| + \|(\hat{U}^T \Sigma \hat{U})^{-1} E_1\| + \|F \hat{U}^T \Sigma \hat{U}_\perp \hat{U}_\perp^T U^*\| + \|F E_1\| \\
&\leq \|(\hat{U}^T \Sigma \hat{U})^{-1}\| \|(\hat{U}^T \Sigma \hat{U}_\perp \hat{U}_\perp^T U^*)\| + \|(\hat{U}^T \Sigma \hat{U})^{-1}\| \|E_1\| + \|F\| \|\hat{U}^T \Sigma \hat{U}_\perp \hat{U}_\perp^T U^*\| + \|F\| \|E_1\| \\
&\leq \lambda_{min}(\Sigma)^{-1} \|\Sigma\| \|\hat{U}_\perp^T U^*\| + \lambda_{min}(\Sigma)^{-1} \|E_1\| + \|F\| \|\Sigma\| \|\hat{U}_\perp^T U^*\| + \|F\| \|E_1\| \\
&\leq \lambda_{min}(\Sigma)^{-1} \|\Sigma\| \Delta + \lambda_{min}(\Sigma)^{-1} \mathcal{O}(\lambda_{min}(\Sigma)\Delta) + \frac{1}{3} \lambda_{min}(\Sigma)^{-1} \|\Sigma\| \Delta + \frac{1}{3} \lambda_{min}(\Sigma)^{-1} \mathcal{O}(\lambda_{min}(\Sigma)\Delta) \\
&\leq \mathcal{O}(\|\Sigma\| \Delta)
\end{aligned} \tag{144}$$

Finally, combining (137),(138) and (144), we get

$$\|(\hat{U}(\hat{U}^T X^T X \hat{U})^{-1} \hat{U}^T X^T X U^* - U^*)\|_2^2 \leq \mathcal{O}(\|\Sigma\|^2 \Delta^2), \tag{145}$$

with probability at least $1 - \delta$, which is what we want. $\square$

*Proof of Lemma B.11.*

$$\begin{aligned}
\|\hat{U}(\hat{U}^T X^T X \hat{U})^{-1} \hat{U}^T X^T W\|_2^2 &\leq \|(\hat{U}^T X^T X \hat{U})^{-1} \hat{U}^T X^T W\|_2^2 \\
&= ((\hat{U}^T X^T X \hat{U})^{-1} \hat{U}^T X^T W)^T ((\hat{U}^T X^T X \hat{U})^{-1} \hat{U}^T X^T W) \\
&= W^T (\frac{1}{n} \frac{X\hat{U}}{\sqrt{n}} (\hat{U}^T \frac{X^T X}{n} \hat{U})^{-2} \frac{\hat{U}^T X^T}{\sqrt{n}}) W.
\end{aligned} \tag{146}$$

Let $A = \frac{1}{n} \frac{X\hat{U}}{\sqrt{n}} (\hat{U}^T \frac{X^T X}{n} \hat{U})^{-2} \frac{\hat{U}^T X^T}{\sqrt{n}}$, $W = \sigma V$, then $V \sim N(0, I_n)$. By Hanson-Wright inequality (see Vershynin (2018) Theorem 6.2.1),

$$\mathbb{P}(|V^T A V - \mathbb{E}[V^T A V]| \geq t) \leq 2 \exp(-c \min(\frac{t^2}{\|A\|_F^2}, \frac{t}{\|A\|_2})). \tag{147}$$

Hence with probability at least $1 - \delta$,

$$V^T A V \leq \mathbb{E}[V^T A V] + \mathcal{O}(\|A\|_F \sqrt{\log \frac{2}{\delta}}) + \mathcal{O}(\|A\|_2 \log \frac{2}{\delta}). \tag{148}$$

Notice that $\mathbb{E}[V^T A V] = \text{Tr}(A)$, therefore it remains to bound $\text{Tr}(A)$, $\|A\|_F$ and $\|A\|_2$. If we define $B = \frac{X\hat{U}}{\sqrt{n}} \in \mathbb{R}^{n \times r}$, then $A = \frac{1}{n} B(B^T B)^{-2} B^T$. Therefore

$$\begin{aligned}
\text{Tr}(A) &= \text{Tr}(\frac{1}{n} B(B^T B)^{-2} B^T) \\
&= \frac{1}{n} \text{Tr}((B^T B)^{-2} B^T B) \\
&= \frac{1}{n} \text{Tr}((B^T B)^{-1}) \\
&\leq \frac{r}{n} \|(B^T B)^{-1}\|_2
\end{aligned} \tag{149}$$

Let the SVD of $B$ be $B = PMQ^T$, where $P \in \mathbb{R}^{n \times r}$, $M, Q \in \mathbb{R}^{r \times r}$, then

$$\begin{aligned}
\|A\|_2 &= \frac{1}{n} \|B(B^T B)^{-2} B^T\|_2 \\
&= \frac{1}{n} \|PMQ^T (QM^2 Q^T)^{-2} QMP^T\|_2 \\
&= \frac{1}{n} \|PM^{-2} P^T\|_2 \\
&\leq \frac{1}{n} \|M^{-2}\|_2 \\
&= \frac{1}{n} \|(B^T B)^{-1}\|_2
\end{aligned} \tag{150}$$

Also notice that $A$ is rank $r$, therefore $\|A\|_F \leq \sqrt{r} \|A\|_2$. Thus it remains to bound $\|(B^T B)^{-1}\|_2 = \|(\hat{U}^T \frac{X^T X}{n} \hat{U})^{-1}\|_2$. Let $F = (\hat{U}^T \frac{X^T X}{n} \hat{U})^{-1} - (\hat{U}^T \Sigma \hat{U})^{-1}$. Recall (142), which states that with probability at least $1 - \delta$, we have $\|F\| \leq \frac{1}{3} \lambda_{min}(\Sigma)^{-1}$. Therefore

$$\begin{aligned}
\|(\hat{U}^T \frac{X^T X}{n} \hat{U})^{-1}\| &= \|(\hat{U}^T \Sigma \hat{U})^{-1} + F\| \\
&\leq \|(\hat{U}^T \Sigma \hat{U})^{-1}\| + \|F\| \\
&\leq \mathcal{O}(\lambda_{min}(\Sigma)^{-1}).
\end{aligned} \tag{151}$$

Thus $\|A\| \leq \mathcal{O}(\frac{1}{n} \lambda_{min}(\Sigma)^{-1})$, $\|A\|_F \leq \mathcal{O}(\frac{\sqrt{r}}{n} \lambda_{min}(\Sigma)^{-1})$, $\text{Tr}(A) \leq \mathcal{O}(\frac{r}{n} \lambda_{min}(\Sigma)^{-1})$. Therefore with probability at least $1 - 2\delta$,

$$\begin{aligned}
V^T A V &\leq \mathbb{E}[V^T A V] + \mathcal{O}(\|A\|_F \sqrt{\log \frac{2}{\delta}}) + \mathcal{O}(\|A\|_2 \log \frac{2}{\delta}) \\
&\leq \mathcal{O}(\frac{r}{n} \lambda_{min}(\Sigma)^{-1}) + \mathcal{O}(\frac{\sqrt{r}}{n} \lambda_{min}(\Sigma)^{-1} \sqrt{\log \frac{2}{\delta}}) + \mathcal{O}(\frac{1}{n} \lambda_{min}(\Sigma)^{-1} \log \frac{2}{\delta}) \\
&\leq \mathcal{O}(\frac{r}{n} \lambda_{min}(\Sigma)^{-1} \log \frac{2}{\delta}) \\
&= \mathcal{O}(\frac{r}{n} \log \frac{2}{\delta}).
\end{aligned} \tag{152}$$

The last line holds since $\lambda_{min}(\Sigma) = 1$. Recall

$$\|\hat{U}(\hat{U}^T X^T X \hat{U})^{-1} \hat{U}^T X^T W\|_2^2 = W^T A W = \sigma^2 V^T A V, \tag{153}$$

combining this with the above bound for $V^T A V$ yields our desired result. $\qquad \square$

Finally we prove Lemma B.12 in the following.

*Proof of Lemma B.12.* In the first step, we have $m$ unlabeled data $\{x_i\}_{i=1}^m$ i.i.d. sample from $N(0, \Sigma)$. Let $\hat{\Sigma} = \frac{1}{m} \sum_{i=1}^m x_i x_i^T$ be the empirical covariance matrix. Then by Lemma B.13, with probability at least $1 - \delta$,

$$\|\Sigma - \hat{\Sigma}\| \leq \mathcal{O}(\|\Sigma\|(\sqrt{\frac{d}{m}} + \frac{d}{m} + \sqrt{\frac{\log(1/\delta)}{m}} + \frac{\log(1/\delta)}{m})) \tag{154}$$

We claim that

$$\|\hat{B}\hat{B}^T - (\hat{\Sigma} - I_d)\|_2 \leq \|\hat{\Sigma} - \Sigma\|, \tag{155}$$

and the proof of this claim will be at the end of this section. With the claim,

$$\begin{aligned}
\|\hat{B}\hat{B}^T - B^* B^{*T}\| &= \|\hat{B}\hat{B}^T - (\hat{\Sigma} - I_d) + (\hat{\Sigma} - I_d) - (\Sigma - I_d)\| \\
&\leq \|\hat{B}\hat{B}^T - (\hat{\Sigma} - I_d)\| + \|\Sigma - \hat{\Sigma}\| \\
&\leq 2\|\Sigma - \hat{\Sigma}\|.
\end{aligned} \tag{156}$$

Notice that

$$\begin{aligned}
C^{*T} C^* &= (B^* B^{*T} + I_d)^{-1} B^* B^{*T} (B^* B^{*T} + I_d)^{-1} \\
&= (B^* B^{*T} + I_d)^{-1} - (B^* B^{*T} + I_d)^{-2}
\end{aligned} \tag{157}$$

Similarly

$$\hat{C}^T \hat{C} = (\hat{B}\hat{B}^T + I_d)^{-1} - (\hat{B}\hat{B}^T + I_d)^{-2}. \tag{158}$$

Let $E_2 = (\hat{B}\hat{B}^T + I_d) - (B^* B^{*T} + I_d)$, $F_2 = (\hat{B}\hat{B}^T + I_d)^{-1} - (B^* B^{*T} + I_d)^{-1}$. Then

$$\|E_2\| \leq 2\|\Sigma - \hat{\Sigma}\| \leq \mathcal{O}(\|\Sigma\|(\sqrt{\frac{d}{m}} + \frac{d}{m} + \sqrt{\frac{\log(1/\delta)}{m}} + \frac{\log(1/\delta)}{m})). \tag{159}$$

Therefore when $m \gtrsim \|\Sigma\|^2 d \log(1/\delta)$, $\|E_2\| \leq \mathcal{O}(\|\Sigma\| \sqrt{\frac{d \log(1/\delta)}{m}})$, $\|E_2 \Sigma^{-1}\| \leq \|E_2\| \|\Sigma^{-1}\| \leq 1/4$. Then we can apply Lemma B.14, which gives

$$\begin{aligned}
\|F_2\| &\leq \frac{4}{3} \|\Sigma^{-1}\| \|E_2 \Sigma^{-1}\| \\
&\leq \frac{4}{3} \|\Sigma^{-1}\|^2 \|E_2\| \\
&\leq \mathcal{O}(\lambda_{min}^{-2}(\Sigma) \|\Sigma\| \sqrt{\frac{d \log(1/\delta)}{m}}) \\
&= \mathcal{O}(\|\Sigma\| \sqrt{\frac{d \log(1/\delta)}{m}}).
\end{aligned} \tag{160}$$

The last line holds since $\lambda_{min}(\Sigma) = 1$. Thus

$$\begin{aligned}
\|C^{*T} C^* - \hat{C}^T \hat{C}\| &= \|(\Sigma^{-1} + F_2) - (\Sigma^{-1} + F_2)^2 - (\Sigma^{-1} - \Sigma^{-2})\| \\
&= \|F_2 - \Sigma^{-1} F_2 - F_2 \Sigma^{-1} - F_2^2\| \\
&\leq \|F_2\| + 2\|\Sigma^{-1}\| \|F_2\| + \|F_2\|^2 \\
&\leq \mathcal{O}(\|\Sigma\| \sqrt{\frac{d \log(1/\delta)}{m}}).
\end{aligned} \tag{161}$$

Therefore by Davis-Kahan theorem,

$$\Delta = dist(U^*, \hat{U}) \leq \mathcal{O}(\lambda_r^{-1}(C^{*T} C^*) \|C^{*T} C^* - \hat{C}^T \hat{C}\|). \tag{162}$$

Combining the above three inequalities, we have

$$\Delta^2 \leq \mathcal{O}(\|\Sigma\|^2 \frac{d \log(1/\delta)}{m} \lambda_r^{-2}(C^{*T} C^*)). \tag{163}$$

Finally we will need to prove the claim (155). Notice that the MLE estimator $\hat{B}$ is given by

$$
\begin{aligned}
\hat{B} &= \arg\max_{B \in \mathbb{R}^{d \times r}} \sum_{i=1}^{m} p_B(x_i) \\
&= \arg\max_{B \in \mathbb{R}^{d \times r}} (-\log\det(BB^T + I_d) - \text{Tr}(\hat{\Sigma}(BB^T + I_d)^{-1})) \\
&= \arg\min_{B \in \mathbb{R}^{d \times r}} (\log\det(BB^T + I_d) + \text{Tr}(\hat{\Sigma}(BB^T + I_d)^{-1}))
\end{aligned}
\tag{164}
$$

Let $\hat{\Sigma} = \hat{U}\hat{\Lambda}\hat{U}^T$ and $(BB^T + I_d) = U\Lambda U^T$, where $\hat{U}$ and $U$ are orthogonal matrices, $\hat{\Lambda} = \text{diag}(\hat{\lambda}_1, \cdots, \hat{\lambda}_d)$, $\Lambda = \text{diag}(\lambda_1, \cdots, \lambda_d)$ and $\hat{\lambda}_1 \geq \ldots \geq \hat{\lambda}_d, \lambda_1 \geq \ldots \geq \lambda_d$. Since $\text{rank}(BB^T) \leq r$, we have $\lambda_{r+1} = \ldots \lambda_d = 1$. By Ruhe's trace inequality (see P341 of Marshall et al. (2011)), we have

$$
\text{Tr}(\hat{\Sigma}(BB^T + I_d)^{-1})) \geq \sum_{j=1}^{d} \lambda_j^{-1}\hat{\lambda}_j,
\tag{165}
$$

and the equality holds only when the two matrices have simultaneous ordered spectral decomposition, i.e., $U = \hat{U}$. Therefore

$$
\begin{aligned}
&\min_{B \in \mathbb{R}^{d \times r}} (\log\det(BB^T + I_d) + \text{Tr}(\hat{\Sigma}(BB^T + I_d)^{-1})) \\
&= \min_{\{\lambda_j\}_{j=1}^d} \sum_{j=1}^{d} (\log\lambda_j + \lambda_j^{-1}\hat{\lambda}_j) \quad \text{subject to } \lambda_1 \geq \cdots \geq \lambda_r \geq \lambda_{r+1} = \cdots = \lambda_d = 1
\end{aligned}
\tag{166}
$$

and the minimum is achieved when $\lambda_j = \hat{\lambda}_j$, for $j = 1, \cdots, r$. Therefore the MLE estimator $\hat{B}$ satisfies $(\hat{B}\hat{B}^T + I_d) = \hat{U}\Lambda\hat{U}^T$ where $\Lambda = \text{diag}(\hat{\lambda}_1, \cdots, \hat{\lambda}_r, 1, \cdots, 1)$. Thus, we have $\hat{B}\hat{B}^T = \hat{U}(\Lambda - I_d)\hat{U}^T$, which implies

$$
\begin{aligned}
&\|\hat{B}\hat{B}^T - (\hat{\Sigma} - I_d)\|_2 \\
&= \|\hat{U}(\Lambda - I_d)\hat{U}^T - \hat{U}(\hat{\Lambda} - I_d)\hat{U}^T\| \\
&\leq \|\Lambda - \hat{\Lambda}\| \\
&= \max_{j=r+1,\cdots,d} |\hat{\lambda}_j - 1| \\
&\leq \max_{j=1,\cdots,d} |\hat{\lambda}_j - \lambda_j(\Sigma)| \\
&\leq \|\hat{\Sigma} - \Sigma\|.
\end{aligned}
\tag{167}
$$

Here the last inequality follows from Weyl's Theorem. Thus, we prove claim (155). □

## C  PROOFS FOR SECTION 5

In Section C.1, we show that GMM with classification as downstream tasks has $c_2$-transferability for some absolute constants $c_2$ (Lemma 5.2). In Section C.2 and Section C.3, we prove two lemmas that will be used in the proof of Theorem 5.3. To be specific, in Section C.2, we upper bound the bracketing number of the set $\mathcal{P}(\mathcal{U})$ by using $\epsilon$-discretization (Lemma C.5). In Section C.3, we prove Lemma C.6, which will be used to upper bound the Rademacher complexity of the function class $\ell \circ \mathcal{G}_{\mathbf{u},\Psi}$. Finally, in Section C.4, we prove Theorem 5.3.

### C.1  PROOFS FOR LEMMA 5.2

Before going to the proof of this theorem, we first state some basic definitions and useful lemmas. We define the balls of radius $8\sqrt{d\log K}$ around each $u_i^*$ and $u_i$ as

$$
\Omega_i^* := \left\{ x \in \mathbb{R}^d \mid \|x - u_i^*\| \leq 8\sqrt{d\log K} \right\}
\tag{168}
$$

$$
\Omega_i := \left\{ x \in \mathbb{R}^d \mid \|x - u_i\| \leq 8\sqrt{d\log K} \right\}
\tag{169}
$$

We denote the p.d.f of $\mathcal{N}(u_i, I_d)$ and $\mathcal{N}(u_i^*, I_d)$ by $P_i$ and $P_i^*$ respectively.

**Lemma C.1.** *If*

$$d_{\text{TV}}\left(p_{\mathbf{u}}(x), p_{\mathbf{u}^*}(x)\right) \leq \frac{1}{4K}, \tag{170}$$

*then there exists a permutation of $\mathbf{u}$ such that $\|u_i^* - u_i\| \leq 16\sqrt{d\log K}$ holds for every $1 \leq i \leq K$.*

Before proving Lemma C.1, we first state a useful result of Gaussian norm concentration.
**Lemma C.2.** *Let $X \sim \mathcal{N}\left(0, I_d\right)$, then*

$$\mathbb{P}(\|X\| \geq t) \leq 2\exp(-\frac{t^2}{16d}). \tag{171}$$

*Proof.* This is a simple application of Jin et al. (2019) Lemma 1.3. Notice that $X$ is 1-subGaussian, therefore taking $\sigma = \sqrt{d}$ in Jin et al. (2019) Lemma 1.3 yields what we want. $\square$

*Proof of Lemma C.1.* We prove by contradiction. If the statement is not true, since the separation satisfies $100\sqrt{d\log K} \geq 2 \cdot 16\sqrt{d\log K}$, there must exist a $u_i^*$ (W.L.O.G., denote it by $u_1^*$), such that $\|u_1^* - u_j\| > 16\sqrt{d\log K}$ for any $1 \leq j \leq K$. Then

$$
\begin{aligned}
2d_{\text{TV}}\left(p_{\mathbf{u}}(x), p_{\mathbf{u}^*}(x)\right) &= \int_{\mathbb{R}^d} \left| \frac{1}{K}\sum_{i=1}^K P_i^* - \frac{1}{K}\sum_{i=1}^K P_i \right| \mathrm{d}x \\
&\geq \int_{\Omega_1^*} \left| \frac{1}{K}\sum_{i=1}^K P_i^* - \frac{1}{K}\sum_{i=1}^K P_i \right| \mathrm{d}x \\
&\geq \int_{\Omega_1^*} \frac{1}{K}\sum_{i=1}^K P_i^* \mathrm{d}x - \int_{\Omega_1^*} \frac{1}{K}\sum_{i=1}^K P_i \mathrm{d}x \\
&\geq \int_{\Omega_1^*} \frac{1}{K} P_1^* \mathrm{d}x - \frac{1}{K}\sum_{i=1}^K \int_{\Omega_1^*} P_i \mathrm{d}x \\
&= \frac{1}{K}\mathbb{P}(\mathcal{N}\left(u_1^*, I_d\right) \in \Omega_1^*) - \frac{1}{K}\sum_{i=1}^K \mathbb{P}(\mathcal{N}\left(u_i, I_d\right) \in \Omega_1^*) \tag{172}
\end{aligned}
$$

Since $\|u_1^* - u_i\| > 16\sqrt{d\log K}$, therefore $\Omega_1^* \cap \Omega_i = \emptyset$, which implies (by Lemma C.2)

$$\mathbb{P}(\mathcal{N}\left(u_i, I_d\right) \in \Omega_1^*) \leq \mathbb{P}(\mathcal{N}\left(u_i, I_d\right) \in \Omega_i^{\mathbf{C}}) \leq 2\exp(-\frac{(8\sqrt{d\log K})^2}{16d}) = 2e^{-4\log K} \tag{173}$$

Also, by Lemma C.2,

$$\mathbb{P}(\mathcal{N}\left(u_1^*, I_d\right) \in \Omega_1^*) \geq 1 - 2\exp(-\frac{(8\sqrt{d\log K})^2}{16d}) = 1 - 2e^{-4\log K} \tag{174}$$

Therefore,

$$
\begin{aligned}
2d_{\text{TV}}\left(p_{\mathbf{u}}(x), p_{\mathbf{u}^*}(x)\right) &\geq \frac{1}{K}\mathbb{P}(\mathcal{N}\left(u_1^*, I_d\right) \in \Omega_1^*) - \frac{1}{K}\sum_{i=1}^K \mathbb{P}(\mathcal{N}\left(u_i, I_d\right) \in \Omega_1^*) \\
&\geq \frac{1}{K}(1 - 2e^{-4\log K}) - \frac{1}{K}\sum_{i=1}^K 2e^{-4\log K} \\
&= \frac{1}{K} - (2 + \frac{2}{K})e^{-4\log K} \\
&\geq \frac{1}{K} - 3e^{-4\log K} \\
&= \frac{1}{K} - 3(\frac{1}{K})^4 \\
&= \frac{1}{2K} \tag{175}
\end{aligned}
$$

which is a contradiction. $\square$

We then state the core lemmas of proving Lemma 5.2.

**Lemma C.3.** *If for any $i$, $\|u_i - u_i^*\| \leq 16\sqrt{d \log K}$, then for $\Omega_1^*$ (corresponding results hold for each $\Omega_i^*$),*

$$\int_{\Omega_1^*} |P_1^* - P_1| \mathrm{d}x \geq c_1 \min\{\|u_1^* - u_1\|, 1\}, \tag{176}$$

*where $c_1 = \frac{1}{200}$.*

**Lemma C.4.** *If for any $i$, $\|u_i - u_i^*\| \leq 16\sqrt{d \log K}$, then for $\Omega_1^*$ (corresponding results hold for each $\Omega_i^*$), then for every $j \neq 1$,*

$$\int_{\Omega_1^*} |P_j^* - P_j| \mathrm{d}x \leq \frac{c_2}{K} \min\{\|u_j^* - u_j\|, 1\}, \tag{177}$$

*where $c_2 = 2688 \left(\frac{1}{2}\right)^{69}$.*

With these lemmas, we are now able to prove Lemma 5.2.

*Proof of Lemma 5.2.* By Lemma C.1, there exists a permutation of $\boldsymbol{u}$ such that $\|u_i^* - u_i\| \leq 16\sqrt{d \log K}$ holds for every $1 \leq i \leq K$. Therefore Lemma C.3, C.4 can be applied. Notice that

$$
\begin{aligned}
\int_{\Omega_1^*} |p_{\mathbf{u}}(x) - p_{\mathbf{u}^*}(x)| \mathrm{d}x &= \int_{\Omega_1^*} \left| \frac{1}{K} \sum_{i=1}^K P_i^* - \frac{1}{K} \sum_{i=1}^K P_i \right| \mathrm{d}x \\
&\geq \int_{\Omega_1^*} \left| \frac{1}{K} P_1^* - \frac{1}{K} P_i \right| \mathrm{d}x - \int_{\Omega_1^*} \left| \frac{1}{K} \sum_{i=2}^K P_i^* - \frac{1}{K} \sum_{i=2}^K P_i \right| \mathrm{d}x \\
&\geq \frac{1}{K} \int_{\Omega_1^*} |P_1^* - P_i| \mathrm{d}x - \frac{1}{K} \sum_{i=2}^K \int_{\Omega_1^*} |P_i^* - P_i| \mathrm{d}x \\
&\geq \frac{c_1}{K} \min\{\|u_1^* - u_1\|, 1\} - \frac{c_2}{K^2} \sum_{i=2}^K \min\{\|u_i^* - u_i\|, 1\}, \tag{178}
\end{aligned}
$$

where the last line comes from Lemma C.3, C.4.

Sum up all the equations above for corresponding $1 \leq i \leq K$, since $\{\Omega_i^*\}_{i=1}^K$ are disjoint, we have

$$
\begin{aligned}
d_{\mathrm{TV}}(p_{\mathbf{u}}(x), p_{\mathbf{u}^*}(x)) &= \frac{1}{2} \int_{\mathbb{R}^d} |p_{\mathbf{u}}(x) - p_{\mathbf{u}^*}(x)| \mathrm{d}x \\
&\geq \frac{1}{2} \sum_{i=1}^K \int_{\Omega_i^*} |p_{\mathbf{u}}(x) - p_{\mathbf{u}^*}(x)| \mathrm{d}x \\
&\geq \frac{1}{2} \left( \frac{c_1}{K} - \frac{(K-1)c_2}{K^2} \right) \sum_{i=1}^K \min\{\|u_i^* - u_i\|, 1\} \\
&\geq \frac{1}{2} (c_1 - c_2) \cdot \frac{1}{K} \sum_{i=1}^K \min\{\|u_i^* - u_i\|, 1\} \\
&= \frac{1}{2} \left( \frac{1}{200} - 2688 \left(\frac{1}{2}\right)^{69} \right) \cdot \frac{1}{K} \sum_{i=1}^K \min\{\|u_i^* - u_i\|, 1\} \\
&\geq \frac{1}{500} \cdot \frac{1}{K} \sum_{i=1}^K \min\{\|u_i^* - u_i\|, 1\}. \tag{179}
\end{aligned}
$$

In the end, we refer to Lemma B.1, which states that

$$d_{\mathrm{TV}}(\mathcal{N}(u_i^*, I_d), \mathcal{N}(u_i, I_d)) \leq \min(\|u_i^* - u_i\|, 1). \tag{180}$$

Take $\sigma(\boldsymbol{u}) = \{u_i\}_{i=1}^K$,

$$d_{\text{TV}}\left(p_{\sigma(\mathbf{u})}(x,z), p_{\mathbf{u}^*}(x,z)\right) = \sum_{i=1}^{K}\mathbb{P}(z=i)d_{\text{TV}}(\mathcal{N}(u_i^*, I_d), \mathcal{N}(u_i, I_d))$$

$$\leq \sum_{i=1}^{K}\frac{1}{K}\min(\|u_i^* - u_i\|, 1)$$

$$\leq 500 d_{\text{TV}}\left(p_{\mathbf{u}}(x), p_{\mathbf{u}^*}(x)\right). \tag{181}$$

$\square$

Finally we state the proof of Lemma C.3 and C.4.

*Proof of Lemma C.3.* W.L.O.G.,let $u_1^* = 0$, $\Delta := \|u_1\| \leq 16\sqrt{d\log K}$, and $u_1 = (-\Delta, 0, 0, \cdots, 0)$. The densities are given by

$$P_1^*(x) = \left(\frac{1}{\sqrt{2\pi}}\right)^d e^{-\frac{1}{2}\|x\|^2} \tag{182}$$

$$P_1(x) = \left(\frac{1}{\sqrt{2\pi}}\right)^d e^{-\frac{1}{2}\|x-u_1\|^2} \tag{183}$$

We consider an area $S \subset \Omega_1^*$:

$$S := \left\{x = (x_1, \cdots, x_d)\middle| x \in \Omega_1^*, x_1 \geq \frac{1}{10}\right\} \tag{184}$$

Then for any $x \in S$, $\|x\|^2 \leq \|x - u_1\|^2$, which implies $P_1^*(x) \geq P_1(x)$. Therefore

$$\int_{\Omega_1^*}|P_1^* - P_1|\mathrm{d}x \geq \int_S |P_1^* - P_1|\mathrm{d}x$$

$$= \int_S \left(\frac{1}{\sqrt{2\pi}}\right)^d \left(e^{-\frac{1}{2}\|x\|^2} - e^{-\frac{1}{2}\|x-u_1\|^2}\right)\mathrm{d}x$$

$$= \int_S \left(\frac{1}{\sqrt{2\pi}}\right)^d e^{-\frac{1}{2}\|x\|^2}\left(1 - e^{\frac{1}{2}\|x\|^2 - \frac{1}{2}\|x-u_1\|^2}\right)\mathrm{d}x$$

$$\geq \min_{x \in S}\left(1 - e^{\frac{1}{2}\|x\|^2 - \frac{1}{2}\|x-u_1\|^2}\right)\int_S \left(\frac{1}{\sqrt{2\pi}}\right)^d e^{-\frac{1}{2}\|x\|^2}\mathrm{d}x$$

$$= \min_{x \in S}\left(1 - e^{\frac{1}{2}\|x\|^2 - \frac{1}{2}\|x-u_1\|^2}\right)\mathbb{P}(\mathcal{N}(0, I_d) \in S) \tag{185}$$

For $\min_{x \in S}\left(1 - e^{\frac{1}{2}\|x\|^2 - \frac{1}{2}\|x-u_1\|^2}\right)$, notice that for any $x = (x_1, \cdots, x_d) \in S$,

$$\frac{1}{2}\|x\|^2 - \frac{1}{2}\|x - u_1\|^2 = -x_1\Delta - \frac{1}{2}\Delta^2 \leq -\frac{1}{10}\Delta \tag{186}$$

Thus

$$\min_{x \in S}\left(1 - e^{\frac{1}{2}\|x\|^2 - \frac{1}{2}\|x-u_1\|^2}\right) \geq 1 - e^{-\frac{1}{10}\Delta} \tag{187}$$

Take $c_3 = \frac{1}{20}$. We claim that

$$\min_{x \in S}\left(1 - e^{\frac{1}{2}\|x\|^2 - \frac{1}{2}\|x-u_1\|^2}\right) \geq c_3\min\{\Delta, 1\}. \tag{188}$$

In fact, when $0 \leq \Delta \leq 1$,

$$\min_{x \in S}\left(1 - e^{\frac{1}{2}\|x\|^2 - \frac{1}{2}\|x-u_1\|^2}\right) \geq 1 - e^{-\frac{1}{10}\Delta} \geq \frac{1}{20}\Delta. \tag{189}$$

The last inequality holds, since if we let $f(x) = e^{-\frac{1}{10}x} + \frac{1}{20}x - 1$, Then $f(0) = 0$,

$$f'(x) = -\frac{1}{10}e^{-\frac{1}{10}x} + \frac{1}{20} \leq 0 \tag{190}$$

for any $x \in [0, 10\log 2]$. Thus for any $\Delta \in [0, 1]$,

$$e^{-\frac{1}{10}\Delta} + \frac{1}{20}\Delta - 1 = f(\Delta) \leq f(0) = 0. \tag{191}$$

When $1 \leq \Delta \leq 16\sqrt{d\log K}$,

$$\min_{x \in S}\left(1 - e^{\frac{1}{2}\|x\|^2 - \frac{1}{2}\|x - u_1\|^2}\right) \geq 1 - e^{-\frac{1}{10}\Delta} \geq 1 - e^{-\frac{1}{10}} \geq \frac{1}{20} \cdot 1 \tag{192}$$

Therefore we have shown that

$$\min_{x \in S}\left(1 - e^{\frac{1}{2}\|x\|^2 - \frac{1}{2}\|x - u_1\|^2}\right) \geq c_3 \min\{\Delta, 1\}. \tag{193}$$

where $c_3 = \frac{1}{20}$.

As for $\mathbb{P}(\mathcal{N}(0, I_d) \in S)$, take

$$S' := \left\{x = (x_1, \cdots, x_d)\middle| 2\sqrt{d\log 2} \geq x_1 \geq \frac{1}{10}, x_2^2 + \cdots + x_d^2 \leq 60d\log K\right\}. \tag{194}$$

Then $S' \subset S$. Therefore

$$\begin{aligned}
\mathbb{P}(\mathcal{N}(0, I_d) \in S) &\geq \mathbb{P}(\mathcal{N}(0, I_d) \in S')\\
&= \mathbb{P}(2\sqrt{d\log 2} \geq x_1 \geq \frac{1}{10}, x_2^2 + \cdots + x_d^2 \leq 60d\log K, x \sim \mathcal{N}(0, I_d))\\
&= \mathbb{P}\left(2\sqrt{d\log 2} \geq \mathcal{N}(0, 1) \geq \frac{1}{10}\right)\mathbb{P}\left(\|\mathcal{N}(0, I_{d-1})\|^2 \leq 60d\log K\right)\\
&\geq \mathbb{P}\left(2\sqrt{\log 2} \geq \mathcal{N}(0, 1) \geq \frac{1}{10}\right)\mathbb{P}\left(\|\mathcal{N}(0, I_{d-1})\|^2 \leq 60(d-1)\log 2\right)\\
&> \mathbb{P}\left(2\sqrt{\log 2} \geq \mathcal{N}(0, 1) \geq \frac{1}{10}\right) \cdot (1 - 2e^{-2}) \quad \text{(by Lemma C.2)}\\
&> \frac{1}{4} \cdot (1 - 2e^{-2})\\
&> \frac{1}{10}
\end{aligned} \tag{195}$$

Combine all these results, we have

$$\begin{aligned}
\int_{\Omega_1^*}|P_1^* - P_1|\mathrm{d}x &\geq \min_{x \in S}\left(1 - e^{\frac{1}{2}\|x\|^2 - \frac{1}{2}\|x - u_1\|^2}\right)\mathbb{P}(\mathcal{N}(0, I_d) \in S)\\
&\geq c_3\min\{\Delta, 1\} \cdot \frac{1}{10}\\
&= \frac{1}{200}\min\{\|u_1^* - u_1\|, 1\}
\end{aligned} \tag{196}$$

$\square$

*Proof of Lemma C.4.* For any $i \neq 1$,

$$\int_{\Omega_1^*}|P_i^* - P_i|\mathrm{d}x = \int_{\Omega_1^*}(\frac{1}{\sqrt{2\pi}})^d|e^{-\frac{1}{2}\|x - u_i^*\|^2} - e^{-\frac{1}{2}\|x - u_i\|^2}|\mathrm{d}x. \tag{197}$$

Notice that if we denote $a(x) := \|x - u_i^*\|$, $\delta(x) := \|x - u_i^*\| - \|x - u_i\|$, $\Delta := \|u_i - u_i^*\|$, then $|\delta(x)| \leq \Delta \leq 16\sqrt{d\log K}$, and for any $x \in \Omega_1^*$, $a(x) \geq 92\sqrt{d\log K}$ (due to separation condition).

Therefore

$$\max_{x \in \Omega_1^*} \left| e^{-\frac{1}{2}\|x-u_i^*\|^2} - e^{-\frac{1}{2}\|x-u_i\|^2} \right|$$

$$= \max_{x \in \Omega_1^*} \left| e^{-\frac{1}{2}a(x)^2} - e^{-\frac{1}{2}(a(x)-\delta(x))^2} \right|$$

$$\leq \max \left\{ \left| e^{-\frac{1}{2}a(x)^2} - e^{-\frac{1}{2}(a(x)-\delta(x))^2} \right| \Big| a(x) \geq 92\sqrt{d\log K}, |\delta(x)| \leq \Delta \right\}$$

$$\leq \max_{a \geq 92\sqrt{d\log K}} \{\max(|e^{-\frac{a^2}{2}} - e^{-\frac{(a-\Delta)^2}{2}}|, |e^{-\frac{a^2}{2}} - e^{-\frac{(a+\Delta)^2}{2}}|)\}$$

$$= \max_{a \geq 92\sqrt{d\log K}} \{\max(e^{-\frac{(a-\Delta)^2}{2}} - e^{-\frac{a^2}{2}}, e^{-\frac{a^2}{2}} - e^{-\frac{(a+\Delta)^2}{2}})\}$$

$$\leq \max( \max_{a \geq 92\sqrt{d\log K}} e^{-\frac{(a-\Delta)^2}{2}} - e^{-\frac{a^2}{2}}, \max_{a \geq 92\sqrt{d\log K}} e^{-\frac{a^2}{2}} - e^{-\frac{(a+\Delta)^2}{2}}))$$

$$\leq \max( \max_{a \geq 76\sqrt{d\log K}} e^{-\frac{a^2}{2}} - e^{-\frac{(a+\Delta)^2}{2}}, \max_{a \geq 92\sqrt{d\log K}} e^{-\frac{a^2}{2}} - e^{-\frac{(a+\Delta)^2}{2}})). \tag{198}$$

The last inequality holds since $\Delta \leq 16\sqrt{d\log K}$. For fixed $\Delta$, let $f(a) = e^{-\frac{a^2}{2}} - e^{-\frac{(a+\Delta)^2}{2}}$. Then

$$f'(a) = -ae^{-\frac{a^2}{2}} + (a+\Delta)e^{-\frac{(a+\Delta)^2}{2}} \tag{199}$$

We first show that $f'(a) \leq 0$, for any $a \geq 76\sqrt{d\log K}$. Notice that

$$f'(a) = -ae^{-\frac{a^2}{2}} + (a+\Delta)e^{-\frac{(a+\Delta)^2}{2}} \leq 0$$

$$\iff (a+\Delta)e^{-\frac{(a+\Delta)^2}{2}} \leq ae^{-\frac{a^2}{2}}$$

$$\iff 1 + \frac{\Delta}{a} \leq e^{a\Delta + \frac{1}{2}\Delta^2} \tag{200}$$

The last statement is true because

$$e^{a\Delta + \frac{1}{2}\Delta^2} \geq 1 + a\Delta + \frac{1}{2}\Delta^2 \geq 1 + \frac{\Delta}{a} \tag{201}$$

when $a \geq 76\sqrt{d\log K} > 1$.

Since $f'(a) \leq 0$ for any $a \geq 76\sqrt{d\log K}$, we have

$$f(a) \leq f(76\sqrt{d\log K})$$

$$= \exp(-\frac{1}{2}(76\sqrt{d\log K})^2) - \exp(-\frac{1}{2}(76\sqrt{d\log K} + \Delta)^2)$$

$$= e^{-\frac{1}{2}(76\sqrt{d\log K})^2}(1 - e^{-76\sqrt{d\log K}\Delta - \frac{1}{2}\Delta^2})$$

$$\leq e^{-\frac{1}{2}(76\sqrt{d\log K})^2}(76\sqrt{d\log K}\Delta + \frac{1}{2}\Delta^2)$$

$$\leq e^{-\frac{1}{2}(76\sqrt{d\log K})^2} \cdot 84\sqrt{d\log K}\Delta \quad \text{(since } \Delta \leq 16\sqrt{d\log K}). \tag{202}$$

Which shows

$$\max_{a \geq 76\sqrt{d\log K}} e^{-\frac{a^2}{2}} - e^{-\frac{(a+\Delta)^2}{2}} \leq e^{-\frac{1}{2}(76\sqrt{d\log K})^2} \cdot 84\sqrt{d\log K}\Delta \tag{203}$$

Similarly

$$\max_{a \geq 92\sqrt{d\log K}} e^{-\frac{a^2}{2}} - e^{-\frac{(a+\Delta)^2}{2}} \leq e^{-\frac{1}{2}(92\sqrt{d\log K})^2} \cdot 100\sqrt{d\log K}\Delta \tag{204}$$

Therefore

$$\max_{x \in \Omega_1^*} \left| e^{-\frac{1}{2}\|x-u_i^*\|^2} - e^{-\frac{1}{2}\|x-u_i\|^2} \right| \leq e^{-\frac{1}{2}(76\sqrt{d\log K})^2} \cdot 84\sqrt{d\log K}\Delta \leq c_4 \min\{\|u_i^* - u_i\|, 1\} \tag{205}$$

where $c_4 = e^{-\frac{1}{2}(76\sqrt{d \log K})^2} \cdot 1344 d \log K$ (Since $\Delta \leq 16\sqrt{d \log K} \min\{\Delta, 1\}$). Notice that

$$
\begin{aligned}
c_4 &= e^{-\frac{1}{2}(76\sqrt{d \log K})^2} \cdot 1344 d \log K \\
&\leq e^{-\frac{1}{2}(76\sqrt{d \log K})^2} \cdot 1344 k^d K \\
&\leq e^{-\frac{1}{2}(76\sqrt{d \log K})^2} \cdot 1344 k^{2d} \\
&= 1344 e^{-2886 d \log K} \\
&\leq 1344 e^{-\frac{1}{2}(70\sqrt{d \log K})^2}
\end{aligned}
\tag{206}
$$

W.L.O.G., let $u_1^* = 0$, and define $u' = (50\sqrt{d \log K}, 0, \cdots, 0)$, then

$$
\begin{aligned}
\int_{\Omega_1^*} |P_i^* - P_i| \mathrm{d}x &= \int_{\Omega_1^*} (\frac{1}{\sqrt{2\pi}})^d |e^{-\frac{1}{2}\|x - u_i^*\|^2} - e^{-\frac{1}{2}\|x - u_i\|^2}| \mathrm{d}x \\
&\leq \int_{\Omega_1^*} (\frac{1}{\sqrt{2\pi}})^d \max_{x \in \Omega_1^*} |e^{-\frac{1}{2}\|x - u_i^*\|^2} - e^{-\frac{1}{2}\|x - u_i\|^2}| \mathrm{d}x \\
&\leq \int_{\Omega_1^*} (\frac{1}{\sqrt{2\pi}})^d 1344 e^{-\frac{1}{2}(70\sqrt{d \log K})^2} \min\{\|u_i^* - u_i\|, 1\} \mathrm{d}x \\
&= \min\{\|u_i^* - u_i\|, 1\} \int_{\Omega_1^*} (\frac{1}{\sqrt{2\pi}})^d 1344 e^{-\frac{1}{2}(70\sqrt{d \log K})^2} \mathrm{d}x \\
&\leq 1344 \min\{\|u_i^* - u_i\|, 1\} \int_{\Omega_1^*} (\frac{1}{\sqrt{2\pi}})^d e^{-\frac{1}{2}\|x - u'\|^2} \mathrm{d}x \\
&\leq 1344 \min\{\|u_i^* - u_i\|, 1\} \, \mathbb{P}(\mathcal{N}(u', I_d) \in \Omega_1^*) \\
&\leq 1344 \min\{\|u_i^* - u_i\|, 1\} \, \mathbb{P}(\|\mathcal{N}(u', I_d) - u'\| \geq 34\sqrt{d \log K}) \\
&\leq 1344 \min\{\|u_i^* - u_i\|, 1\} \cdot 2 \exp(-\frac{(34\sqrt{d \log K})^2}{16d}) \quad \text{(by Lemma C.2)} \\
&\leq 1344 \min\{\|u_i^* - u_i\|, 1\} \cdot 2 \exp(-70 \log K) \\
&= 2688 \min\{\|u_i^* - u_i\|, 1\} \left(\frac{1}{K}\right)^{70} \\
&\leq 2688 \left(\frac{1}{2}\right)^{69} \left(\frac{1}{K}\right) \min\{\|u_i^* - u_i\|, 1\}
\end{aligned}
\tag{207}
$$

$\square$

## C.2 BRACKETING NUMBER

We upper bound the bracketing number of $\mathcal{P}_{\mathcal{X}}(\mathcal{U})$ as follows.

**Lemma C.5.** *Let*

$$
\mathcal{P}_{\mathcal{X}}(\mathcal{U}) := \left\{ \sum_{i=1}^{K} \frac{1}{K} \mathcal{N}(u_i, I_d) \,\bigg|\, \mathbf{u} = \{u_i\}_{i=1}^K \in \mathcal{U} \right\}.
$$

*We assume there exists $D > 0$ such that for any $\mathbf{u} = \{u_i\}_{i=1}^K \in \mathcal{U}$, it holds that*

$$
\|u_i\|_2 \leq D\sqrt{d \log K}, \ \forall i \in [K].
$$

*Then the entropy can be bounded as follows,*

$$
\log N\big(\mathcal{P}_{\mathcal{X}}(\mathcal{U}), 1/m\big) \leq 2dK \log(6mdKD).
$$

*Proof of Lemma C.5.* First of all, we consider a set of standard Gaussian distribution

$$
\mathcal{P}_{\mathcal{X}}(\mathcal{A}) := \left\{ p_a(x) = \frac{1}{\sqrt{2\pi}} e^{-\frac{\|x - a\|_2^2}{2}} \,\bigg|\, a \in \mathcal{A} \right\},
$$

where $\mathcal{A} = \{a \in \mathbb{R}^d \mid \|a\|_2 \leq D\sqrt{d\log K}\}$. Our goal is to find a $1/m$-bracket $\mathcal{N}_{[\,]}(\mathcal{P}_\mathcal{X}(\mathcal{A}), 1/m)$ of $\mathcal{P}_\mathcal{X}(\mathcal{A})$. In other words, for any $p_a(x) \in \mathcal{P}_\mathcal{X}(\mathcal{A})$, we need to define $\bar{p}_a(x) \in \mathcal{N}_{[\,]}(\mathcal{P}_\mathcal{X}(\mathcal{A}), 1/m)$ such that

- $\bar{p}_a(x) \geq p_a(x),\ \forall x \in \mathbb{R}^d$

- $\int |\bar{p}_a(x) - p_a(x)|\, dx \leq 1/m$.

We consider $\bar{p}_a(x)$ of the form

$$\bar{p}_a(x) = \frac{1}{\sqrt{2\pi}} e^{-\frac{c_1\|x-\bar{a}\|_2^2}{2} + c_2}.$$

We then specify $\bar{a} \in \mathbb{R}^d$, $c_1 \in \mathbb{R}$ and $c_2 \in \mathbb{R}$. Let $a = (a_1, \ldots, a_d)$ and $\epsilon > 0$ be a parameter that will be chosen later. If $a_i \in [k\epsilon, (k+1)\epsilon)$ for some $k \in \mathbb{Z}$, we define $\bar{a}_i := k\epsilon$ and $\bar{a} := (\bar{a}_1, \ldots, \bar{a}_d)$, which implies

$$\|a - \bar{a}\|_2^2 \leq d\epsilon^2. \tag{208}$$

Note that $\bar{p}_a(x) \geq p_a(x)$ holds for any $x \in \mathbb{R}^d$ if and only if

$$(c_1 - 1)\left\| x + \frac{a - c_1\bar{a}}{c_1 - 1}\right\|_2^2 + \frac{c_1}{1 - c_1}\|a - \bar{a}\|_2^2 \leq 2c_2,\ \forall x \in \mathbb{R}^d.$$

Let $c_1 = 1 - \epsilon$. Then, we have $\bar{p}_a(x) \geq p_a(x)$ if and only if

$$-\epsilon\left\| x + \frac{a - c_1\bar{a}}{c_1 - 1}\right\|_2^2 + \frac{1 - \epsilon}{\epsilon}\|a - \bar{a}\|_2^2 \leq 2c_2,\ \forall x \in \mathbb{R}^d.$$

Note that

$$-\epsilon\left\| x + \frac{a - c_1\bar{a}}{c_1 - 1}\right\|_2^2 + \frac{1 - \epsilon}{\epsilon}\|a - \bar{a}\|_2^2 \leq \frac{1 - \epsilon}{\epsilon}\|a - \bar{a}\|_2^2 \leq d(1 - \epsilon)\epsilon,$$

where the last inequality follows from (208). Thus, by choosing $c_2 = d(1 - \epsilon)\epsilon/2$, we obtain $\bar{p}_a(x) \geq p_a(x)$ for any $x \in \mathbb{R}^d$. Note that

$$\int |\bar{p}_a(x) - p_a(x)|\, dx = \frac{1}{\sqrt{c_1}} \cdot e^{c_2} - 1 = \frac{e^{\frac{d(1-\epsilon)\epsilon}{2}}}{\sqrt{1 - \epsilon}} - 1 \leq \big(1 + d(1 - \epsilon)\epsilon\big) \cdot (1 + \epsilon) - 1 \leq (1 + 2d)\epsilon.$$

Here the first inequality follows from the fact that $e^x \leq 1 + 2x$ and $\frac{1}{\sqrt{1-x}} \leq 1 + x$ for any $0 < x < 1/2$. Let $(1 + 2d)\epsilon = m^{-1}$. It then holds that

$$\int |\bar{p}_a(x) - p_a(x)|\, dx \leq (1 + 2d)\epsilon = \frac{1}{m}.$$

Recall that for any $a \in \mathcal{A}$, it holds that $\|a\|_2 \leq D\sqrt{d\log K}$. Thus, we have

$$N_{[\,]}(\mathcal{P}_\mathcal{X}(\mathcal{A}), 1/m) \leq \left(\frac{2D\sqrt{d\log K}}{\epsilon}\right)^d = \left(2mD(1 + 2d)\sqrt{d\log K}\right)^d.$$

Then, we consider a set of Gaussian mixture model

$$\mathcal{P}_\mathcal{X}(\mathcal{U}) := \left\{ \sum_{i=1}^K \frac{1}{K}\mathcal{N}(u_i, I_d) \ \middle|\ \mathbf{u} = \{u_i\}_{i=1}^K \in \mathcal{U} \right\},$$

where $\mathcal{U} = \{\{u_i\}_{i=1}^K \mid \|u_i\|_2 \leq D\sqrt{d\log K}, \forall i \in [K]\}$. Our goal is to find a $1/m$-bracket $\mathcal{N}(\mathcal{P}_\mathcal{X}(\mathcal{U}), 1/m)$ of $\mathcal{P}_\mathcal{X}(\mathcal{U})$. For any $p_\mathbf{u}(x) \in \mathcal{P}_\mathcal{X}(\mathcal{U})$, it holds that

$$p_\mathbf{u}(x) = \sum_{i=1}^K \frac{1}{K}p_{u_i}(x),$$

where $p_{u_i}(x) \in \mathcal{P}_\mathcal{X}(\mathcal{A})$. Note that for any $i \in [K]$, there exists $\bar{p}_{u_i}(x) \in \mathcal{N}_{[\,]}(\mathcal{P}_\mathcal{X}(\mathcal{A}), 1/m)$, such that

- $\bar{p}_{u_i}(x) \geq p_{u_i}(x), \; \forall x \in \mathbb{R}^d$

- $\int |\bar{p}_{u_i}(x) - p_{u_i}(x)| \, dx \leq 1/m.$

We define

$$\bar{p}_{\mathbf{u}}(x) = \sum_{i=1}^{K} \frac{1}{K} \bar{p}_{u_i}(x).$$

It then holds that

$$\bar{p}_{\mathbf{u}}(x) = \sum_{i=1}^{K} \frac{1}{K} \bar{p}_{u_i}(x) \geq \sum_{i=1}^{K} \frac{1}{K} p_{u_i}(x) = p_{\mathbf{u}}(x), \; \forall x \in \mathbb{R}^d$$

and

$$\int |\bar{p}_{\mathbf{u}}(x) - p_{\mathbf{u}}(x)| \, dx \leq \sum_{i=1}^{K} \frac{1}{K} \int |\bar{p}_{u_i}(x) - p_{u_i}(x)| \, dx \leq \sum_{i=1}^{K} \frac{1}{mK} = \frac{1}{m}.$$

Thus, we obtain that

$$N_{[\,]}(\mathcal{P}_{\mathcal{X}}(\mathcal{U}), 1/m) \leq \left( N_{[\,]}(\mathcal{P}_{\mathcal{X}}(\mathcal{A}), 1/m) \right)^K \leq \left( 2mD(1 + 2d)\sqrt{d \log K} \right)^{dK},$$

which implies that

$$\log N_{[\,]}(\mathcal{P}_{\mathcal{X}}(\mathcal{U}), 1/m) \leq dK \log \left( 2mD(1 + 2d)\sqrt{d \log K} \right) \leq 2dK \log(6mdKD).$$

$\square$

## C.3 RADEMACHER COMPLEXITY

Given labeled data $\{x_j, y_j\}_{j=1}^n$ and the pretrained $\hat{\mathbf{u}}$, the function class

$$\left\{ \left( \mathbb{1}_{g_{\hat{\mathbf{u}},\psi}(x_1) \neq y_1}, \dots, \mathbb{1}_{g_{\hat{\mathbf{u}},\psi}(x_n) \neq y_n} \right) \,\middle|\, \psi \in \Psi \right\}$$

is a finite function class, whose Rademacher complexity can be bounded by the following lemma.

**Lemma C.6.** *Let $A = \{a^1, \dots, a^N\}$ be a finite set of vectors in $\mathbb{R}^n$. Then, the Rademacher complexity can be bounded as follows,*

$$R_n(A) \leq \max_{a \in A} \|a\|_2 \cdot \frac{2\sqrt{2 \log N}}{n}.$$

*Proof.* Note that for any $\lambda > 0$

$$R_n(A) = \mathbb{E}\left[ \sup_{a \in A} \frac{2}{n} \sum_{i=1}^{n} \sigma_i a_i \right] \leq \frac{1}{\lambda} \log \mathbb{E}\left[ e^{\sup_{a \in A} \frac{2\lambda}{n} \sum_{i=1}^{n} \sigma_i a_i} \right]$$

$$\leq \frac{1}{\lambda} \log \sum_{a \in A} \mathbb{E}\left[ e^{\frac{2\lambda}{n} \sum_{i=1}^{n} \sigma_i a_i} \right] = \frac{1}{\lambda} \log \sum_{a \in A} \prod_{i=1}^{n} \mathbb{E}\left[ e^{\frac{2\lambda}{n} \sigma_i a_i} \right], \tag{209}$$

where the first inequality follows from Jensen's inequality. Recall that $\sigma_i$ is a Rademacher random variable. Thus, we have

$$\mathbb{E}\left[ e^{\frac{2\lambda}{n} \sigma_i a_i} \right] = \frac{1}{2} e^{\frac{2\lambda}{n} a_i} + \frac{1}{2} e^{-\frac{2\lambda}{n} a_i} \leq e^{\frac{2\lambda^2 a_i^2}{n^2}}, \tag{210}$$

where the last inequality follows from the fact that $(e^x + e^{-x})/2 \leq e^{x^2/2}$. By (209) and (210), we have

$$R_n(A) \leq \frac{1}{\lambda} \log \sum_{a \in A} e^{\frac{2\lambda^2 \|a\|^2}{n^2}} \leq \frac{1}{\lambda} \log |A| e^{\frac{2\lambda^2}{n^2} \cdot \max_{a \in A} \|a\|^2} = \frac{1}{\lambda} \log N + \frac{2\lambda}{n^2} \cdot \max_{a \in A} \|a\|^2. \tag{211}$$

Let $\lambda = \sqrt{n \log N / 2 \max_{a \in A} \|a\|^2}$. We obtain that

$$R_n(A) \leq \max_{a \in A} \|a\| \cdot \frac{2\sqrt{2 \log N}}{n}.$$

$\square$

## C.4 PROOFS FOR THEOREM 5.3

In the sequel, we prove Theorem 5.3.

*Proof.* Let $\Phi = \mathcal{U}$ and $\Psi$ be the set of $2^K$ classifications. Recall that the loss function is defined as $\ell(x, y) = \mathbb{1}_{\{x \neq y\}}$, which is upper bound by 1. Let $m = \tilde{\Omega}(dK^3)$. By Theorem 3.3 and Lemma C.5, it holds that

$$d_{\mathrm{TV}}\big(\mathbb{P}_{\hat{\phi}}(x), \mathbb{P}_{\phi^*}(x)\big) \lesssim \sqrt{\frac{1}{m} \log \frac{N_{[\,]}(\mathcal{P}_{\mathcal{X}}(\Phi), 1/m)}{\delta}} \lesssim \sqrt{\frac{dK}{m} \log \frac{mdKD}{\delta}} \lesssim \frac{1}{K}.$$

Then, by Lemma 5.2, Assumption 3.2 holds for Gaussian mixture models. By Theorem 3.4, with probability at least $1 - \delta$, we have the following excess risk bound,

$$\mathrm{Error}_\ell(\hat{\phi}, \hat{\psi}) \leq 2 \max_{\phi \in \Phi} R_n(\ell \circ \mathcal{G}_{\phi, \Psi}) + \sqrt{\frac{2}{n} \log \frac{4}{\delta}} + 12\kappa \cdot \sqrt{\frac{1}{m} \log \frac{2N(\mathcal{P}_{\mathcal{X}}(\Phi), 1/m)}{\delta}},$$

where $\kappa = c_2$ is some absolute constants that represents the transferability of the model. By Lemma C.5, we further have

$$\mathrm{Error}_\ell(\hat{\phi}, \hat{\psi}) \leq 2 \max_{\phi \in \Phi} R_n(\ell \circ \mathcal{G}_{\phi, \Psi}) + \sqrt{\frac{2}{n} \log \frac{4}{\delta}} + 12\kappa \cdot \sqrt{\frac{2dK}{m} \log \frac{12mdKD}{\delta}}. \tag{212}$$

For any $\phi \in \Phi$, we have

$$R_n(\ell \circ \mathcal{G}_{\phi, \Psi}) = \mathbb{E}\left[ \sup_{\psi \in \Psi} \frac{1}{n} \sum_{i=1}^n \sigma_i \mathbb{1}_{\{g_{\phi, \psi}(x_i) \neq y_i\}} \right]. \tag{213}$$

Note that $|\Psi| = 2^K$. By Lemma C.6, it holds for any $\phi \in \Phi$ that

$$R_n(\ell \circ \mathcal{G}_{\phi, \Psi}) \leq \sqrt{n} \cdot \frac{2\sqrt{2 \log 2^K}}{n} = 2\sqrt{\frac{2K \log 2}{n}}. \tag{214}$$

By (212) and (214), we have

$$\mathrm{Error}_\ell(\hat{\phi}, \hat{\psi}) \leq 4\sqrt{\frac{2K \log 2}{n}} + \sqrt{\frac{2}{n} \log \frac{4}{\delta}} + 12\kappa \cdot \sqrt{\frac{2dK}{m} \log \frac{12mdKD}{\delta}}$$

$$= \mathcal{O}\left( \sqrt{\frac{K \log \frac{1}{\delta}}{n}} + \kappa \sqrt{\frac{dK \log \frac{mdKD}{\delta}}{m}} \right)$$

$$= \tilde{\mathcal{O}}\left( \sqrt{\frac{K}{n}} + \kappa \sqrt{\frac{dK}{m}} \right),$$

where $\kappa = c_2$ is some absolute constants that represents the transferability of the model. $\square$

Thus, we prove Theorem 5.3.

## D PROOFS FOR SECTION 6

In Section D.1, we show that contrastive learning with linear regression as downstream tasks is $\kappa^{-1}$-weakly-informative by proving Lemma 6.1. In Section D.2, we prove Theorem 6.2.

### D.1 PROOFS FOR LEMMA 6.1

Recall that in the setting of contrastive learning, we assume that $x$ and $x'$ are sampled independently from the same distribution $\mathbb{P}(x)$. And we assume the label $t$ that captures the similarity between $x$ and $x'$ satisfies

$$\mathbb{P}(t = 1 \,|\, x, x') = \frac{1}{1 + e^{-f_{\theta^*}(x)^T f_{\theta^*}(x')}},$$

$$\mathbb{P}(t = -1 \,|\, x, x') = \frac{1}{1 + e^{f_{\theta^*}(x)^T f_{\theta^*}(x')}}.$$

Lemma 6.1 directly follows from the following lemma.

**Lemma D.1.** *There exists $O \in \mathbb{R}^{r \times r}$, $O^T O = OO^T = I_r$ such that*

$$d_{\mathrm{TV}}\big(\mathbb{P}_{Of_\theta}(x,z), \mathbb{P}_{f_{\theta^*}}(x,z)\big) \leq c \cdot \sqrt{\frac{1}{\sigma_{\min}(\mathbb{E}[f_{\theta^*}(x)f_{\theta^*}(x)^T])}} \cdot H\big(\mathbb{P}_{f_\theta}(x,x',t), \mathbb{P}_{f_{\theta^*}}(x,x',t)\big).$$

*Here $c$ is some absolute constants.*

We first prove the following lemma, which is the core of the proof of Lemma D.1.

**Lemma D.2.** *Suppose that $\mathbb{E}[f_\theta(x)f_{\theta^*}(x)^T] = \mathbb{E}[f_{\theta^*}(x)f_\theta(x)^T]$ are positive semi-definite matrices. Then we have*

$$\mathbb{E}\big[\big(f_\theta(x)^T f_\theta(x') - f_{\theta^*}(x)^T f_{\theta^*}(x')\big)^2\big] \geq (2\sqrt{2}-2)\sigma_{\min}\big(\mathbb{E}[f_{\theta^*}(x)f_{\theta^*}(x)^T]\big) \cdot \mathbb{E}[\|f_{\theta^*}(x) - f_\theta(x)\|_2^2].$$

*Proof of Lemma D.2.* For notation simplicity, we denote $\Delta(x) := f_{\theta^*}(x) - f_\theta(x)$. It then holds that

$$\mathbb{E}\big[\big(f_\theta(x)^T f_\theta(x') - f_{\theta^*}(x)^T f_{\theta^*}(x')\big)^2\big]$$
$$= \mathbb{E}\big[\big(f_{\theta^*}(x)^T \Delta(x') + \Delta(x)^T f_{\theta^*}(x') - \Delta(x)^T \Delta(x')\big)^2\big]$$
$$= \mathbb{E}\big[\big(\Delta(x)^T \Delta(x')\big)^2 - 2\sqrt{2}\Delta(x)^T \Delta(x') f_{\theta^*}(x')^T \Delta(x) + 2f_{\theta^*}(x)^T \Delta(x') f_{\theta^*}(x')^T \Delta(x)\big]$$
$$+ (4 - 2\sqrt{2})\mathbb{E}[f_\theta(x')^T \Delta(x)\Delta(x)^T f_{\theta^*}(x')] + (2\sqrt{2} - 2)\mathbb{E}[f_{\theta^*}(x')^T \Delta(x)\Delta(x)^T f_{\theta^*}(x')]. \tag{215}$$

For the first term of (215), we have

$$\mathbb{E}\big[\big(\Delta(x)^T \Delta(x')\big)^2 - 2\sqrt{2}\Delta(x)^T \Delta(x') f_{\theta^*}(x')^T \Delta(x) + 2f_{\theta^*}(x)^T \Delta(x') f_{\theta^*}(x')^T \Delta(x)\big]$$
$$= \mathrm{Tr}\Big(\mathbb{E}[\Delta(x')\Delta(x')^T \Delta(x)\Delta(x)^T - 2\sqrt{2}\Delta(x')f_{\theta^*}(x')^T \Delta(x)\Delta(x)^T + 2\Delta(x')f_{\theta^*}(x')^T \Delta(x)f_{\theta^*}(x)^T]\Big)$$
$$= \mathrm{Tr}\Big(\big(\mathbb{E}[\Delta(x)\Delta(x)^T]\big)^2 - 2\sqrt{2}\mathbb{E}[\Delta(x)f_{\theta^*}(x)^T] \cdot \mathbb{E}[\Delta(x)\Delta(x)^T] + 2\big(\mathbb{E}[\Delta(x)f_{\theta^*}(x)^T]\big)^2\Big)$$
$$= \mathrm{Tr}\Big(\big(\mathbb{E}[\Delta(x)\Delta(x)^T] - \sqrt{2}\mathbb{E}[\Delta(x)f_{\theta^*}(x)^T]\big)^2\Big), \tag{216}$$

where the second equation follows from our assumption that $x, x'$ are i.i.d. Note that $\mathbb{E}[f_\theta(x)f_{\theta^*}(x)^T] = \mathbb{E}[f_{\theta^*}(x)f_\theta(x)^T]$. Thus, we obtain

$$\Big(\mathbb{E}[\Delta(x)\Delta(x)^T] - \sqrt{2}\mathbb{E}[\Delta(x)f_{\theta^*}(x)^T]\Big)^T = \mathbb{E}[\Delta(x)\Delta(x)^T] - \sqrt{2}\mathbb{E}[f_{\theta^*}(x)\Delta(x)^T]$$
$$= \mathbb{E}[\Delta(x)\Delta(x)^T] - \sqrt{2}\mathbb{E}[\Delta(x)f_{\theta^*}(x)^T], \tag{217}$$

which implies that $\mathbb{E}[\Delta(x)\Delta(x)^T] - \sqrt{2}\mathbb{E}[\Delta(x)f_{\theta^*}(x)^T]$ is symmetric. It then holds that

$$\mathbb{E}\big[\big(\Delta(x)^T \Delta(x')\big)^2 - 2\sqrt{2}\Delta(x)^T \Delta(x') f_{\theta^*}(x')\Delta(x) + 2f_{\theta^*}(x)^T \Delta(x') f_{\theta^*}(x')^T \Delta(x)\big]$$
$$= \mathrm{Tr}\Big(\big(\mathbb{E}[\Delta(x)\Delta(x)^T] - \sqrt{2}\mathbb{E}[\Delta(x)f_{\theta^*}(x)^T]\big)^2\Big) \geq 0. \tag{218}$$

For the second term of (215), we have

$$\mathbb{E}[f_\theta(x')^T \Delta(x)\Delta(x)^T f_{\theta^*}(x')] = \mathrm{Tr}\Big(\mathbb{E}[f_{\theta^*}(x')f_\theta(x')^T] \cdot \mathbb{E}[\Delta(x)\Delta(x)^T]\Big) \geq 0, \tag{219}$$

where the inequality follows from the fact $\mathbb{E}[f_{\theta^*}(x')f_\theta(x')^T] \succcurlyeq 0$ and $\mathbb{E}[\Delta(x)\Delta(x)^T] \succcurlyeq 0$.

For the third term of (215), we have

$$\mathbb{E}[f_{\theta^*}(x')^T \Delta(x)\Delta(x)^T f_{\theta^*}(x')] = \mathrm{Tr}\Big(\mathbb{E}[f_{\theta^*}(x)f_{\theta^*}(x)^T] \cdot \mathbb{E}[\Delta(x)\Delta(x)^T]\Big)$$
$$\geq \sigma_{\min}\big(\mathbb{E}[f_{\theta^*}(x)f_{\theta^*}(x)^T]\big)\mathrm{Tr}\Big(\mathbb{E}[\Delta(x)\Delta(x)^T]\Big)$$
$$= \sigma_{\min}\big(\mathbb{E}[f_{\theta^*}(x)f_{\theta^*}(x)^T]\big)\mathbb{E}[\|\Delta(x)\|_2^2]. \tag{220}$$

Combining (215), (218), (219) and (220), we have

$$\mathbb{E}\big[\big(f_\theta(x)^T f_\theta(x') - f_{\theta^*}(x)^T f_{\theta^*}(x')\big)^2\big] \geq (2\sqrt{2} - 2)\sigma_{\min}\big(\mathbb{E}[f_{\theta^*}(x)f_{\theta^*}(x)^T]\big)\mathbb{E}[\|\Delta(x)\|_2^2] \tag{221}$$

$\square$

With Lemma D.2, we prove Lemma D.1 in the following.

*Proof of Lemma D.1.* We consider the singular value decomposition (SVD) of $\mathbb{E}[f_\theta(x)f_{\theta^*}(x)^T] = U_1\Sigma_1 V_1^T$ and $\mathbb{E}[f_{\theta^*}(x)f_\theta(x)^T] = (\mathbb{E}[f_\theta(x)f_{\theta^*}(x)^T])^T = V_1\Sigma_1 U_1^T$. We define $O := V_1 U_1^T \in \mathbb{R}^{r\times r}$, which satisfies $O^T O = OO^T = I_r$. It then holds that

$$\mathbb{E}[Of_\theta(x)f_{\theta^*}(x)^T] = \mathbb{E}\big[f_{\theta^*}(x)\big(Of_\theta(x)\big)^T\big] = V_1\Sigma_1 V_1^T, \tag{222}$$

which are positive semi-definite matrices. By Lemma D.2, we have

$$\mathbb{E}\big[\big(f_\theta(x)^T f_\theta(x') - f_{\theta^*}(x)^T f_{\theta^*}(x')\big)^2\big]$$
$$\geq (2\sqrt{2}-2)\sigma_{\min}\big(\mathbb{E}[f_{\theta^*}(x)f_{\theta^*}(x)^T]\big) \cdot \mathbb{E}[\|f_{\theta^*}(x) - Of_\theta(x)\|_2^2]. \tag{223}$$

For Hellinger distance, we have

$$2H^2\big(\mathbb{P}_{f_\theta}(x, x', t), \mathbb{P}_{f_{\theta^*}}(x, x', t)\big)$$
$$= \int \left(\sqrt{p_{f_\theta}(x, x', t)} - \sqrt{p_{f_{\theta^*}}(x, x', t)}\right)^2 dt dx dx'$$
$$= \int \left(\sqrt{p_{f_\theta}(t = 1 \mid x, x')} - \sqrt{p_{f_{\theta^*}}(t = 1 \mid x, x')}\right)^2 p(x, x')\, dx dx'$$
$$+ \int \left(\sqrt{p_{f_\theta}(t = 0 \mid x, x')} - \sqrt{p_{f_{\theta^*}}(t = 0 \mid x, x')}\right)^2 p(x, x')\, dx dx' \tag{224}$$

For the first term of (224), we have

$$\int \left(\sqrt{p_{f_\theta}(t = 1 \mid x, x')} - \sqrt{p_{f_{\theta^*}}(t = 1 \mid x, x')}\right)^2 p(x, x')\, dx dx'$$
$$= \int \left(\sqrt{h\big(f_\theta(x)^T f_\theta(x')\big)} - \sqrt{h\big(f_{\theta^*}(x)^T f_{\theta^*}(x')\big)}\right)^2 p(x, x')\, dx dx', \tag{225}$$

where

$$h(a) := \frac{1}{1 + e^{-a}}. \tag{226}$$

By Cauchy-Schwartz inequality, we have $|f_\theta(x)^T f_\theta(x')| \leq \|f_\theta(x)\|_2 \|f_\theta(x')\|_2 \leq 1$. Note that for any $a, b \in [-1, 1]$, we have

$$\left(\sqrt{h(a)} - \sqrt{h(b)}\right)^2$$
$$= \frac{\big(h(a) - h(b)\big)^2}{\left(\sqrt{h(a)} + \sqrt{h(b)}\right)^2} \geq \frac{1}{4}\big(h(a) - h(b)\big)^2 = \frac{1}{4}h'(\xi)^2(a - b)^2 \geq \frac{1}{2 + e + e^{-1}}(a - b)^2. \tag{227}$$

Thus, it holds that

$$\int \left(\sqrt{p_{f_\theta}(t = 1 \mid x, x')} - \sqrt{p_{f_{\theta^*}}(t = 1 \mid x, x')}\right)^2 p(x, x')\, dx dx'$$
$$\geq \frac{1}{2 + e + e^{-1}} \int \big(f_\theta(x)^T f_\theta(x') - f_{\theta^*}(x)^T f_{\theta^*}(x')\big)^2 p(x, x')\, dx dx'$$
$$= \frac{1}{2 + e + e^{-1}}\mathbb{E}\big[\big(f_\theta(x)^T f_\theta(x') - f_{\theta^*}(x)^T f_{\theta^*}(x')\big)^2\big]. \tag{228}$$

Similarly, For the second term of (224), we have

$$\int \left(\sqrt{p_{f_\theta}(t = 0 \mid x, x')} - \sqrt{p_{f_{\theta^*}}(t = 0 \mid x, x')}\right)^2 p(x, x')\, dx dx'$$
$$\geq \frac{1}{2 + e + e^{-1}}\mathbb{E}\big[\big(f_\theta(x)^T f_\theta(x') - f_{\theta^*}(x)^T f_{\theta^*}(x')\big)^2\big] \tag{229}$$

Combining (224), (228) and (229), we have

$$H^2\big(\mathbb{P}_{f_\theta}(x, x', t), \mathbb{P}_{f_{\theta^*}}(x, x', t)\big) \geq \frac{1}{2 + e + e^{-1}}\mathbb{E}\big[\big(f_\theta(x)^T f_\theta(x') - f_{\theta^*}(x)^T f_{\theta^*}(x')\big)^2\big]. \quad (230)$$

We choose $O \in \mathbb{R}^{r \times r}$ that satisfies (223). For the TV distance, we have

$$d_{\mathrm{TV}}\big(\mathbb{P}_{Of_\theta}(x, z), \mathbb{P}_{f_{\theta^*}}(x, z)\big) = \frac{1}{2}\int |p_{Of_\theta}(z\,|\,x) - p_{f_{\theta^*}}(z\,|\,x)|p(x)\,dx. \quad (231)$$

Note that $z\,|\,x \sim \mathcal{N}(f_\theta(x), I_r)$. By Lemma B.1, we have

$$\begin{aligned}
d_{\mathrm{TV}}\big(\mathbb{P}_{Of_\theta}(x, z), \mathbb{P}_{f_{\theta^*}}(x, z)\big) &= \frac{1}{2}\int |p_{Of_\theta}(z\,|\,x) - p_{f_{\theta^*}}(z\,|\,x)|p(x)\,dx \\
&\leq \frac{1}{2}\int \min\{1, \|Of_\theta(x) - f_{\theta^*}(x)\|_2\}p(x)\,dx \\
&\leq \frac{1}{2}\min\left\{1, \int \|Of_\theta(x) - f_{\theta^*}(x)\|_2 p(x)\,dx\right\} \\
&= \frac{1}{2}\min\big\{1, \mathbb{E}[\|Of_\theta(x) - f_{\theta^*}(x)\|_2]\big\}. \quad (232)
\end{aligned}$$

Combining (223), (230) and (232), we show that

$$\begin{aligned}
&d_{\mathrm{TV}}\big(\mathbb{P}_{Of_\theta}(x, z), \mathbb{P}_{f_{\theta^*}}(x, z)\big) \\
&\leq \frac{1}{2}\mathbb{E}[\|Of_\theta(x) - f_{\theta^*}(x)\|_2] \\
&\leq \frac{1}{2}\sqrt{\mathbb{E}[\|Of_\theta(x) - f_{\theta^*}(x)\|_2^2]} \\
&\leq \frac{1}{2}\sqrt{\frac{1}{(2\sqrt{2} - 2)\sigma_{\min}\big(\mathbb{E}[f_{\theta^*}(x)f_{\theta^*}(x)^T]\big)}\mathbb{E}\big[\big(f_\theta(x)^T f_\theta(x') - f_{\theta^*}(x)^T f_{\theta^*}(x')\big)^2\big]} \\
&\leq \frac{1}{2}\sqrt{\frac{2 + e + e^{-1}}{(2\sqrt{2} - 2)\sigma_{\min}\big(\mathbb{E}[f_{\theta^*}(x)f_{\theta^*}(x)^T]\big)}H\big(\mathbb{P}_{f_\theta}(x, x', t), \mathbb{P}_{f_{\theta^*}}(x, x', t)\big)}. \quad (233)
\end{aligned}$$

Thus, we prove Lemma D.1. $\qquad\square$

Lemma D.1 directly implies Lemma 6.1.

*Proof of Lemma 6.1.* For any $\theta \in \Theta$, we choose $O \in \mathbb{R}^{r \times r}$ that satisfies Lemma D.1. It then holds that

$$\begin{aligned}
d_{\mathrm{TV}}\big(\mathbb{P}_{f_\theta, O^T\beta^*}(x, y), \mathbb{P}_{f_{\theta^*}, \beta^*}(x, y)\big) &= d_{\mathrm{TV}}\big(\mathbb{P}_{Of_\theta, \beta^*}(x, y), \mathbb{P}_{f_{\theta^*}, \beta^*}(x, y)\big) \\
&\leq d_{\mathrm{TV}}\big(\mathbb{P}_{Of_\theta}(x, z), \mathbb{P}_{f_{\theta^*}}(x, z)\big) \\
&\leq c \cdot \sqrt{\frac{1}{\sigma_{\min}(\mathbb{E}[f_{\theta^*}(x)f_{\theta^*}(x)^T])}} \cdot H\big(\mathbb{P}_{f_\theta}(x, x', t), \mathbb{P}_{f_{\theta^*}}(x, x', t)\big).
\end{aligned}$$

Thus, we prove that the model is $\kappa^{-1}$-weakly-informative, where

$$\kappa = c \cdot \sqrt{\frac{1}{\sigma_{\min}(\mathbb{E}[f_{\theta^*}(x)f_{\theta^*}(x)^T])}}. \quad (234)$$

Here $c$ is some absolute constants.

$\qquad\square$

## D.2 Proofs for Theorem 6.2

In this section, we prove Theorem 6.2. Suppose that $\hat{\theta}, \hat{\beta}$ are the outputs of Algorithm 1. Let $\ell$ be the squared loss and $\tilde{\ell}$ be its truncation with truncation level $L$. The optimal predictor defined in (1) has the following closed form solution

$$g_{\theta,\beta}(x) = \mathbb{E}_{\theta,\beta}[y \mid x] = \beta^T f_\theta(x). \tag{235}$$

We have the following guarantees.

**Lemma D.3.** *Let the truncation level $L = 36(D^2 + 1) \log n$. It then holds that*

$$\sup_{\theta,\beta} \left\{ \mathbb{E}_{\theta^*,\beta^*} \left[ \ell\big(g_{\theta,\beta}(x), y\big) \right] - \mathbb{E}_{\theta^*,\beta^*} \left[ \tilde{\ell}\big(g_{\theta,\beta}(x), y\big) \right] \right\} \leq \sqrt{\frac{18(D^2 + 1) \log n}{\pi n}}. \tag{236}$$

*Proof of Lemma D.3.* Note that

$$\big(g_{\theta,\beta}(x) - y\big)\big|x = \big(\beta^T f_\theta(x) - y\big)\big|x \sim \mathcal{N}\big(\beta^T f_\theta(x) - \beta^{*T} f_{\theta^*}(x), 1\big) \tag{237}$$

We denote by $c(x) := \beta^T f_\theta(x) - \beta^{*T} f_{\theta^*}(x)$. It holds that $|c(x)| \leq 2D$. Thus, it holds for any $\theta, \beta$ that

$$
\begin{aligned}
& \mathbb{E}_{\theta^*,\beta^*} \left[ \ell\big(g_{\theta,\beta}(x), y\big) - \tilde{\ell}\big(g_{\theta,\beta}(x), y\big) \,\big|\, x \right] \\
&= \mathbb{E}_{\theta^*,\beta^*} \left[ \Big( \big(g_{\theta,\beta}(x) - y\big)^2 - L \Big) \mathbb{1}_{\{(g_{\theta,\beta}(x) - y)^2 > L\}} \,\big|\, x \right] \\
&= \int_{\sqrt{L}}^{+\infty} (u^2 - L) \cdot \frac{1}{\sqrt{2\pi}} e^{-\frac{(u - c(x))^2}{2}} \, du \\
&= \int_{\sqrt{L} - c(x)}^{+\infty} \big((u + c(x))^2 - L\big) \cdot \frac{1}{\sqrt{2\pi}} e^{-\frac{u^2}{2}} \, du \\
&= \frac{\sqrt{L} + c(x)}{\sqrt{2\pi}} e^{-\frac{\left(\sqrt{L} - c(x)\right)^2}{2}} + \frac{1 + c(x)^2 - L}{\sqrt{2\pi}} \int_{\sqrt{L} - c(x)}^{+\infty} e^{-\frac{u^2}{2}} \, du \\
&\leq \frac{\sqrt{L} + c(x)}{\sqrt{2\pi}} e^{-\frac{\left(\sqrt{L} - c(x)\right)^2}{2}} \qquad (L \geq 4D^2 + 1 \geq c(x)^2 + 1) \\
&\leq \frac{2(\sqrt{L} - c(x))}{\sqrt{2\pi}} e^{-\frac{\left(\sqrt{L} - c(x)\right)^2}{2}} \qquad (L \geq 36D^2 \geq (3c(x))^2) \\
&\leq \frac{2(\sqrt{L} - 2D)}{\sqrt{2\pi}} e^{-\frac{\left(\sqrt{L} - 2D\right)^2}{2}} \\
&\leq \frac{\sqrt{L}}{\sqrt{2\pi}} e^{-\frac{L}{8}} \qquad (\sqrt{L} - 2D \geq \frac{\sqrt{L}}{2})
\end{aligned} \tag{238}
$$

As a result, we show that

$$
\begin{aligned}
& \sup_{\theta,\beta} \left\{ \mathbb{E}_{\theta^*,\beta^*} \left[ \ell\big(g_{\theta,\beta}(x), y\big) \right] - \mathbb{E}_{\theta^*,\beta^*} \left[ \tilde{\ell}\big(g_{\theta,\beta}(x), y\big) \right] \right\} \\
&\leq \mathbb{E}_{\theta^*,\beta^*} \left[ \sup_{\theta,\beta} \mathbb{E}_{\theta^*,\beta^*} \left[ \ell\big(g_{\theta,\beta}(x), y\big) - \tilde{\ell}\big(g_{\theta,\beta}(x), y\big) \,\big|\, x \right] \right] \\
&\leq \frac{\sqrt{L}}{\sqrt{2\pi}} e^{-\frac{L}{8}} \\
&\leq \sqrt{\frac{18(D^2 + 1) \log n}{\pi n}}. \qquad (L = 36(D^2 + 1) \log n)
\end{aligned} \tag{239}
$$

$\square$

**Lemma D.4.** *Suppose that $\hat{\theta}, \hat{\beta}$ are the outputs of Algorithm 1. Let $\tilde{\ell}$ be the truncated squared loss with truncation level $L$. Then there exists an absolute constant $c$ such that with probability at least $1 - \delta$ that*

$$\mathbb{E}_{\theta^*, \beta^*}\left[\tilde{\ell}\big(g_{\hat{\theta}, \hat{\beta}}(x), y\big)\right] - \mathbb{E}_{\theta^*, \beta^*}\left[\tilde{\ell}\big(g_{\theta^*, \beta^*}(x), y\big)\right]$$

$$\leq c\kappa L \cdot \sqrt{\frac{1}{m} \log \frac{N_{[\,]}\big(\mathcal{P}_{\mathcal{X} \times \mathcal{S}}(\mathcal{F}_\theta), 1/m^2\big)}{\delta}} + cL\sqrt{\frac{\log 1/\delta}{n}} + c\sqrt{L} \sup_{\theta \in \Theta} R_n(\mathcal{G}_{\theta, \mathcal{B}}), \tag{240}$$

*where*

$$\kappa = c_3 \sqrt{\frac{1}{\sigma_{\min}\big(\mathbb{E}[f_{\theta^*}(x) f_{\theta^*}(x)^T]\big)}}$$

*for some absolute constants $c_3$. Here $R_n(\mathcal{G}_{\theta, \mathcal{B}})$ is the Rademacher complexity defined as*

$$R_n(\mathcal{G}_{\theta, \mathcal{B}}) = \mathbb{E}\left[\sup_{\beta \in \mathcal{B}} \frac{2}{n} \sum_{i=1}^{n} \sigma_i g_{\theta, \beta}(x_i)\right], \tag{241}$$

*where $\sigma_i$ are Rademacher random variables.*

*Proof of Lemma D.4.* With Lemma B.7 and Lemma 6.1 in hand, Lemma D.4 follows directly from Theorem 3.7 and the fact that $\tilde{\ell}$ is $2\sqrt{L}$-Lipschitz.

$\square$

With Lemma D.3 and Lemma D.4 in hand, we are now ready to prove Theorem 6.2.

*Proof of Theorem 6.2.* Note that

$$\begin{aligned}
\text{Error}_\ell(\hat{\theta}, \hat{\beta}) &= \mathbb{E}_{\theta^*, \beta^*}\left[\ell\big(g_{\hat{\theta}, \hat{\beta}}(x), y\big)\right] - \mathbb{E}_{\theta^*, \beta^*}\left[\ell\big(g_{\theta^*, \beta^*}(x), y\big)\right] \\
&= \mathbb{E}_{\theta^*, \beta^*}\left[\ell\big(g_{\hat{\theta}, \hat{\beta}}(x), y\big)\right] - \mathbb{E}_{\theta^*, \beta^*}\left[\tilde{\ell}\big(g_{\hat{\theta}, \hat{\beta}}(x), y\big)\right] \\
&\quad + \mathbb{E}_{\theta^*, \beta^*}\left[\tilde{\ell}\big(g_{\hat{\theta}, \hat{\beta}}(x), y\big)\right] - \mathbb{E}_{\theta^*, \beta^*}\left[\tilde{\ell}\big(g_{\theta^*, \beta^*}(x), y\big)\right] \\
&\quad + \mathbb{E}_{\theta^*, \beta^*}\left[\tilde{\ell}\big(g_{\theta^*, \beta^*}(x), y\big)\right] - \mathbb{E}_{\theta^*, \beta^*}\left[\ell\big(g_{\theta^*, \beta^*}(x), y\big)\right] \\
&\leq \sup_{\theta, \beta}\left\{\mathbb{E}_{\theta^*, \beta^*}\left[\ell\big(g_{\theta, \beta}(x), y\big)\right] - \mathbb{E}_{\theta^*, \beta^*}\left[\tilde{\ell}\big(g_{\theta, \beta}(x), y\big)\right]\right\} \\
&\quad + \mathbb{E}_{\theta^*, \beta^*}\left[\tilde{\ell}\big(g_{\hat{\theta}, \hat{\beta}}(x), y\big)\right] - \mathbb{E}_{\theta^*, \beta^*}\left[\tilde{\ell}\big(g_{\theta^*, \beta^*}(x), y\big)\right].
\end{aligned} \tag{242}$$

Let the truncation level be $L = 36(D^2 + 1)\log n$. By Lemma D.3 and Lemma D.4, we have

$$\begin{aligned}
&\text{Error}(\hat{\theta}, \hat{\beta}) \\
&\leq c\kappa L \cdot \sqrt{\frac{1}{m} \log \frac{N_{[\,]}\big(\mathcal{P}_{\mathcal{X} \times \mathcal{S}}(\mathcal{F}_\theta), 1/m^2\big)}{\delta}} + cL\sqrt{\frac{\log 1/\delta}{n}} + c\sqrt{L} \sup_{\theta \in \Theta} R_n(\mathcal{G}_{\theta, \mathcal{B}}) \\
&\quad + \sqrt{\frac{18(D^2 + 1)\log n}{\pi n}}.
\end{aligned} \tag{243}$$

For the Rademacher complexity, we have

$$\begin{aligned}
R_n(\mathcal{G}_{\theta, \mathcal{B}}) &= \mathbb{E}\left[\sup_{\beta \in \mathcal{B}} \frac{2}{n} \sum_{i=1}^{n} \sigma_i g_{\theta, \beta}(x_i)\right] \\
&= \mathbb{E}\left[\sup_{\beta \in \mathcal{B}} \frac{2}{n} \sum_{i=1}^{n} \sigma_i \beta^T f_\theta(x_i)\right] \\
&\leq \frac{2D}{\sqrt{n}},
\end{aligned} \tag{244}$$

where the last inequality follows from Lemma B.6. Combining (243) and (244), we have

$$
\begin{aligned}
&\mathrm{Error}(\hat{\theta}, \hat{\beta}) \\
&\leq c\kappa L \cdot \sqrt{\frac{1}{m} \log \frac{N_{[\,]}(\mathcal{P}_{\mathcal{X} \times \mathcal{S}}(\mathcal{F}_\theta), 1/m^2)}{\delta}} + cL\sqrt{\frac{\log 1/\delta}{n}} + 2cD\sqrt{\frac{L}{n}} \\
&\quad + \sqrt{\frac{18(D^2+1)\log n}{\pi n}} \\
&= \tilde{\mathcal{O}}\left(\kappa L\sqrt{\frac{\log N_{[\,]}(\mathcal{P}_{\mathcal{X} \times \mathcal{S}}(\mathcal{F}_\theta), 1/m^2)}{m}} + L\sqrt{\frac{1}{n}}\right),
\end{aligned}
\tag{245}
$$

where $L = 36(D^2+1)\log n$ and

$$
\kappa = c_3 \sqrt{\frac{1}{\sigma_{\min}\left(\mathbb{E}[f_{\theta^*}(x)f_{\theta^*}(x)^T]\right)}}
$$

for some absolute constants $c_3$. $\qquad\square$

## E  FAILURE OF TWO-PHASE MLE

For simplicity, in the sequel, we consider the case where no side information is available, i.e., we have access to unlabeled data $\{x_i\}_{i=1}^m$ and labeled data $\{x_j, y_j\}_{j=1}^n$. Another natural scheme is to use a two-phase MLE (Algorithm 2). To be specific, in the first phase, we use MLE to estimate $\phi^*$ based on the unlabeled data $\{x_i\}_{i=1}^m$. In the second phase, we use MLE again to estimate $\psi^*$ based on pretrained $\hat{\phi}$ and the labeled data $\{x_j, y_j\}_{j=1}^n$.

---

**Algorithm 2** Two-phase MLE

1: **Input:** $\{x_i\}_{i=1}^m, \{(x_j, y_j)\}_{j=1}^n$
2: Use unlabeled data $\{x_i\}_{i=1}^m$ to learn $\hat{\phi}$ via MLE:

$$
\hat{\phi} \leftarrow \arg\max_{\phi \in \Phi} \sum_{i=1}^m \log p_\phi(x_i).
$$

3: Fix $\hat{\phi}$ and use labeled data $\{(x_j, y_j)\}_{j=1}^n$ to learn $\hat{\psi}$ via MLE:

$$
\hat{\psi} \leftarrow \arg\max_{\psi \in \Psi} \sum_{j=1}^n \log p_{\hat{\phi}, \psi}(x_j, y_j).
$$

4: **Output:** $\hat{\phi}$ and $\hat{\psi}$.

---

Note that the two-phase MLE does not directly associate the learning process with the loss function. Thus, the only way to evaluate the excess risk is to study the total variation distance between $\mathbb{P}_{\hat{\phi}, \hat{\psi}}(x, y)$ and $\mathbb{P}_{\phi^*, \psi^*}(x, y)$. In the pretraining phase, MLE guarantees that the estimator $\mathbb{P}_{\hat{\phi}}$ is close to $\mathbb{P}_{\phi^*}$ in the sense of total variation distance (Theorem 3.3). However, it's still possible that for some $x$, $\mathbb{P}_{\hat{\phi}}(x) = 0$ while $\mathbb{P}_{\phi^*}(x) \neq 0$. This phenomenon may result in $\log p_{\hat{\phi}, \psi^*}(x_j, y_j) = -\infty$ for some labeled data in the learning of downstream tasks, which will dramatically influence the behaviour of MLE for estimating $\psi^*$ and finally lead to the failure of the second phase. Inspired by this idea, we give the following theorem.

**Theorem E.1.** *There exists $\Phi, \Psi, \phi^* \in \Phi, \psi^* \in \Psi$, such that for any constant $c > 0$, there exists $m, n \geq c$ such that with probability at least $\frac{1}{2}(1 - e^{-1})e^{-1}$, we have*

$$
d_{\mathrm{TV}}\left(\mathbb{P}_{\hat{\phi}, \hat{\psi}}(x, y), \mathbb{P}_{\phi^*, \psi^*}(x, y)\right) \geq \frac{1}{8},
$$

*where $\hat{\phi}$ and $\hat{\psi}$ are the outputs of Algorithm 2.*

*Proof of Theorem E.1.* We construct the counter example as follows. Let $(x, y, z) \in \mathbb{N}_+ \times \mathbb{N}_+ \times \mathbb{N}_+$. We assume that the true parameter $(\phi^*, \psi^*) = (\phi_1, \psi_1)$, which satisfies

$$\mathbb{P}_{\phi_1}(x = k, z = k) = \frac{1}{2^k} \ \forall k \in \mathbb{N}_+, \quad \mathbb{P}_{\phi_1}(x = m, z = n) = 0 \ \forall m \neq n,$$

$$\mathbb{P}_{\psi_1}(y = k | z = k) = 1, \ \forall k \in \mathbb{N}_+.$$

For $i \geq 2$, we define $\mathbb{P}_{\phi_i}$ as follows,

$$\mathbb{P}_{\phi_i}(x = 1, z = 1) = \frac{1}{2} + \frac{1}{2^i}, \quad \mathbb{P}_{\phi_i}(x = k, z = k) = \frac{1}{2^k} \ \forall k \notin \{1, i\}$$

$$\mathbb{P}_{\phi_i}(x = m, z = n) = 0 \ \forall m \neq n \text{ or } m = n = i.$$

We define $\mathbb{P}_{\psi_2}$ as follows, for any $k \in \mathbb{N}_+$,

$$\mathbb{P}_{\psi_2}(y = 1 | z = k) = \frac{1}{4}, \quad \mathbb{P}_{\psi_2}(y = 2 | z = k) = \frac{1}{2}$$

$$\mathbb{P}_{\psi_2}(y = j | z = k) = \frac{1}{2^j} \ \forall j \notin \{1, 2\}.$$

We denote $\Phi := \{\phi_i \mid i \in \mathbb{N}_+\}$ and $\Psi := \{\psi_1, \psi_2\}$. In the sequel, we show that Algorithm 2 fails on this case. Recall that we denote by $\{x_i\}_{i=1}^m$ and $\{x_j, y_j\}_{j=1}^n$ the unlabeled data and labeled data, respectively. We have the following observations:

- We define $i := \min\{k \neq 1 \mid k \notin \{x_i\}_{i=1}^m\}$. If we have $1 \in \{x_i\}_{i=1}^m$, then the maximizer of likelihood function $\hat{\phi}$ satisfies $\hat{\phi} = \phi_i$.

- Suppose that $\hat{\phi} = \phi_i$ for some $i \neq 1$ and $i \in \{y_j\}_{j=1}^n$. We then have $\hat{\psi} = \psi_2$.

We define the event $\mathcal{E} := \{\exists i \neq 1, \text{ such that } \hat{\phi} = \phi_i \text{ and } i \in \{y_j\}_{j=1}^n\}$. Under event $\mathcal{E}$, we have $\hat{\phi} = \phi_i$ for some $i \neq 1$ and $\hat{\psi} = \psi_2$, which implies

$$
\begin{aligned}
d_{\mathrm{TV}}\big(\mathbb{P}_{\hat{\phi}, \hat{\psi}}(x, y), \mathbb{P}_{\phi^*, \psi^*}(x, y)\big) &= \frac{1}{2} \int \int |p_{\phi_i, \psi_2}(x, y) - p_{\phi_1, \psi_1}(x, y)| \, dx dy \\
&\geq \frac{1}{2} \int \left| \int p_{\phi_i, \psi_2}(x, y) - p_{\phi_1, \psi_1}(x, y) \, dx \right| dy \\
&= \frac{1}{2} \int |p_{\phi_i, \psi_2}(y) - p_{\phi_1, \psi_1}(y)| \, dy \\
&\geq \frac{1}{2} |\mathbb{P}_{\phi_i, \psi_2}(y = 2) - \mathbb{P}_{\phi_1, \psi_1}(y = 2)| = \frac{1}{8}
\end{aligned}
\tag{246}
$$

In the following, we only need to lower bound the probability of event $\mathcal{E}$. Note that

$$
\begin{aligned}
\mathbb{P}(\mathcal{E}) &= \mathbb{P}\big( \cup_{i=2}^\infty \{\hat{\phi} = \phi, i \in \{y_j\}_{j=1}^n\}\big) \\
&= \sum_{i=2}^\infty \mathbb{P}\big(\hat{\phi} = \phi_i, i \in \{y_j\}_{j=1}^n\big) \\
&= \sum_{i=2}^\infty \mathbb{P}\big(\hat{\phi} = \phi_i\big) \cdot \mathbb{P}\big(i \in \{y_j\}_{j=1}^n\big) \\
&= \sum_{i=2}^\infty \left(1 - \left(1 - \frac{1}{2^i}\right)^n\right) \cdot \mathbb{P}\big(\hat{\phi} = \phi_i\big).
\end{aligned}
\tag{247}
$$

Thus, it holds for any $L \geq 2$ that

$$
\begin{aligned}
\mathbb{P}(\mathcal{E}) &\geq \sum_{i=2}^L \left(1 - \left(1 - \frac{1}{2^i}\right)^n\right) \cdot \mathbb{P}\big(\hat{\phi} = \phi_i\big) \\
&\geq \left(1 - \left(1 - \frac{1}{2^L}\right)^n\right) \cdot \mathbb{P}\big(\exists 2 \leq i \leq L, \hat{\phi} = \phi_i\big).
\end{aligned}
\tag{248}
$$

Note that

$$
\begin{aligned}
&\mathbb{P}\big(\exists 2 \leq i \leq L, \hat{\phi} = \phi_i\big) \\
&= \mathbb{P}\Big(\big\{1 \in \{x_i\}_{i=1}^m\big\} \cap \big\{\exists 2 \leq i \leq L, i \notin \{x_i\}_{i=1}^m\big\}\Big) \\
&\geq \mathbb{P}\Big(\big\{1 \in \{x_i\}_{i=1}^m\big\} \cap \big\{L \notin \{x_i\}_{i=1}^m\big\}\Big) \\
&\geq \mathbb{P}\big(1 \in \{x_i\}_{i=1}^m\big) + \mathbb{P}\big(L \notin \{x_i\}_{i=1}^m\big) - 1 \\
&= \mathbb{P}\big(L \notin \{x_i\}_{i=1}^m\big) - \mathbb{P}\big(1 \notin \{x_i\}_{i=1}^m\big) \\
&= \left(1 - \frac{1}{2^L}\right)^m - \frac{1}{2^m}.
\end{aligned}
\tag{249}
$$

Combining (248) and (249), we have for any $L \geq 2$

$$
\mathbb{P}(\mathcal{E}) \geq \left(1 - \left(1 - \frac{1}{2^L}\right)^n\right) \cdot \left(\left(1 - \frac{1}{2^L}\right)^m - \frac{1}{2^m}\right).
\tag{250}
$$

Setting $2^L = m = n$, we obtain that

$$
\mathbb{P}(\mathcal{E}) \geq \left(1 - \left(1 - \frac{1}{m}\right)^m\right) \cdot \left(\left(1 - \frac{1}{m}\right)^m - \frac{1}{2^m}\right) \to (1 - e^{-1}) \cdot e^{-1}, \text{ as } m \to \infty.
\tag{251}
$$

Thus, for any $c > 0$, there exists $m, n \geq c$ such that

$$
\mathbb{P}(\mathcal{E}) \geq \frac{1}{2}(1 - e^{-1}) \cdot e^{-1}.
$$

$\square$