# OpenReview forum: "On the Provable Advantage of Unsupervised Pretraining"
_ICLR.cc/2024/Conference — ICLR 2024 spotlight_

### Official Review · Reviewer_Ht6r · 2023-11-01

**Soundness:** 4 excellent
**Presentation:** 4 excellent
**Contribution:** 2 fair
**Rating:** 6
**Confidence:** 3

**Summary:**

This paper sets out to provide a theoretical foundation for the benefits of pretraining on unsupervised data in the context of a broader downstream task. The central premise is that an unknown distribution, denoted as $p_{\phi^*}$ and belonging to a known distribution family $\phi^*\in\Phi$, generates pairs of data points $(x,z)$, where $x$ represents the input vector, and $z$ is a latent variable. Labels, or $y$ values, are stochastically determined solely by $z$ through an unknown distribution $p_{\psi^*}$ from a known distribution family $\psi^*\in\Psi$. Consequently, the pretraining procedure utilizes only observations of $x_j$s and Maximum Likelihood Estimation (MLE) to acquire knowledge of $p(z|x)$, while the fine-tuning stage leverages labeled pairs $(x_i,y_i)$ to learn a straightforward classifier, $p(y|z)$.

Notably, the authors introduce the concept of "informativeness" as a means to theoretically elucidate the learnability of $p(z|x)$ solely from observations of $x_j$s. The paper then goes on to present a set of generalization bounds that, while somewhat expected, offer a fresh perspective. These bounds are demonstrated in three distinct scenarios: 1) Factor models, 2) Gaussian Mixture Models, and 3) Contrastive Learning.

The mathematical analysis in this paper appears rigorous and well-founded, and the presented bounds seem to be a novel contribution. However, my skepticism arises regarding the extent to which this paper advances the theoretical understanding of pretraining, as discussed in the "Weaknesses" section. It's noteworthy that the improved generalization bounds are primarily a result of the significant assumptions made about the data generation process. In simpler terms, the authors first assumed the efficacy of pretraining, and then substantiated their own assumption.

Nonetheless, I believe that this paper could be a valuable addition to the Machine Learning Theory (MLT) community and might meet the standards for acceptance at ICLR. Therefore, my current inclination is toward a weak acceptance. However, I remain open to reconsidering my evaluation if the authors can more effectively emphasize the significance of their work.

**Strengths:**

- The paper is well-structured and exceptionally clear, making it easy to read. The problem formulation and subsequent results are presented with great clarity. The mathematical derivations appear sound, and the results are in line with expectations.

- The introduction of the concept of $\kappa$-informativeness, which guarantees the learnability of $p(x,z)$ using only observations from $x$, is both innovative and intriguing.

- The authors have skillfully and rigorously framed their problem and have effectively demonstrated their results in three distinct theoretical scenarios: 1) Factor models, 2) Gaussian Mixture Models, and 3) Contrastive Learning.

**Weaknesses:**

The authors present an analysis of the excess risk associated with a two-stage learning process: first pretraining on a large volume of unlabeled data ($m\gg n$), followed by fine-tuning on a smaller labeled dataset. Their findings reveal that this excess risk is notably smaller compared to training solely on the initial set of $n$ labeled samples. While the presented bounds are mathematically rigorous and insightful, they do not significantly advance our theoretical understanding of the concept of pretraining per se. In essence, the generalization bounds align with conventional expectations: For a limited $n$ and assuming $m\rightarrow\infty$, the training of a small linear head generlizes far better than training the whole NN, therefore, one would naturally expect:
$\sqrt{\mathrm{complexity}(head)/n}+\sqrt{\mathrm{complexity}(body)/m}\ll \sqrt{\mathrm{complexity}(head+body)/n}.$

The authors, in my opinion, do not definitively establish the "advantage" of pretraining, despite the paper's title. In my opinion, the efficacy of unsupervised pretraining primarily stems from its remarkable capacity to mitigate bias, rather than reducing the excess risk. However, the effect of bias is not present in this analysis. The authors postulate the existence of a latent variable, denoted as $z$, which imparts conditional independence between $x$ and $y$. This assumption underpins the essence of pretraining, but the underlying reasons for its practical effectiveness remain elusive. Additionally, the authors rely on the presumption that the "true" joint distribution of $x$ and $z$ adheres to a known class of probability distributions, typically those produced by sufficiently large neural networks, which can be learned using a generic estimator such as MLE. The question of why these assumptions hold in practice and whether they can be theoretically justified remains a mystery, and this study does not provide a resolution to this critical issue.

Due to the aforementioned ungrounded assumptions, particularly in the context of theoretical rigor, the proofs and associated mathematical methodologies in this paper lack enough mathematical sophistication. The derived bounds rely on the application of well-established generalization guarantees pertaining to Empirical Risk Minimization (ERM) and Maximum Likelihood Estimator (MLE).

In light of these concerns, I would suggest to rephrase both the title and abstract of the paper to accurately reflect the highlighted issues.

**Questions:**

For me, Theorem 5.3 is of independent interest. In your work, the data generation model is assumed to belong to a known distribution family, such as a Gaussian Mixture Model. This setting bears a resemblance to Ashtiani et al. (2018)'s work, where they provided slightly improved bounds for learning Gaussian mixture models, albeit not within the context of pretraining.

Following Ashtiani et al., and adopting the notations used in the present work, the error for the downstream classification task can be bounded as follows:


$\hat{\mathrm{Error}}\leq \mathrm{Error}_{optimal} + \mathcal{O}(\sqrt{Kd/m})$

without the term $\mathcal{O}(\sqrt{K/n})$ which appears in Theorem 5.3 of the present paper. Instead, Ashtiani et al. would require $n\ge\tilde{\mathcal{O}}(K)$ (with $\tilde{\mathcal{O}}(\cdot)$ hiding poly-logarithmic factors).

However, as previously mentioned, their focus is not on pretraining, and they pursue a fundamentally different approach. Still, it would be beneficial if the authors could provide further insights into the parallels between these works and offer a comparative analysis of the bounds.

- Ashtiani, Hassan, et al. "Nearly tight sample complexity bounds for learning mixtures of gaussians via sample compression schemes." Advances in Neural Information Processing Systems 31 (2018).

---

> ### Author Response · Authors · 2023-11-22
> **Author Response**
>
> We thank the reviewer for the overall positive reviews.
>
> **Q: The effect of “Bias” in pretraining.**
>
> A: If the reviewer means “mitigate bias” by formally proving that the population loss of the best predictor within the function class is reduced, this is in general rather difficult to control in theory because it involves the unknown data distribution—mathematically describing what is the distribution of natural image/language by itself is a challenging problem. Few theories provide this type of results beyond toy examples.
>
> On the other hand, a practical principle to reduce “bias” is simply by increasing model capacity and using more expressive models. When presented with a large amount of unlabeled data, our theoretical results in fact permit pretraining a rather complex model for representation without paying large generalization error. In this sense, our result indirectly reflects the “mitigate bias” aspect of pretraining.
>
>
> **Q: Justification of assumption**
>
> A: This paper aims to identify the structural condition which allows unsupervised pretraining to be beneficial to learning downstream tasks. Information theoretically speaking, pretraining is only helpful when pretraining tasks “correlates” with downstream tasks. In this paper, we follow the common principle that has been used by a majority of empirical papers, and model this correlation through an intermediate quantity — the representation $z$. One contribution of this paper is to give a mathematical quantification of this “correlation”. Therefore, we view this as a natural approach to make a first attempt in solving this general problem.
>
> We quantify the correlation using density estimation and TV distance, which gives rise to informative conditions (Assumption 3.2). We agree that Assumption 3.2 is only a sufficient condition — weaker conditions possibly exist. We view this as a stepping stone towards identifying richer and more general conditions for this important problem in the future.
>
>
> **Q: relation to Ashtiani et al. (2018)**
>
> A: Thanks for pointing this out. The main difference and similarity between our results of GMM and those of Ashtiani et al. (2018) are as the following:
>
> Difference: Ashtiani et al. (2018) does not involve any downstream tasks, while our analysis for GMM involves certain downstream tasks. To be more specific, Ashtiani et al. (2018) only considers the estimation of the marginal distribution of $x$ in GMM, which does not directly give results for learning representation $z$. Neither can it be directly compared with our Theorem 5.3 where a binary classification downstream task is involved.
>
> Similarity: Ashtiani et al. (2018) and our analysis both propose an algorithm that can find a distribution of $x$ whose TV distance to the ground truth distribution is within $\\tilde{O}(\\sqrt{\\frac{dK}{m}})$. As one of our intermediate result, our analysis achieves the same bound as Ashtiani et al. (2018) for estimating the marginal distribution of $x$: our Lemma C.5 and Theorem 3.3 together give the TV distance bound $\\tilde{O}(\\sqrt{\\frac{dK}{m}})$.
> We will add this discussion in the revised version.

---

### Official Review · Reviewer_rxwf · 2023-11-02

**Soundness:** 4 excellent
**Presentation:** 3 good
**Contribution:** 2 fair
**Rating:** 8
**Confidence:** 4

**Summary:**

The paper introduces an algorithm that under a specified "informative" condition significantly improves the baseline complexity achieved by conventional supervised learning, especially when there is a large amount of unlabeled data (m) compared to labeled data (n). The authors present a versatile framework to analyze various unsupervised pretraining models, emphasizing its applicability across different setups like factor models, Gaussian mixture models, and contrastive learning. The primary contributions are:

Excess Risk Bound: The authors present an upper bound on the excess risk associated with an algorithm (Algorithm 1). The risk is expressed in terms of various parameters, including the Rademacher complexity—a measure that quantifies the function class's capacity to fit random noise.

Weakly Informative Models: The paper introduces the concept of models that are $\kappa^{-1}$-weakly-informative. These models, while not being fully informative, are shown to have certain desirable properties. The authors provide a relaxed assumption (Assumption 3.6) which broadens the class of examples by considering both the Total Variation (TV) and Hellinger distances to quantify the discrepancy between the model's outputs and the true distribution.

Performance Guarantees: Crucially, the authors demonstrate that the bounds provided on excess risk remain valid even under the $\kappa^{-1}$-weakly-informative assumption (Theorem 3.7). This insight is significant as it indicates that despite the potential shortcomings of weakly informative models, one can still establish robust performance guarantees for them.

**Strengths:**

**Originality:**
The paper's introduction of assumption 3.2 showcases a unique approach in how it lays out its foundational groundwork. This can be seen as a new definition or problem formulation, adding to the originality of the work.

**Quality:**
The research stands out in its rigor and the depth of its theoretical contributions. The introduction and use of explicit models that unravel all hidden parameters, particularly $\kappa$, underline the paper's commitment to thoroughness and depth.

**Clarity:**
One of the standout features of this paper is its clarity. The authors smoothly introduce assumption 3.2 via a preceding thought experiment. This structured approach enhances comprehensibility, ensuring that readers are well-prepped before delving into more complex concepts. Furthermore, the theorems provided give a clear indication of the magnitude of parameters involved, aiding readers in understanding the practical implications and boundaries of the proposed methods.

**Significance:**
The paper's structured and well-organized approach to presenting the setup, theorem, and explicit topic application is commendable. Such an approach not only amplifies the paper's readability but also underscores its significance in serving as a potential benchmark for future works in the domain. By ensuring that readers can seamlessly transition through the content, the authors have amplified the paper's potential impact on its audience.

**Weaknesses:**

### Weaknesses:

**Over-reliance on Assumption 3.2:**
The primary weakness of this paper stems from assumption 3.2, which appears to be an overly robust condition. The core challenge of the paper lies in discerning the (constraint) relationship between (x, s) and (x, z). By introducing a new parameter, $\kappa$, to simply constrain this relationship, the paper sidesteps the intricacy of this challenge. While the existence of such a parameter in the three sub-cases presented in the paper is addressed, its presence in more complex scenarios remains questionable. The use of such a strong assumption might detract from the general applicability of the results.

**Potential Pitfalls with Transformation Assumption:**
A significant concern arises from the implications of assumption 3.2. Based on the assumption, it can be inferred that given
$$\mathbb{P}\_{\phi}(x, s) = \mathbb{P}\_{\phi^*}(x, s),$$
there exists a transformation $T_1$ such that
$$\mathbb{P}\_{T\_1 \circ \phi}(x, z) = \mathbb{P}\_{\phi^*}(x, z).$$

However, when considering mixed uniform distributions over specific cubical sets, these assumptions are immediately invalidated. This limitation highlights potential scenarios where the stated assumption may not be practically applicable.

**Recommendations for Improvement:**
It would be beneficial for the authors to provide a more thorough justification or relaxation of assumption 3.2, possibly exploring weaker conditions that achieve similar results.

**Questions:**

**Real-world Relevance**: How do the assumptions and theorems presented in the paper translate to real-world applications? Are there any practical scenarios where these findings could be directly applied?

---

> ### Author Response · Authors · 2023-11-22
> **Author Response**
>
> We thank the reviewer for the positive reviews.
>
> **Q1: Over-reliance on Assumption 3.2**
>
> A: While most prior works focused on specific applications, this paper aims to identify generic structural condition which allows unsupervised pretraining to be beneficial to learning downstream tasks. We view informative conditions (assumption 3.2) as one of the first steps towards understanding in general scenarios. We agree that assumption 3.2 is only a sufficient condition, and identifying weaker assumptions is an interesting future direction.
>
> **Q2: Potential Pitfalls**
>
> A: If we understand the example correctly, we can consider 1-dimensional case, reviewer is saying that for (a) a mixture of uniform distribution over [0, 1] and [1, 2] with mixing weights 0.5 and 0.5, versus (b) a mixture of uniform distribution over [0, 0.5] and [0.5, 2] with mixing weights 0.25 and 0.75; two models have different joint distribution of $(x, z)$ but have the same marginal distribution on $x$, so our assumption is violated. However, we would argue that our assumption in this setting is violated for a good reason: as unsupervised pretraining only observes $x$, it has no ability to distinguish (a) model vs (b) model, that is, unsupervised pretraining will not be helpful in this situation.
>
>
> **Q3: Practical applications**
>
> A: The results of this paper (a) can be directly applied to a number of practical applications; and (b) can be used to provide strong insights to certain advanced practical problems. For (a), for instance, the factor model example in this paper has been widely used in statistics, dealing with dependent measurements in real world applications including genomics, neuroscience, economics, finance, etc [1]. For (b), our setting directly fits the standard pretraining setup (e.g., in large language models) which uses a large amount of unlabeled data to pretrain a deep neural network as a representation, and then uses a small amount of labeled data to only fine-tunes the linear head for downstream tasks. This paper considers the MLE algorithm for pretraining, which provides insights to the success of many practical algorithms that use proxies for MLE (such as optimizing the variational lower bound, as in the case of variational autoencoder) due to computational constraints.
>
> [1] Fan, J., Wang, K., Zhong, Y., & Zhu, Z. (2021). Robust high dimensional factor models with applications to statistical machine learning. Statistical science: a review journal of the Institute of Mathematical Statistics, 36(2), 303.

---

> > ### Comment · Reviewer_rxwf · 2023-11-23
> > **Concerns Regarding Assumption 3.2**
> >
> > Dear Authors,
> >
> > Thank you for your detailed responses. Your work indeed contributes significantly to our understanding of unsupervised pretraining in learning downstream tasks. However, I would like to further discuss Assumption 3.2 and its implications in practical scenarios.
> >
> > In your response, you acknowledged that Assumption 3.2 is a sufficient condition, and identifying weaker assumptions remains an open challenge. To illustrate a potential limitation of this assumption, consider two distributions, $D_1$ and $D_2$, defined as follows:
> >
> > - $D_1$:
> >   - Uniform over [-4, -2] with probability 1/4,
> >   - Uniform over [-3, -1] with probability 1/4,
> >   - Uniform over [1, 3] with probability 1/4,
> >   - Uniform over [2, 4] with probability 1/4.
> >
> > - $D_2$:
> >   - Uniform over [-4, -1] with probability 1/4,
> >   - Uniform over [-3, -2] with probability 1/4,
> >   - Uniform over [1, 4] with probability 1/4,
> >   - Uniform over [2, 3] with probability 1/4.
> >
> > Under these distributions, even though it remains challenging to distinguish between $D_1$ and $D_2$, self-supervised learning can still effectively learn if the label $y$ is, for instance, the sign of the number. This scenario highlights a potential drawback of Assumption 3.2: it does not account for situations where labels can provide critical distinctions not captured by the unsupervised pretraining process.
> >
> > This example suggests that while Assumption 3.2 might be sufficient in certain contexts, it might not encapsulate all scenarios where self-supervised learning can be beneficial. This insight aligns with your acknowledgement that exploring weaker or alternative assumptions could be a promising future direction. It might be beneficial to discuss or at least acknowledge these potential limitations and scenarios in your paper, as it would provide a more comprehensive understanding of the implications and applicability of your findings.
> >
> > Thank you for considering this feedback. I look forward to your continued contributions to this important field.

---

> > > ### Author Response · Authors · 2023-11-23
> > > **Author Response**
> > >
> > > Thanks for your valuable feedback! We will add this discussion in the revised version.

---

### Official Review · Reviewer_9yFE · 2023-11-05

**Soundness:** 4 excellent
**Presentation:** 4 excellent
**Contribution:** 3 good
**Rating:** 8
**Confidence:** 2

**Summary:**

This paper provides a generalization of and subsequent analysis for unsupervised pretraining. The provided framework is able to capture a wide variety of unsupervised pretraining methods, of which factor models, Gaussian mixture models, and contrastive learning are more closely analyzed. The main result is a proof that the excess risk of the provided unsupervised algorithm scales by at most $\tilde{\mathcal{O}}(\sqrt{C_\Phi/m} + \sqrt{C_\Psi/n})$, which compares favorably to a baseline supervised learning excess risk under certain regimes.

**Strengths:**

The presentation is clear. The generic framework provided for unsupervised learning is able to capture a wide variety of unsupervised methods. The theoretical results are able to demonstrate some advantage compared to a baseline supervised learning method.

**Weaknesses:**

Practical applications are unclear due to assumptions made (such as bounded loss) and the large complexity terms in the bounds. Idealized versions of MLE and ERM are used.

**Questions:**

The truncation bounds used for the square loss seem very unnatural. Could you explain the use of those particular bounds?

All of the examples are in relatively simple settings (linear functions, separated Gaussian mixtures). Is this due to the way the bounds scale with model/distribution complexity?

Could you explain how you derive the excess risk for the baseline supervised algorithm?

---

> ### Author Response · Authors · 2023-11-22
> **Author Response**
>
> We thank the reviewer for the positive reviews.
>
> **Q1: explain the use of truncation bounds for square loss**
>
> A: While a large number of learning theories apply to bounded functions, squared loss with Gaussian noise is technically not bounded. Fortunately, for light-tailed random variables such as Gaussian, the unbounded scenario happens with very low probability. It is a standard statistical treatment to use truncation to properly manage the low probability of encountering unboundedness, without affecting the high probability scenario where the random variable behaves well and remains bounded.
>
> **Q2: examples being relatively simple**
>
> A: We choose these representative examples to illustrate the core ideas and showcase the applicability of our generic framework. While our theoretical framework does apply to more advanced/complex applications, the computation of model complexity and informative condition coefficients of these applications can be quite involved and technical, which may qualify for a separate paper by themselves (we note for instance, there are many papers in the literature focusing on just computing the Rademacher complexity, or PAC bayes bound for certain architecture of neural networks). We leave those as exciting future works.
>
>
> **Q3: excess risk for the baseline supervised algorithm**
>
> The excess risk bound for baseline supervised algorithm follows from standard risk bound (see, e.g., Section 4.1.2 and Theorem 4.10 in [1]) in supervised learning by plugging in $G_{\Phi,\Psi}$ (defined at the end of page 3 of our paper) as the function class.
>
> [1]Wainwright, Martin J. High-dimensional statistics: A non-asymptotic viewpoint. Vol. 48. Cambridge university press, 2019.

---

### Official Review · Reviewer_Qu76 · 2023-11-10

**Soundness:** 4 excellent
**Presentation:** 3 good
**Contribution:** 3 good
**Rating:** 6
**Confidence:** 3

**Summary:**

This paper develops a framework for understanding the advantages of unsupervised pretraining. The paper factorizes the learning task into two parts: unsupervised pretraining by MLE and supervised learning for downstream tasks via ERM. This paper derives its risk bound via an informative assumption justified by a zero-informative counter-example that any unsupervised pretraining gives no further information. Based on that assumption, the core result is an excess risk bound of $\tilde{O}(\sqrt{\text{Complexity of pretraining class}/\text{Unsupervised pretraining samples}} + \sqrt{\text{Complexity of downstream class}/\text{Supervised learning samples}})$. This result prevails the risk guarantee of sole supervised learning when provided with abundant unsupervised pretraining samples. The authors also extend their results with three concrete examples.

**Strengths:**

1. This paper is well-written and easy to follow.

2. This paper provides a general framework for analyzing the benefits of unsupervised pretraining. The "MLE+ERM" approach covers most examples in unsupervised pretraining. The authors also justify the ERM approach by constructing an MLE counter-example.

3. The main result is impressive and natural. Breaking supervised learning's statistical learning bound into two parts is critical to understanding the advantage of unsupervised pretraining.

**Weaknesses:**

1. The main assumption (Assumption 3.2) is not fully justified. The zero-informative counter-example is solid, but justifying the informative condition by showing "if some example is zero-informative, then the unsupervised pretraining could be useless" is a tautology. In short, the paper's conclusion of the superiority of unsupervised pretraining is built on the presumption of believing the pretraining to be useful.

2. This paper shows three instances for its framework: factor models, Gaussian mixture models, and contrastive learning. But the last needs more careful handling since it is usually considered a self-supervised learning task.

3. The contribution of this paper is mainly on the modeling part, while the proof techniques seem standard,

**Questions:**

1. I wonder if the authors could comment more on justifying Assumption 3.2. I don't mean that "Assumption 3.2 is incorrect"; rather I just want a bit more demonstration on the mechanics behind this informative condition. For example, the authors compute three examples' informative conditions, but are those constants essential to the problem? Any theoretical results (for example, lower bounds of $\Omega(\kappa)$) or empirical ones will do.

2. The third example, contrastive learning (Section 6), seems to differ from the mainstream contrastive learning modeling. A common approach is to assume access to positive pairs $(x_i, x_i^+)$ from the same latent class and negative pairs $(x_i, x_i^-)$, while this paper models it in a somewhat "reverse" way: each sample of the pair $(x, x^\prime)$ is i.i.d. (which is often not satisfied in practice; see HaoChen et al. 2021), and the positive/negative signs are logistically distributed according to their latent similarity. I am not sure if this model is representative of a standard contrastive learning model, and it would be great if the authors could provide more explanations.

---

> ### Author Response · Authors · 2023-11-22
> **Author Response**
>
> We thank the reviewer for the overall positive reviews.
>
> **Q1: Justification of main assumption (Assumption 3.2), lower bound**
>
> A: This paper aims to identify the structural condition which allows unsupervised pretraining to be beneficial to learning downstream tasks. Information theoretically speaking, pretraining is only helpful when pretraining tasks “correlates” with downstream tasks. In the setup of this paper, this means pretraining objectives must “correlate” with learning representation $z$ (since $z$ is the intermediate quantity connects pretraining to downstream). One contribution of this paper is to give a natural, mathematical quantification of this “correlation”. We did so using density estimation and TV distance, which gives rise to informative conditions (Assumption 3.2). We agree that Assumption 3.2 is only a sufficient condition — weaker conditions possibly exist. We view this as a first step towards identifying richer and more general conditions for this important problem in the future.
>
> While we do not yet have formal lower bound on $\kappa$, we would like to argue that certain $\kappa$ dependency is necessary in risk bound $\kappa \sqrt{\frac{C_{\Phi}}{m}}+ \sqrt{\frac{C_{\Psi}}{n}}$ in worst cases. That is, the alternative risk bound $\sqrt{\frac{C_{\Phi}}{m}}+ \sqrt{\frac{C_{\Psi}}{n}}$ without $\kappa$ is not possible: we still consider the zero-informative example where $z=Ax$ + noise and $x$ is sampled from standard Gaussian with matrix $A$ unknown. In this case, the unsupervised pretraining step only observes the distribution of $x$, which information theoretically does not reveal anything about $A$ and representation $z$. We assume the downstream task is very simple — the map from $z$ to label $y$ is known (which can be realized by choosing a function class that only contains a single function; this makes $C_{\Psi} = 0$ ). We consider the case where the number of unlabeled data $m$ is extremely large and goes to infinity. Then the alternative risk bound would imply that even without any labeled data, we can still achieve 0 excess risk. We know this is not possible since we still need to learn $A$ from noisy labeled data.
>
> **Q2: Contrastive learning model is slightly different from HaoChen et al. 2021)**
>
> A: We view contrastive learning as a general class of approaches to learning that focuses on extracting meaningful representations by contrasting positive and negative pairs of instances. Our model captures this core idea, and fits this description. While we agree that our model would generate contrastive loss that is slightly different from the loss used in practice (such as in CLIP), such slight mismatch is not uncommon in theory work due to the requirement of rigorous math treatment — the difference in loss also exists for prior theoretical work for contrastive learning such as Haochen et al. and [1]. We believe that bridging this gap, and consider theory that involves practical contrastive loss is an important future direction.
>
> [1] Saunshi, N., Plevrakis, O., Arora, S., Khodak, M. and Khandeparkar, H., 2019, May. A theoretical analysis of contrastive unsupervised representation learning. In International Conference on Machine Learning (pp. 5628-5637). PMLR.

---

### Meta-Review · Area_Chair_Tkg8 · 2023-12-08

**Metareview:**

This paper provides new results on the theory of unsupervised pretraining, and was unanimously scored as an accept by reviewers.

**Justification For Why Not Higher Score:**

Uniform support from reviewers, but maybe not groundbreaking work.

**Justification For Why Not Lower Score:**

Uniform support from reviewers.

---

### Decision · Program_Chairs · 2024-01-16

Accept (spotlight)